# CDK4 inactivation inhibits apoptosis via mitochondria-ER contact remodeling in triple-negative breast cancer

Dorian V. Ziegler [1], Kanishka Parashar [1], Lucia Leal-Esteban[1], Jaime López-Alcalá [1,2], Wilson Castro [3], Nadège Zanou [4], Laia Martinez-Carreres[1], Katharina Huber[1], Xavier Pascal Berney[1], María M. Malagón [2,5], Catherine Roger[1], Marie-Agnès Berger[6], Yves Gouriou [6], Giulia Paone[1], Hector Gallart-Ayala [7], George Sflomos[8], Carlos Ronchi [8], Julijana Ivanisevic [7], Cathrin Brisken [8,9], Jennifer Rieusset[6], Melita Irving [3] & Lluis Fajas [1,10] ✉

The energetic demands of proliferating cells during tumorigenesis require close coordination between the cell cycle and metabolism. While CDK4 is known for its role in cell proliferation, its metabolic function in cancer, particularly in triple-negative breast cancer (TNBC), remains unclear. Our study, using genetic and pharmacological approaches, reveals that CDK4 inactivation only modestly impacts TNBC cell proliferation and tumor formation. Notably, CDK4 depletion or long-term CDK4/6 inhibition confers resistance to apoptosis in TNBC cells. Mechanistically, CDK4 enhances mitochondria-endoplasmic reticulum contact (MERCs) formation, promoting mitochondrial fission and ER-mitochondrial calcium signaling, which are crucial for TNBC metabolic flexibility. Phosphoproteomic analysis identified CDK4's role in regulating PKA activity at MERCs. In this work, we highlight CDK4's role in mitochondrial apoptosis inhibition and suggest that targeting MERCs-associated metabolic shifts could enhance TNBC therapy.

Cell cycle regulators are key factors for cell proliferation and they participate in processes such as development, tissue regeneration, and cancer[1]. In dividing cells, entry into the S-phase depends on the activation of G1 Cyclins/Cyclin-Dependent Kinases (CDKs) 4 and 6 phosphorylate the retinoblastoma protein (RB), releasing the E2F transcription factors to drive cell cycle progression[2]. Dysregulation of this pathway is common in tumors, making CDK4/6 inhibitors (CDK4/6i) a valuable antitumor therapeutic approach[3,4]. However, clinical trials in triple-negative breast cancer (TNBC) revealed limited benefits[5].

CDK4 influences cellular processes beyond the cell cycle, including metabolism, autophagy, apoptosis and senescence, suggesting broader implications for CDK4/6i treatment[6–12].

[1]Center for Integrative Genomics, University of Lausanne, Faculty of Biology and Medicine, Lausanne, Switzerland. [2]Department of Cell Biology, Physiology and Immunology, Instituto Maimónides de Investigación Biomédica de Córdoba (IMIBIC)/University of Córdoba/Reina Sofía University Hospital, Córdoba, Spain. [3]Ludwig Institute for Cancer Research, University of Lausanne, Faculty of Biology and Medicine, Lausanne, Switzerland. [4]Institute of Sport Sciences and Department of Biomedical Sciences, University of Lausanne, Faculty of Biology and Medicine, Lausanne, Switzerland. [5]CIBER Fisiopatología de la Obesidad y Nutrición (CIBERobn), Instituto de Salud Carlos III, Madrid, Spain. [6]Laboratoire CarMeN, UMR INSERM U1060/INRA U1397, Université Claude Bernard Lyon1, F-69310 Pierre-Bénite, France. [7]Metabolomics Platform, University of Lausanne, Faculty of Biology and Medicine, Rue du Bugnon 19, 1005 Lausanne, Switzerland. [8]ISREC-Swiss Institute for Experimental Cancer Research, School of Life Sciences, Ecole Polytechnique Fédérale de Lausanne (EPFL), Lausanne, Switzerland. [9]The Breast Cancer Now Toby Robins Breast Cancer Research Centre, The Institute of Cancer Research, London, UK. [10]Inserm, Occitanie Méditerranée, Montpellier, France. ✉e-mail: lluis.fajas@unil.ch

Otto Warburg suggested that mitochondria, which is the major energy factory of the cells, was not essential for cancer progression[13]. Mitochondrial activity in cancer is also important, however, for the regulation of intrinsic cell death. Regulating multiple pro-survival pathways, including anabolic biosynthesis, calcium signaling and redox balance, mitochondrial reprogramming is now considered a hallmark of tumor cells[14]. These pathways are indeed regulated by the mitochondria-endoplasmic reticulum contact sites (MERCs)[15–19]. However, the upstream control of these MERCs in cancer-specific contexts is poorly understood. Specifically, the role of cell cycle regulators in the regulation of MERCs is completely unknown.

This study examines how CDK4 affects MERCs and its implications for TNBC cell metabolism, apoptosis, and resistance to therapy.

## Results

### CDK4 is dispensable for tumor growth in vitro and in vivo

To discriminate between cell cycle-mediated and cell cycle-independent effects of CDK4, we used a CRISPR CDK4 knockout (KO) model in MDA-MB-231 TNBC cells (Fig. 1a)[10]. CDK4 depletion resulted in a sustained RB phosphorylation at serine 780 (S780) and serine 807/811 (S807/811) (Fig. 1a, b) (Sup. Fig. 1a, b), independent of the stabilization of CDK6 or CDK2 (Fig. 1a) but associated with slightly increased level of CDK4/6-associated Cyclin D3, but not Cyclin D1 (Sup. Fig. 1a–c). The CDK4/6 inhibitor abemaciclib similarly reduced phosphorylation of S780 and S807/811 in both WT and CDK4-KO TNBC cells, while a CDK2 inhibitor only mildly hindered their phosphorylation indicating a compensatory effect of CDK6 rather than CDK2 (Sup. Fig. 1a, b). Moreover, CDK4-KO TNBC cells only slightly reduced their proliferation (Fig. 1c) and the expression of cell cycle-related genes (Sup. Fig. 1d). Interestingly, short-term treatment of WT cells with abemaciclib reduced RB S780 phosphorylation, whereas long-term chemical inhibition did not (Fig. 1e, f), indicating that short-term CDK4/6 inhibition reduced the early proliferation of TNBC cells but long-term exposure resulted in the generation of resistant cells (Fig. 1g–h). The effects of long-term CDK4/6 inhibition were, therefore, reminiscent of those in CDK4-KO TNBC cells.

We next explored the capacity of CDK4-KO MDA-MB-231 cells to form tumors in mice upon orthotopic implantation into the mammary fat pads of immunocompromised mice. Strikingly, CDK4-KO cells were able to form tumors, with the same penetrance as CDK4-WT cells, but with a latency of 30 days (Fig. 1i) (Sup. Fig. 1e–g). After this period, however, a similar tumor growth rate (Fig. 1j) and an equivalent tumor volume at sacrifice were observed between the CDK4-KO and WT groups (Sup. Fig. 1e, f). CDK4-KO tumors did not show changes in RB S780 phosphorylation (Fig. 1k, l) or in the expression of the pro-liferation marker Ki67 (Sup. Fig. 1h).

Collectively, these results suggest that CDK4 is dispensable for TNBC growth, allowing further exploration of its non-canonical functions.

### CDK4 facilitates genotoxic stress-induced cell death in TNBC cells

The treatment with CDK4/6i alone or in combination with chemotherapy is not efficient in patients with TNBC due to significant therapeutic resistance[5]. CDK4-KO cells exhibited resistance to chemotherapy (cisplatin, doxorubicin, 5-FU) and oxidative stress (H$_2$O$_2$, O + A), with increased cell viability compared to WT cells (Fig. 2a, b). Similarly, we observed significant resistance of CDK4-KO cells and WT cells pretreated with CDK4/6i, to UV$_B$ or TRAIL cell death inducers, and to oxidative- (H$_2$O$_2$) and mitochondrial stress (oligomycin and antimycin A) (O + A) (Sup. Fig. 2a, b). Resistance of cell death to H$_2$O$_2$ or O + A was validated in independent CDK4-KO clones (Sup. Fig. 2c, d). Long-term abemaciclib treated WT cells were also resistant to the chemotherapeutic drugs (Fig. 2c, d), and to a lesser extent to H$_2$O$_2$ or O + A (Sup. Fig. 2a). Notably, we observed that CDK6 inhibition with abemaciclib had a minor impact on cell death resistance in CDK4-KO cells (Fig. 2e and Sup. Fig. 2e), highlighting that the cell death-induced resistance by the CDK4/6i is predominantly mediated by CDK4 and not CDK6.

Resistance was validated in patient-derived xenografts and other TNBC lines (HCC1806, BT-474) (Fig. 2f, g and Sup. Fig. 2f–h). Interestingly, the RB-deficient TNBC cell line MDA-MB-468 showed no synergistic effect of the CDK4/6 inhibition with chemotherapeutic treatments whereas the (non-TNBC) ER + PR + BC cell line, MCF7, did (Sup. Fig. 2h, i).

In vivo, both WT and CDK4-KO cells successfully formed tumors in engrafted mice, though with a latency of four weeks (Fig. 2h). Mice were treated with either low or high doses of cisplatin (4 mg/kg and 8 mg/kg, respectively) (Fig. 2h and Sup. Fig. 2j, k) when tumor volumes reached 80-120 mm³. Both low and high doses of cisplatin inhibited the growth of WT xenograft tumors (Fig. 2i, j and Sup. Fig. 2l–o) whereas CDK4-KO xenograft tumors were resistant to the treatment (Fig. 2i, j and Sup. Fig. 2l–o).

### CDK4 regulates cell death via mitochondrial effectors of apoptosis in TNBC

Reintroduction of CDK4 into the KO cells restored the cell death phenotype in response to cisplatin, H$_2$O$_2$ and O + A (Sup. Fig. 3a–c), proving that the observed effects were mediated by CDK4 and not CDK6. We excluded next the possibility that resistance to cell death in CDK4-KO cells was secondary to an increase in DNA damage by showing that the frequency of γH2AX (an indicator of double-stranded DNA breaks) was equivalent in both WT and CDK4-KO cells one day after treatment with either cisplatin, H$_2$O$_2$ or O + A (Fig. 3a, b).

To better understand the mechanisms by which CDK4 modulates cell death, we investigated the mitochondrial apoptotic pathways. We found that the expression of the anti-apoptotic proteins BCL-xL and BCL-2 were increased in CDK4-KO cells, but not in the anti-apoptotic phosphorylation of BAD S113 (Fig. 3c, d). CDK4-KO cells also displayed enhanced expression of the pro-apoptotic BAD, but not in BAX or cytochrome C expression (Fig. 3c, d). These data suggested that the mitochondrial apoptotic pathway is altered and underlie, at least partially, resistance to cell death in CDK4-KO cells. Accordingly, the increased viability of both CDK4-KO cells and CDK4/6-inhibited cells was associated with failure to induce cleavage of caspase 3, a marker of apoptosis (Fig. 3e and Sup. Fig. 3g–k). Moreover, the induction of caspase 3 cleavage observed in CDK4-WT xenograft tumors upon cisplatin treatment was also blunted in the CDK4-KO xenograft tumors (Sup. Fig. 3i, j). Inhibition of caspases with zVAD, concomitantly with cisplatin, further demonstrated that that resistance to cell death in CDK4-KO cells is due to blockade of caspase-3 cleavage (Fig. 3f). Indeed, this treatment increased the viability of WT cells but not of CDK4-KO cells. However, zVAD treatment only partially rescued H$_2$O$_2$- and O + A-induced cell death (Fig. 3f), suggesting that additional cell death mechanisms were involved.

Mitochondrial calcium uptake is a determinant of the early steps of apoptosis induction[20]. Indeed, a deficiency in mitochondrial calcium uptake upon proapoptotic stress may allow cells to evade apoptosis[21,22]. In CDK4-KO cells, mitochondrial calcium uptake during apoptotic stress was markedly reduced compared to WT cells. This defect was consistent across various stress conditions, including H$_2$O$_2$ and O + A treatments. As a result, the threshold for initiating mitochondrial apoptosis was significantly higher in CDK4-KO cells (Fig. 3g–j and Sup. Fig. 3k, l).

Furthermore, mitochondrial membrane permeabilization (MOMP) represents a critical, irreversible step in apoptosis, characterized by the release of proteins from the intermembrane space into the cytosol[23]. Minority MOMP, can lead to failed apoptosis in various cancer cells[24]. To investigate whether CDK4 influences MOMP efficiency, we assessed mitochondrial membrane potential using

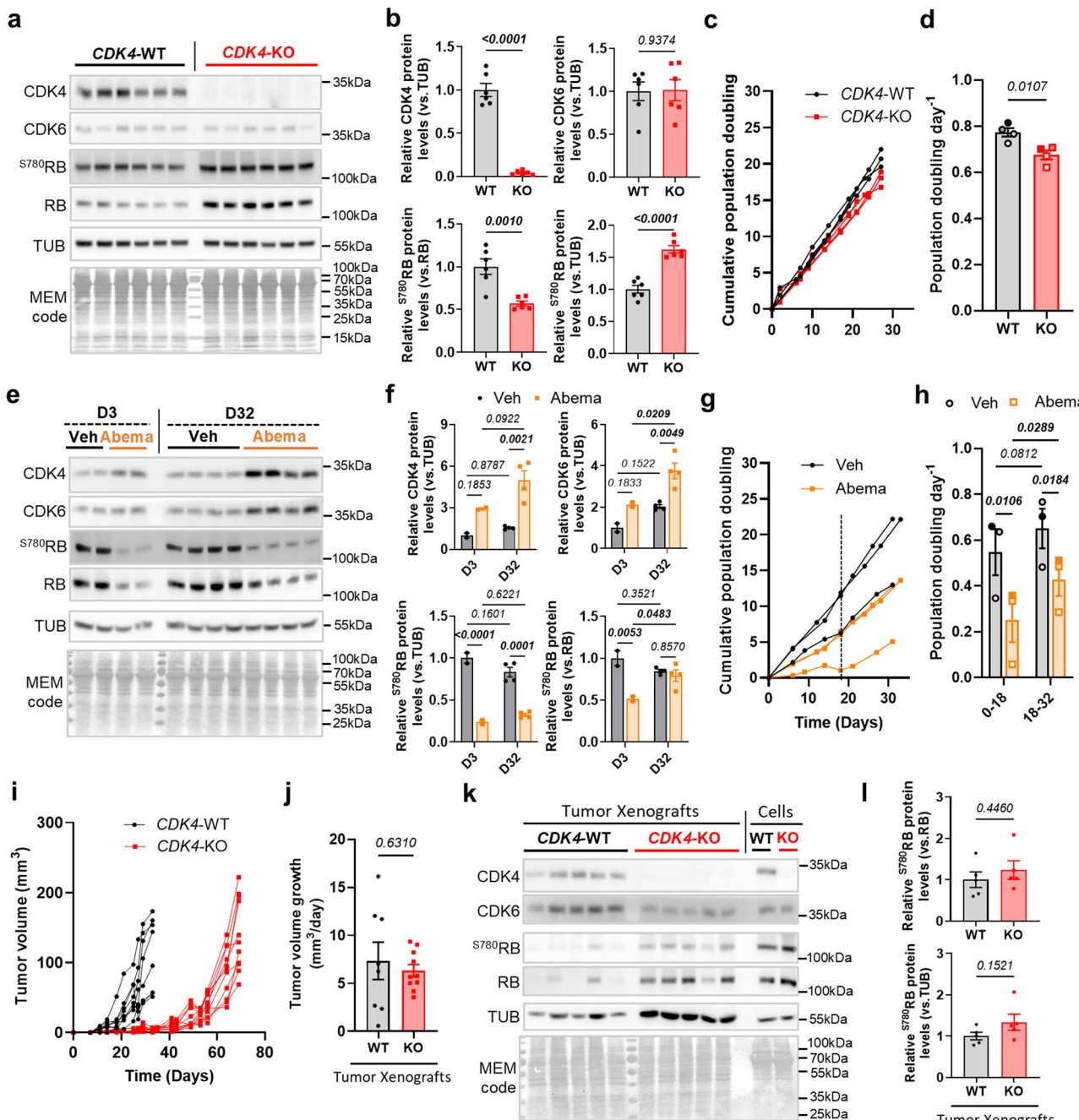

**Fig. 1 | CDK4 is dispensable for TNBC tumor growth in vitro and in vivo. a, b** Immunoblots and relative protein levels of CDK4, CDK6, $^{S780}$RB, RB, Tubulin (TUB) and MEM code of CDK4-WT and -KO MDA-MB-231 TNBC cells. Mean +/- SEM of N = 6 independent biological replicates. Two-sided Unpaired T-tests. **c, d** Growth curves and proliferation rates of CDK4-WT and -KO TNBC cells. Mean +/- SEM of N = 4 independent biological replicates. Two-sided Unpaired T-test. **e, f** Immunoblots and relative protein levels of CDK4, CDK6, $^{S780}$RB, RB, Tubulin (TUB) and MEM code of TNBC cells, upon treatment with Vehicle (Veh) or Abemaciclib (Abema) for 3 days (D3) or 32 days (D32). Mean +/- SEM of N = 2 (D3) and N = 4 (D32) independent biological replicates. 2-way ANOVA; Tukey's multiple comparison tests. **g,**

**h** Growth curves and proliferation rates of TNBC cells, upon treatment with Veh or Abema. Mean +/- SEM of N = 3 independent biological replicates. 2-way ANOVA; Tukey's multiple comparison tests. **i** Tumor volume of CDK4-WT and -KO tumor xenografts. One curve represents one xenograft. N = 8 WT and N = 10 KO. **j** Tumor volume growth after tumors reached 20 mm³ of volume of CDK4-WT and -KO xenografts. Mean +/- SEM of N = 8 WT and N = 10 KO. Two-sided Welch's T-test. **k, l** Immunoblots and relative protein levels of CDK4, CDK6, $^{S780}$RB, RB, Tubulin (TUB) and MEM code of CDK4-WT and -KO tumor xenografts, and CDK4-WT and -KO TNBC cells. Mean +/- SEM of N = 5 WT and KO tumors. Two-sided Unpaired T-tests. Exact *p-values* are displayed in italic (bold italic if <0.05).

TMRM staining in both CDK4-WT and CDK4-KO cells following O + A treatment (Fig. 3k, l). CDK4-WT cells exhibited a significant drop in mitochondrial membrane potential after treatment, whereas this decline was notably limited in CDK4-KO cells (Fig. 3k, l).

Collectively, these findings highlight the role of CDK4 in promoting genotoxic-induced cell death in TNBC cells, primarily via apoptosis but also other cell death mechanisms. Moreover, the resistance to death mechanism in CDK4-KO cells involves mitochondrial effectors of apoptosis including calcium uptake and MOMP, rather than a DNA damage response.

## CDK4 participates in mitochondrial fission

Changes in mitochondrial morphology are often correlated with its function. Using transmission electron microscopy (TEM), we

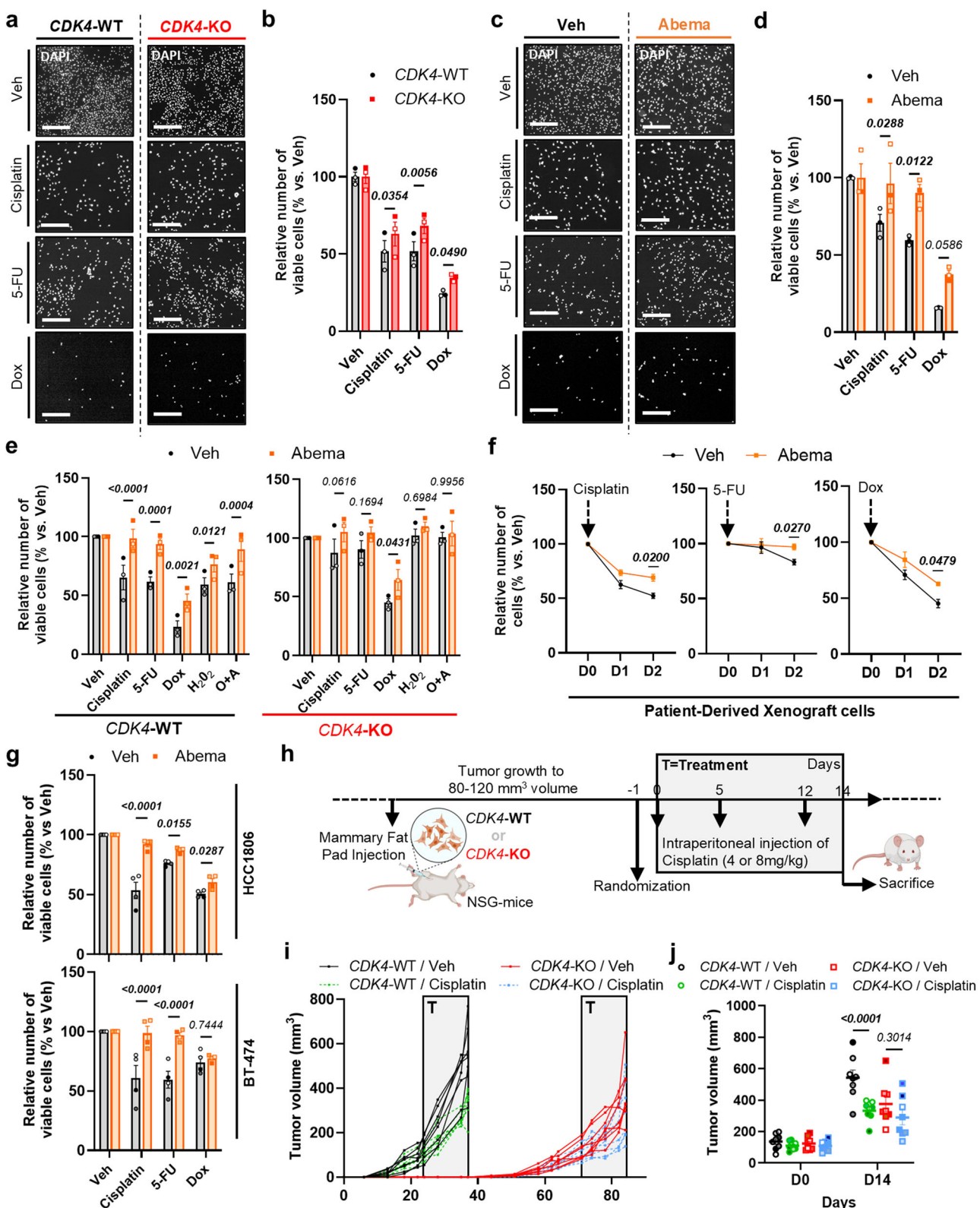

revealed that CDK4-KO cells exhibit enlarged mitochondria (Fig. 4a, b), the appearance of giant mitochondria (Fig. 4c and Sup. Fig. 4a) and enhanced mitochondrial DNA quantity (Sup. Fig. 4b). To a lesser extent, increased mitochondria area was also observed in CDK4-KO xenograft tumors (Sup. Fig. 4c). Interestingly, the total mitochondrial number was not affected by the loss of CDK4 (Sup. Fig. 4d). By gene set enrichment analysis (GSEA) of the RNA

sequencing (RNA-seq) data, we found that Peroxisome Proliferator-Activated Receptor Gamma Coactivator 1-Alpha (PGC1-α) and Estrogen-Related Receptor Alpha (ERRα)-target genes were not enriched in CDK4-KO cells (Sup. Fig. 4e), suggesting that these giant mitochondria were not the consequence of increased mitochondrial biogenesis. Consistent with this hypothesis, the mRNA levels of genes involved in mitochondrial biogenesis, namely, *POLMRT, TBF1M*

**Fig. 2 | CDK4 inhibition confers to TNBC cells resistance to chemotherapy in vitro and in vivo. a, b** Representative pictures of DAPI staining and quantification of the number of viable CDK4-WT and -KO MDA-MB-231 TNBC cells, upon treatment with Vehicle, Cisplatin (10 μM), 5-FU (20 μM), or Doxorubicin (5 μM) (Dox). Scale bars: 400 μm. Mean +/- SEM of N = 3 independent biological replicates. 2-way ANOVA; Sidák's multiple comparison tests. **c, d** Representative pictures of DAPI staining and quantification of the number of viable pretreated cells with Vehicle or Abemaciclib (Abema) for 2 days and after treatment with Vehicle, Cisplatin (10 μM), 5-FU (20 μM), or Dox (5 μM). Scale bars: 400 μm. Mean +/- SEM of N = 3 independent biological replicates. 2-way ANOVA; Sidák's multiple comparison tests. **e** Number of viable CDK4-WT and -KO TNBC cells, pretreated with Vehicle or Abema for 2 days and after 2 days of treatment with Vehicle, Cisplatin (10 μM), 5-FU (20 μM), Dox (5 μM), $H_2O_2$ (250μM-2hours) or Oligomycin and Antimycin (O + A) (1 + 10μM-2hours). Mean +/- SEM of N = 3 independent biological replicates. 2-way ANOVA; Sidák's multiple comparison tests. **f** Relative number of triple-negative NST breast cancer patient-derived cells pretreated cells with Vehicle or Abema for 2 days. Cisplatin (12 μM), 5-FU (40 μM) or Dox (10 μM) treatments were performed at D0. Mean +/- SEM of N = 4 independent biological replicates. 2-way ANOVA; Sidák's multiple comparison tests. **g** Number of viable HCC1806 and BT-474 TNBC cells, pretreated with Vehicle or Abema for 2 days and after treatment with Vehicle, Cisplatin (20 μM), 5-FU (40 μM), or Dox (5 μM). Mean +/- SEM of N = 4 independent biological replicates. 2-way ANOVA; Sidák's multiple comparison tests. **h** CDK4-WT and -KO MDA-MB-231 TNBC cells were injected in fat pad of the mammary gland of female NSG mice. Randomization was performed when tumor volume reached 80-120mm³ and Cisplatin IP (4-8 m/kg) was performed at D0, D5 and D12, before sacrifice at D14. **i** Tumor volume of CDK4-WT and -KO tumor xenografts, treated with vehicle (Veh) or Cisplatin (4 mg/kg). T=Treatment. N = 8 mice per group. **j** Tumor volume of CDK4-WT and -KO tumor xenografts at the end of the sacrifice. 2-way ANOVA; Sidák's multiple comparison tests. Mean +/- SEM of N = 8 mice. Exact *p-values* are displayed in italic (bold italic if <0.05).

and *TBF2M* (Sup. Fig. 4f) were similar between WT and CDK4-KO TNBC cells. Using fluorescence tracking and electron microscopy, we observed that mitochondria in CDK4-KO cells had higher aspect ratio and form factor (Fig. 4d and Sup. Fig. 4g, h), suggesting increased elongated mitochondria. Furthermore, dynamic cell imaging showed a persistent fragmented mitochondrial network in WT cells and a permanent hyperfused mitochondrial network in CDK4-KO cells (Fig. 4e, Sup. Movies 1 and 2). Mitochondrial dynamics are governed by the fission/fusion equilibrium, which involves coordination between fission proteins (MFF, MIEF and DRP1) and fusion proteins (MFN1, OMA1 and OPA1)[25]. The expression of mitochondrial fusion genes were not changed in the CDK4-KO cells (Sup. Fig. 3i), but the level of S616-phosphorylated DRP1, a pro-fission protein, was decreased in CDK4-KO cells (Fig. 4f, g). This was also observed in TNBC cells treated with CDK4/6i (Sup. Fig. 4j, k), which also showed fused mitochondrial network (Sup. Fig. 4l). Finally, ectopic expression of CDK4 in the KO cells normalized the mitochondrial aspect ratio and form factor and restored a normal mitochondrial shape (Fig. 4h).

## CDK4 modulates endoplasmic reticulum to-mitochondria (ER-MT) calcium signaling

To examine CDK4's role in mitochondrial calcium uptake during apoptosis, we investigated calcium signaling in CDK4-KO cells, focusing on pro-apoptotic ER-to-mitochondria calcium flux[20]. Using the ratiometric calcium probe 4mtD3cpv[26] CDK4-KO cells displayed elevated baseline mitochondrial calcium levels (Fig. 5a). Key ER-MT calcium channels, including ITPR1, ITPR3, and VDAC1, were upregulated, while ITPR2 was downregulated, with no changes observed in SERCA1 or MCU levels (Fig. 5b, c).

To evaluate ER-MT calcium transfer functionality, thapsigargin (TG) and histamine (Hist) were used to induce ER calcium release. CDK4-WT cells showed robust mitochondrial calcium accumulation, whereas CDK4-KO cells exhibited reduced peak amplitude and area under the curve (Fig. 5d–g, Sup. Fig. 5a, b). Reduced immediate ER-to-MT calcium flux was also confirmed using the ratiometric probe (Sup. Fig. 5c, d). However, secondary calcium transfer remained unaffected, suggesting compensatory effects from elevated ITPR1, ITPR3, and VDAC1 (Sup. Fig. 5e).

Cytosolic calcium release kinetics were comparable between CDK4-WT and CDK4-KO cells, indicating that mitochondrial deficits were not due to impaired ER calcium release (Fig. 5h–k, Sup. Fig. 5f, g). Reintroducing CDK4 into KO cells restored mitochondrial calcium uptake in response to TG (Fig. 5l, m, Sup. Fig. 5h).

Finally, knockdown of VDAC1 with ITPR3 or MCU partially rescued 13–16% of the cell death induced by $H_2O_2$ and O + A in CDK4-WT cells but had no effect on cisplatin-induced cell death (Sup. Fig. 5i–k). These results indicate that CDK4 inactivation impairs ER-MT calcium signaling, contributing to apoptosis resistance in TNBC cells.

## CDK4 participates in the establishment of MERCs

ER-MT calcium transfer involves membrane contact sites between the ER membrane and the outer mitochondrial membrane (OMM), known as MERCs[15]. These sites are essential for mitochondrial fission, where the ER physically wraps mitochondria[27,28]. Since we observed both reduced ER-MT calcium signaling and decreased mitochondrial fission in CDK4-KO cells (Figs. 4, 5), we next assessed whether the structure of MERCs is modified in CDK4-KO cells using complementary technical approaches, such as TEM, in situ proximity ligation assay (PLA) and immunofluorescence staining.

TEM analysis showed fewer MERCs per mitochondrion in CDK4-KO cells (Fig. 6a, b) and an overall decrease in MERC percentage (Fig. 6c). Additionally, both mean and minimum ER-mitochondria distances increased in CDK4-KO cells (Fig. 6d, Sup. Fig. 6a), indicating looser ER-mitochondria contacts. Similar results were found in xenograft tumors, with fewer MERCs per mitochondrion (Sup. Fig. 6b, c). Immunofluorescence further confirmed decreased colocalization of ER and mitochondrial markers (Sup. Fig. 6d, e). PLA analysis showed reduced close appositions (< 40 nm) between protein pairs VAPB/PTPIP51 and ITPR1/VDAC1 in CDK4-KO cells (Fig. 6e, f, Sup. Fig. 6f, g). Restoring CDK4 in KO cells re-established MERCs, increasing ITPR1-VDAC1 interactions (Fig. 6f).

To better understand how CDK4 may impact the composition of MERCs, we performed a large-scale proteomic analysis in both WT and CDK4-KO cells. Based on a meta-analysis of various proteomic profiles of MERC fractions isolated from human cells[29–33], we first established a list of 176 MERC-associated proteins (Sup. Table 1). In our data, we detected 139 of these 176 MERC proteins (79%) (Fig. 6g). Of these 139 detected proteins, 98 were differentially regulated upon CDK4 deletion (70%), and most of these were down-regulated (Fig. 6g, h). To more precisely delineate this mechanism of CDK4 regulation, the MERC proteins were clustered according to their known-associated functions and pleiotropic ones for specific proteins (e.g. CYCS, EIF2AK3, HSPA5, MFN2, VPS13D) (Sup. Table 2). The MERC tethering protein category showed the greatest down-regulation in CDK4-KO cells (Fig. 6h, i). In contrast, lipid metabolism and signaling-associated proteins were most upregulated in CDK4-KO cells (Fig. 6h, i), probably stemming from compensatory mechanisms activated to allow the cells to rely on a smaller number of MERCs but with a high abundance of lipid-associated and signaling proteins.

## PKA activity is regulated by CDK4 but is not sufficient to mediate apoptosis

Since CDK4 is a serine-threonine kinase, we hypothesized that it exerts its effects via the phosphorylation of other targets implicated in the

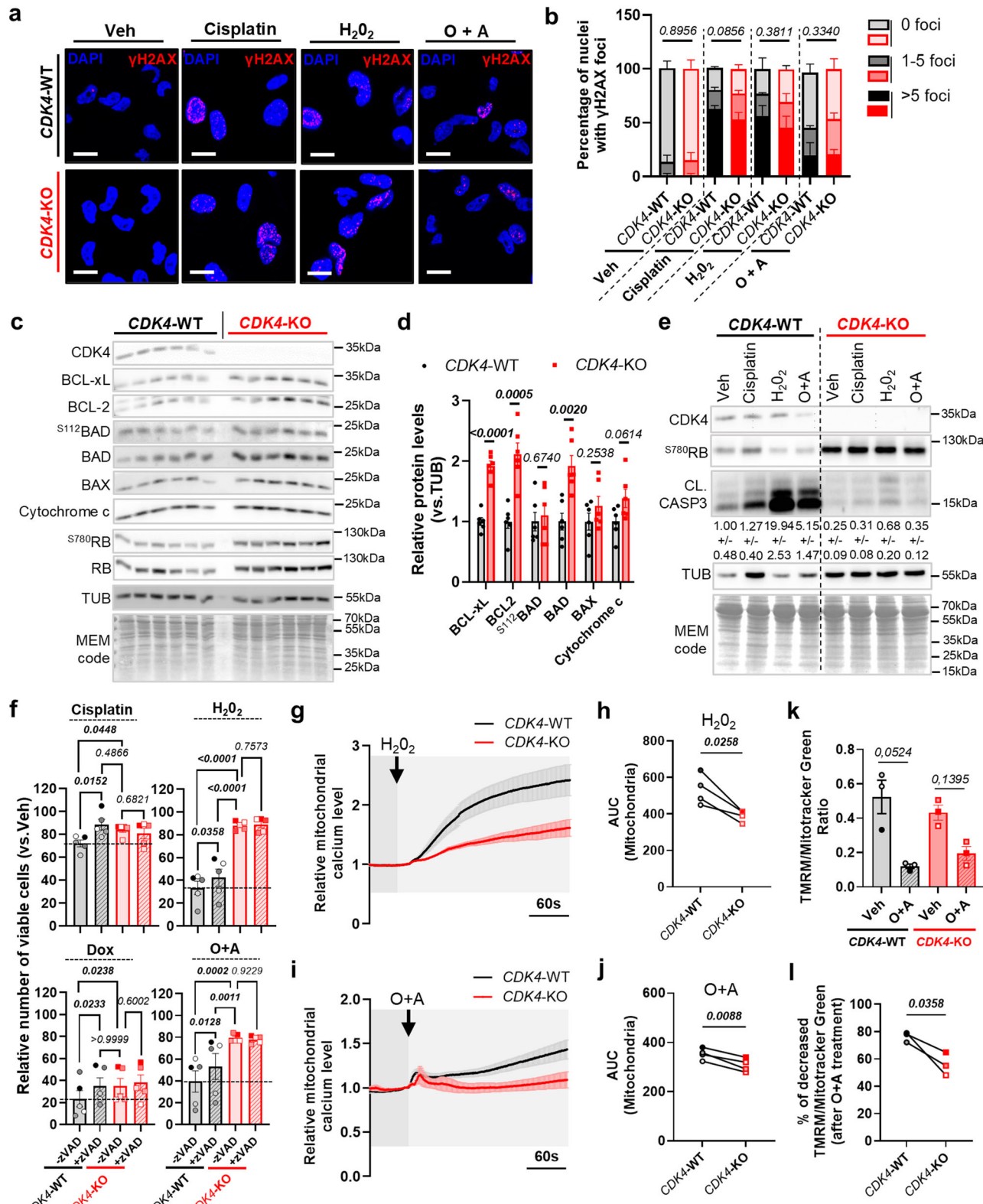

control of MERCs. We therefore performed PamGene phosphoproteomic analysis coupled to Integrative Inferred Kinase Activity (INKA) analysis[34]. The results of these two independent and complementary analyses converged to identify PKA alpha as the kinase with the most down-regulated kinase activity in CDK4-KO cells compared to CDK4-WT cells (Fig. 7a and Sup. Fig. 7a). Phospho-PKA substrates were consistently down-regulated in CDK4-KO cells (Fig. 7b). Reintroduction of CDK4 into CDK4-KO cells was sufficient to restore PKA activity

(Fig. 7c), highlighting a CDK4-mediated effect on this activity. We further assessed whether the kinase activity of CDK4 was needed for maintaining PKA activity. To this end, we used a kinase-dead CDK4 mutant (K35M) and a non-inhibitable CDK4 mutant (R24C; to rescue CDK4 activity in KO cells) and revealed that CDK4 K35M could not reverse the decreases in the expression of phospho-PKA substrates, in contrast to CDK4 R24C (Fig. 7c), proving that the kinase activity of CDK4 is required for PKA activity.

**Fig. 3 | CDK4 regulates cell death modulating mitochondrial effectors of apoptosis in TNBC. a** Representative pictures of DAPI (blue) and γH2AX (red) staining in CDK4-WT and -KO MDA-MB–231 TNBC cells 24 hours after treatment with Vehicle, Cisplatin (10 μM), $H_2O_2$ (250μM-2hours), or Oligomycin and Antimycin (O + A) (1 + 10μM-2hours). Scale bars: 10 μm. **b** Relative proportions of nuclei displaying 0, 1-5 or more than 5 γH2AX foci. Mean +/- SEM of N = 3 independent biological replicates. Two-sided Freeman-Halton extension of Fisher's exact test. **c, d** Immunoblots and relative protein levels of CDK4, BCL-XL, BCL-2, $^{S112}$BAD, BAD, BAX, Cytochrome C, $^{S780}$RB, RB, Tubulin (TUB) and MEM code of CDK4-WT and -KO TNBC cells. Multiple two-sided Unpaired T-tests. **e** Immunoblots of CDK4, $^{S780}$RB, Cleaved Caspase-3 (CL.CASP3), Tubulin (TUB) and MEM code of CDK4-WT and -KO TNBC cells, upon treatment with Vehicle, Cisplatin, $H_2O_2$ or O + A. Quantification of relative cleaved Caspase-3 protein levels (normalized to tubulin level). Mean +/- SEM of N = 3 independent biological replicates. **f** Number of viable CDK4-WT and -KO TNBC cells pretreated during 3 hours with or without zVAD (20 μM) and then exposed to Cisplatin, Doxorubicin, $H_2O_2$, or O + A. Mean +/- SEM of N = 5

independent biological replicates. 2-way ANOVA; Tukey's multiple comparison tests. **g-j** Relative mitochondrial calcium levels of CDK4-WT and -KO TNBC cells upon $H_2O_2$ (2.5 mM) or O + A (100 μM, 10 μM) injections. Curves based on Rhod-2AM probe fluorescence intensity. Arrows indicate the time of injection. Associated quantification of the area under the curves (AUC). Mean +/- SEM of N = 4 independent biological replicates representing a total of n = 135 cells for 10 independent injections (WT-$H_2O_2$), n = 123 cells for 9 independent injections (KO-$H_2O_2$), n = 128 cells for 9 independent injections (WT-O + A) and n = 128 cells for 9 independent injections (KO-O + A). Two-sided Paired-T-tests. **k** Quantification of the ratio between TMRM and Mitotracker Green fluorescence intensities in CDK4-WT and -KO TNBC cells 2 hours after vehicle (Veh) or O + A (1 + 10 μM) treatments. Mean +/- SEM of N = 3 independent biological replicates. 2-way ANOVA; Tukey's multiple comparison tests. **l** Percentage of decreased of TMRM/Mitotracker Green ratio after O + A treatment in CDK4-WT and -KO TNBC cells. N = 3 independent biological replicates. Two-sided Paired-T-test. Exact *p-values* are displayed in italic (bold italic if <0.05).

---

PKA is a tetrameric holoenzyme composed of two regulatory and two catalytic subunits and is inactive in this state[35]. The PKA enzymatic complex is organized by A-kinase anchoring proteins (AKAPs), which facilitate its interactions, including those of the regulatory subunits and its substrates. AKAPs are also important for the specific intracellular localization of PKA. Upon the binding of cAMP to the PKA regulatory subunits, the catalytic subunits are released from the complex, and PKA is activated. Among these components, phosphoproteomics analysis identified PRKAR1A and PRKAR1B as putative direct targets of CDK4, all presenting a SPXK/R/P phospho-site, a CDK4 motif, downregulated in CDK4-KO cells (Sup. Fig. 7b). Interestingly, PLA using exogenous forms of CDK4-Myc and PRKAR1A-HA or PRKAR1B-HA showed a preferential interaction of CDK4 with PRKAR1A (Sup. Fig. 7c). Importantly, while the reactivation of PKA activity was possible in CDK4-KO cells with Forskolin, as measured by the activation of the adenylyl cyclase activity and the intracellular cAMP formation (Fig. 7d), this was not sufficient to promote a restoration of cell death in CDK4-KO cells (Fig. 7e).

## CDK4 regulates PKA signaling at MERCs modulating ER-MT calcium signaling

While the cytosolic PKA (cPKA) is recognized for targeting and inhibiting pro-apoptotic proteins such as BAD, the function of mitochondrial PKA (mtPKA) in the regulation of pro- and anti-apoptotic proteins remains less well defined[36]. PKA can indeed be recruited to OMM by mitochondrial AKAP, including AKAP1[37]. We therefore sought to identify PKA substrates and elucidate the localization of PKA and CDK4 at MERCs. We performed proteomic and phosphoproteomic analyses in isolated crude/pure mitochondria, ER, and MERC fractions. The purity of the fractions was validated using specific markers for the ER, mitochondria and MERC fractions (Fig. 8a). CDK4 was notably detected in the cytosolic, mitochondrial, ER, and MERC fractions (Fig. 8a). From our analysis, 3659 proteins were enriched in the MERC fractions. Taking as reference the established list of 176 core MERC proteins from our previous proteomic data (Sup. Table 1), we detected 115 of these in the MERC fraction. Among these, 40 proteins (23%) were exclusive to the MERC fraction, and 54 proteins (31%) were enriched in the MERC fraction compared to the whole-cell lysate (WCL) (Sup. Fig. 8a).

Proteomic analysis revealed that the majority of these 115 MERC-associated proteins were upregulated in CDK4-KO cells (Fig. 8b), suggesting a compensatory response to the reduced number of MERCs in these cells. Intriguingly, the MERC fraction also contained at least two PKA regulatory subunits (PRKAR1A, PRKAR2A), two PKA catalytic subunits (PRKACA, PRKACB), and four AKAPs (AKAP1, AKAP2, AKAP5, and AKAP10) (Fig. 8b). We confirmed the localization of the putative CDK4 target PRKAR1A (Sup. Fig. 7b, c) (Fig. 8a) and verified the presence of the PRKACA catalytic subunit at MERCs using PLA against ITPR1-VDAC1 coupled with PRKACA immunostaining (Fig. 8c).

Crucially, phosphoproteomic enrichment analysis demonstrated that phosphopeptides with PKA and CDK4 motifs were significantly deregulated in the MERCs of CDK4-KO cells (Fig. 8d and Sup. Fig. 8b, c). Consistent with this, most PKA substrates were downregulated in the MERC fractions of CDK4-KO cells (Fig. 8e). Specifically, the phosphorylation of ITPR1 at S1756, a known PKA target site critical for potentiating ITPR-mediated calcium release[38], was markedly decreased in CDK4-KO cells (Fig. 8f, g).

Beyond PKA-targeted phosphosites, we identified 59 differentially regulated phosphosites across 49 MERC-associated proteins (Sup. Fig. 8b). Among these, several calcium-related channels involved in ER-MT calcium transfer, including ITPR1, ITPR2, VDAC1, and VDAC2, were down-regulated in CDK4-KO cells (Sup. Fig. 8b).

To assess the role of PKA signaling at MERCs in ER-MT calcium signaling, we pretreated WT and CDK4-KO cells with Forskolin. Remarkably, PKA activation enhanced mitochondrial calcium uptake upon thapsigargin injection exclusively in WT cells, with no effect observed in CDK4-KO cells (Fig. 8h, i). These findings suggest that MERCs integrity is essential for mediating PKA signaling at these contact sites. Furthermore, they indicate that CDK4 is not only involved in stabilizing MERCs but also regulates PKA activity at the ER-MT interface to enhance calcium signaling.

In line with the absence of cell death in CDK4-KO cells treated with Forskolin (Fig. 7d, e), these results underscore the importance of both MERCs structure and functional PKA signaling at MERCs in inducing cell death in TNBC.

## CDK4 activity is positively correlated with the apoptosis signature and a better response to neoadjuvant chemotherapy (NAC) in TNBC human patients

The clinical relevance of our model was evaluated using two large patients cohort datasets, SCAN-B and TCGA bulk RNA-sequencing[39,40]. High CDK4 expression in TNBC patients was associated with poor prognosis, although CDK4 expression did not correlate with the apoptosis signature in either dataset (Fig. 9a and Sup. Table 3). It is important to note that CDK4 activity is regulated by the expression of D-type Cyclins (CCNDs). Therefore, we assessed the prognosis of TNBC patients based on the independent or combined expression of Cyclins D1, D2, and D3 (Signature CCNDs) (Fig. 9b). Interestingly, while no significant survival differences were observed based on Cyclin D1 or D3 expression, low Cyclin D2 expression—and to a lesser extent, a low CCNDs signature—was associated with poor prognosis (Fig. 9a and Sup. Table 3), suggesting that reduced CDK4 activity may be detrimental for TNBC patients. Additionally, Cyclin D2 and the CCNDs signature were both highly correlated with the apoptosis signature in the same datasets (Fig. 9b).

To further explore the impact of CDK4 activity on the apoptosis response during chemotherapy, we analyzed another dataset with

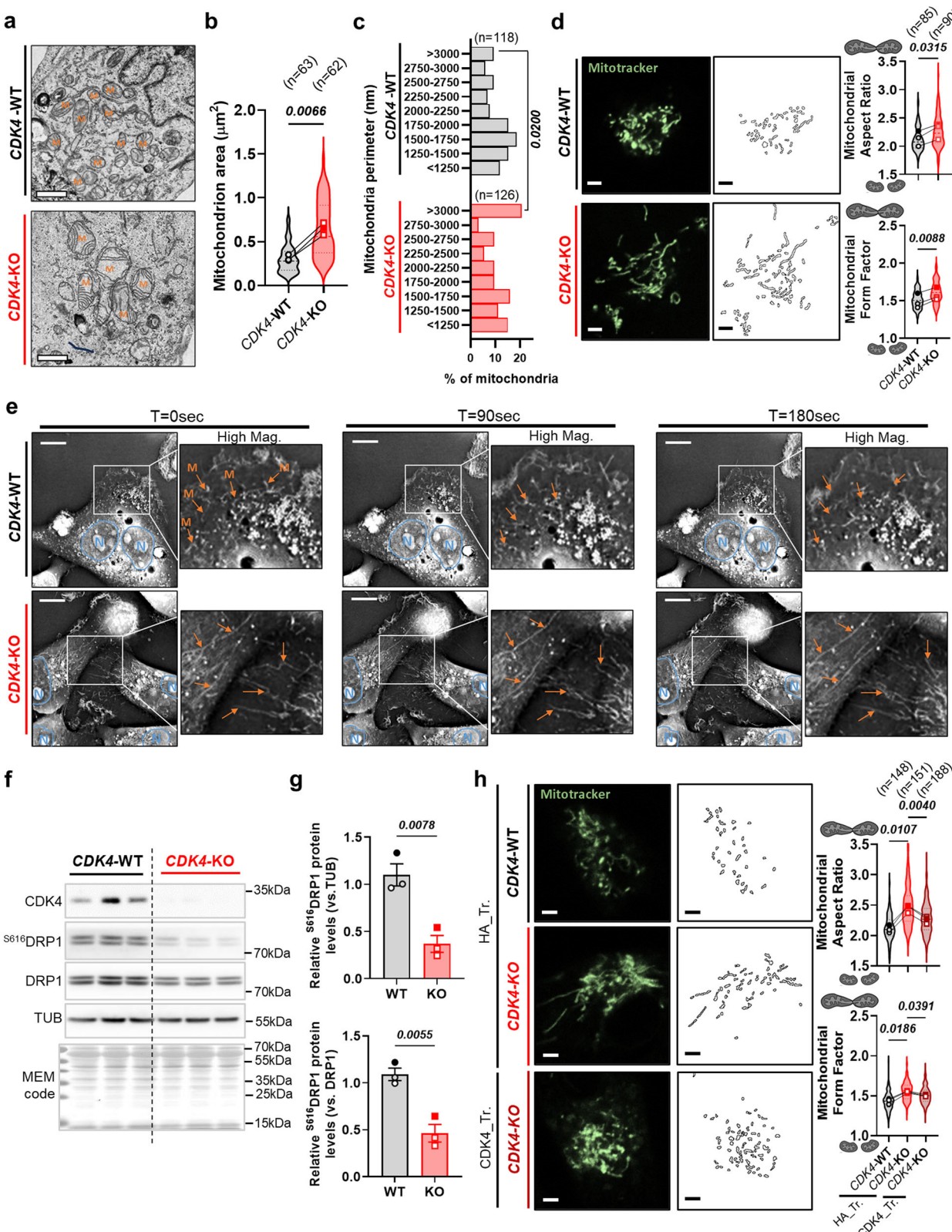

RNA-sequencing data collected at baseline and three weeks after neoadjuvant chemotherapy (NAC)[41] (Fig. 9c). The apoptosis signature was enhanced after NAC treatment, particularly in patients who achieved a pathological complete response (pCR), compared to non-responders (non-pCR). Notably, NAC induced a greater expression of the CCNDs signature in the tumors of pCR patients than in non-pCR patients (Fig. 9c). We confirmed that the CCND signature was

associated with the apoptosis signature in this third cohort, independently of the NAC response (Fig. 9d).

We also examined the relative impact of MERC components expression, based on the list in Sup. Table 1, as well as two PKA signatures, specifically PKA Phosph. CREB and BIOCARTA CREB PATHWAY. No significant differences in overall or recurrence-free survival were found in TNBC patients from the SCAN-B and TCGA cohorts

**Fig. 4 | CDK4 participates in mitochondrial fission of TNBC. a** Representative electron micrographs of CDK4-WT and -KO MDA-MB-231 TNBC cells. Scale bars: 1 µm. M=Mitochondria. **b** Quantification of mitochondria area of CDK4-WT and -KO TNBC cells. Means of N = 3 independent biological replicates from n=mitochondria represented with violon plot. Two-sided Paired T-test. **c** Distribution histogram of mitochondria perimeter of CDK4-WT and -KO TNBC cells. n=mitochondria number representative from N = 3 independent biological replicates. Two-sided Fisher's exact test. **d** Representative pictures of mitochondria staining using Mitotracker of CDK4-WT and -KO TNBC cells. Scale bars: 4 µm. Associated quantification of mitochondrial aspect ratio (major axis/minor axis) and form factor (1/circularity) in CDK4-WT and -KO TNBC cells. Means of N = 3 independent biological replicates from n=cells represented with violin plot. **e** Time-live tracking of mitochondria of

CDK4-WT and -KO TNBC cells at T = 0 sec, T = 90 sec and T = 180 sec. N delineation indicates nuclei (Blue) and M arrows indicate mitochondria (Orange). Scale bars: 4 µm. Representative of N = 3 independent biological replicates. **f, g** Immunoblots and relative protein levels of CDK4, $^{S616}$DRP1, DRP1, Tubulin (TUB) and MEM code of CDK4-WT and -KO TNBC cells. Mean +/- SEM of N = 3 independent biological replicates. Two-sided Unpaired T-tests. **h** Representative pictures of mitochondria staining using Mitotracker Green of CDK4-WT, CDK4-KO TNBC cells transfected with empty plasmid HA (HA_Tr.) and CDK4-KO TNBC cells expressing endogenous CDK4 (CDK4_Tr.). Scale bars: 4 µm. Associated quantification of mitochondrial aspect ratio and form factor. Means of N = 3 independent biological replicates from n = cells represented with violin plot. RM 1 way ANOVA; Tukey's multiple comparisons test. Exact *p-values* are displayed in italic (bold italic if <0.05).

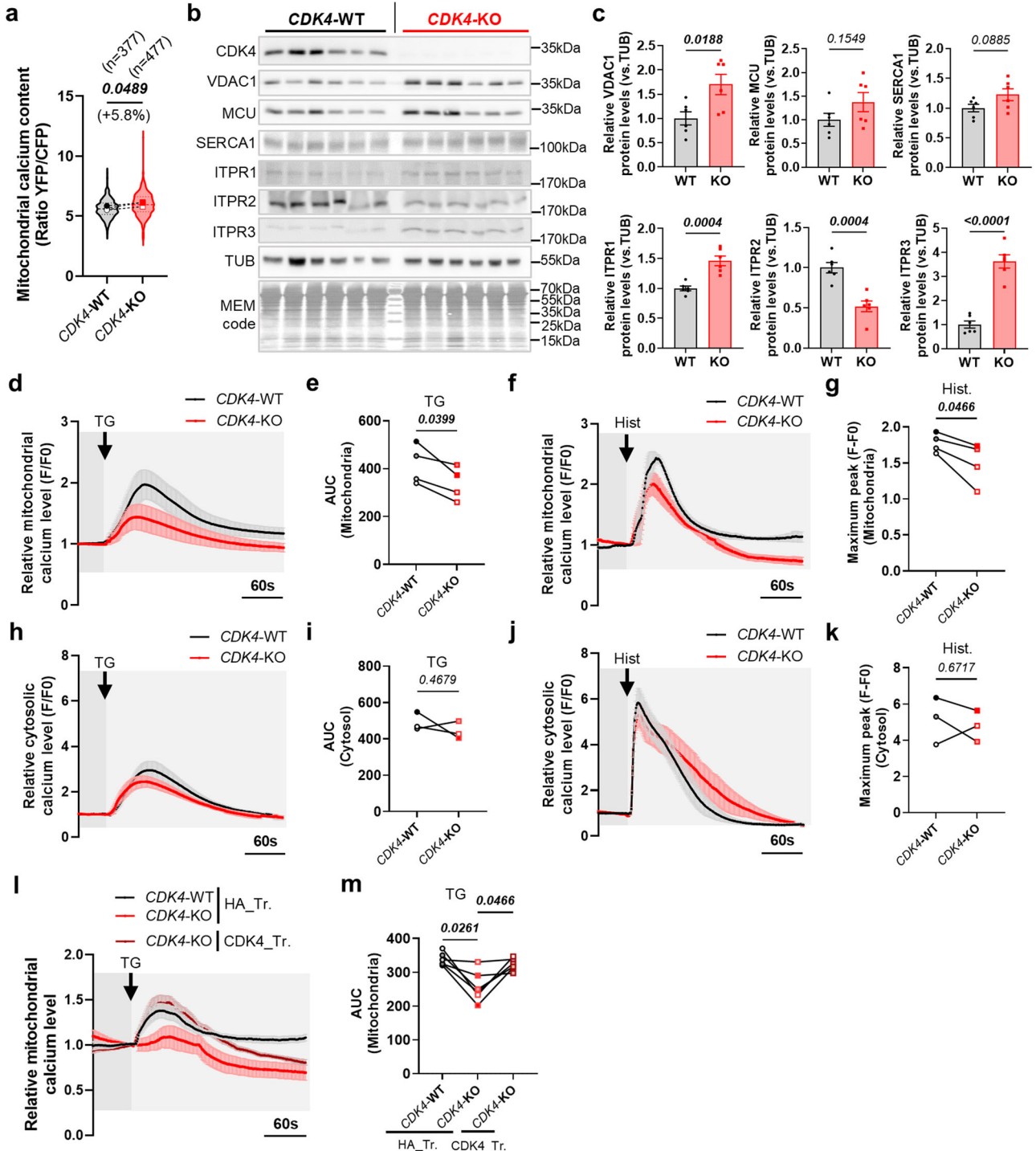

**Fig. 5 | CDK4 regulates ER-mitochondrial calcium signaling. a** Basal mitochondrial calcium content of CDK4-WT and -KO MDA-MB-231 TNBC cells. Medians of N = 3 independent biological replicates from n = 377 (CDK4-WT) and n = 477 (CDK4-KO) cells represented with violin plot. Two-sided Paired t-test. **b, c** Immunoblots and relative protein levels of CDK4, SERCA1, ITPR1, ITPR2, ITPR3, VDAC1, MCU, Tubulin (TUB) and MEM code of CDK4-WT and -KO TNBC cells. Parallel gels were loaded for TUB, with subsequent same antibodies/exposure conditions. Mean +/- SEM of N = 6 independent biological replicates. Two-sided Unpaired T-tests. **d–g** Relative mitochondrial calcium levels of CDK4-WT and CDK4-KO TNBC cells upon Thapsigargin (TG) (2 μM) or Histamine (Hist) (50 μM) injections. Curves based on Rhod-2AM fluorescence intensity. Arrows indicate the time of injection. Associated quantification of the area under the curves (AUC) or maximum amplitude/peak. Mean +/- SEM of N = 4 independent biological replicates from n = 148 cells for 13 independent injections (WT-TG), n = 142 cells for 13 independent injections (KO-TG), n = 133 cells for 10 independent injections (WT-Hist.) and n = 141 cells for 12 independent injections (KO-Hist.). Two-sided Paired T-tests.

**h–k** Relative cytosolic calcium levels of CDK4-WT and CDK4-KO TNBC cells upon Thapsigargin (TG) (2 μM) or Histamine (Hist) (50 μM) injections. Curves based on Fluo-4 fluorescence intensity. Arrows indicate the time of injection. Associated quantification of the area under the curves (AUC) or maximum amplitude/peak. Mean +/- SEM of N = 3 independent biological replicates from n = 107 cells for 9 independent injections (WT-TG), n = 123 cells for 9 independent injections (KO-TG), n = 121 cells for 9 independent injections (WT-Hist.) and n = 118 cells for 9 independent injections (KO-Hist.). Two-sided Paired T-tests. **l, m** Relative mitochondrial calcium levels of CDK4-WT, CDK4-KO TNBC cells transfected with empty plasmid HA (HA_Tr.) and CDK4-KO TNBC cells expressing endogenous CDK4 (CDK4_Tr.) upon Thapsigargin (TG) (2 μM) injection. Arrows indicate the time of injection. Associated quantification of the area under the curves (AUC). Mean +/- SEM of N = 6 independent injections from 2 biological replicates and n = 75 cells (WT/HA_Tr.), n = 55 cells (KO/HA_Tr.) and n = 75 cells (KO/CDK4_Tr.). RM 1 way ANOVA; Tukey's multiple comparisons test. Exact *p-values* are displayed in italic (bold italic if <0.05).

based on MERC or PKA signatures (Fig. 9a and Sup. Table 3). However, NAC treatment induced a stronger MERC signature in pCR patients compared to non-pCR patients (Fig. 9e). Additionally, relative PKA activity was assessed in parallel after NAC treatment using the two PKA signatures (Fig. 9f, g). While both signatures showed enrichment after NAC treatment in both pCR and non-pCR patients, the PKA Phosph. CREB signature was only significantly upregulated in pCR patients (Fig. 9g).

Collectively, these data provide clinical evidence of associations between Cyclins D expression, MERC expression, and PKA signatures, all of which are related to a better response to NAC in TNBC patients.

### CDK4 promotes mitochondrial fitness and metabolic flexibility in TNBC cells

MERCs regulate mitochondrial bioenergetics through calcium fluxes, which are determinants of the activity of many matrix dehydrogenases, such as pyruvate dehydrogenase (PDH), α-ketoglutarate dehydrogenase (αKGDH) or isocitrate dehydrogenase (IDH)[15,16,42]. As CDK4-KO cells display altered MERC tethering and attenuated ER-MT calcium signaling, we evaluated the functionality of their mitochondria. First, we observed that the giant mitochondria in CDK4-KO cells had an altered cristae structure (Fig. 10a), with an increased number of cristae per mitochondrion (Fig. 10b). This increase correlated with the overall decrease in the length of cristae normalized to the mitochondrial area (Fig. 10b). In addition to investigating these markers of cristae remodeling, we evaluated the global mitochondrial membrane potential (MMP) using the ratio of MitoTracker Red (MMP-dependent) to MitoTracker Green (MMP-independent) fluorescence (Fig. 10c). Remarkably, CDK4-KO resulted in a reduction in this ratio, indicating a decreased MMP in CDK4-KO cells compared to WT cells (Fig. 10c).

To assess the changes in the main metabolites in TNBC cells in the absence of CDK4, we performed targeted mass spectrometry-based analysis of 200 polar and moderate metabolites in WT and CDK4-KO cell extracts. CDK4-KO cells showed enrichment of glycolytic metabolites compared to WT cells (Fig. 10d). Furthermore, the levels of four tricarboxylic acid (TCA) cycle metabolites were significantly altered in CDK4-KO cells (Fig. 10d, e). Among them, the isocitrate level was highly increased, in contrast to α-ketoglutarate, succinate and malate, which were reduced in CDK4-KO cells (Fig. 10d, e), suggesting a blockade in the conversion of isocitrate into α-ketoglutarate via IDH. As calcium regulates the activity of mitochondrial-dehydrogenase enzymes[42] (Fig. 10e), we assessed the activity of mitochondrial isocitrate dehydrogenase 3 (IDH3). The activity of mitochondrial IDH3 was consistently decreased in CDK4-KO cells (Fig. 10f). Importantly, exogenous reintroduction of CDK4 partially rescued this attenuated activity in CDK4-KO cells (Fig. 10f). We then further evaluated mitochondrial bioenergetics in CDK4-KO cells. In basal conditions,

dampened mitochondria-ER contact calcium signaling and deficient mitochondrial IDH3 activity were not responsible for a defective mitochondrial oxidative phosphorylation in CDK4-KO cells, as suggested by equal oxygen consumption rates with WT cells (Sup. Fig. 10a-b). We also tested the differential response of CDK4-KO cells to mitochondrial metabolic challenges. Suppression of glucose or its replacement by galactose resulted in the inhibition of glycolysis and induced the reliance on oxidative phosphorylation (OXPHOS) for ATP generation (Fig. 10g, h). While WT cells were able to enhance OXPHOS, CDK4-KO cells displayed less flexibility to boost it (Fig. 10g, h and Sup. Fig. 10c, d). Most importantly, forcing CDK4-KO cells to rely only on OXPHOS drastically reduced their viability (Fig. 10i, j), without affecting caspase-3 cleavage (Fig. 10k). Taken together, these data indicate that CDK4-KO cells, in addition to their ability to resist apoptotic stimuli, exhibit mitochondrial metabolic vulnerabilities that could be mediated by reduced MERC establishment and impaired ER-MT calcium signaling.

## Discussion

Dividing cells become sensitive to most chemotherapeutic agents, whereas resting cells are relatively resistant. Paradoxically, although CDK4/6 inhibition (CDK4/6i) have cytostatic effects, TNBC and ovarian cancer cells treated with doxorubicin, paclitaxel, or carboplatin were resistant to cytotoxic effects and the associated cell death in response to CDK4/6 inhibition[43–45]. In fact, CDK4/6i increases the expression of *BCL2L1* (encoding Bcl-xL) and *MCL1*, which encode two antiapoptotic members of the Bcl-2 protein family, ultimately resulting in apoptosis evasion[46]. CDK4/6i promotes the survival of kidney cells upon cisplatin or etoposide treatment[47–49] and of normal intestinal cells upon radiation-induced injury[50]. Furthermore, CDK4/6i increases the tolerability of chemotherapy in lung cancer patients, as observed by the diminished hematological toxicity and myelopreservation in multiple haematopoietic lineages[51]. Collectively, these data suggest that CDK4/6i can protect cells against chemotoxicity. Furthermore, recent data from the Malumbres laboratory showed that pretreatment of cancer cells with CDK4/6i rendered the cells resistant to chemotherapeutic drugs, whereas treatment after chemotherapy resulted in synergistic effects[52].

In our study, we reproduced the resistance to CDK4i using an in vivo xenograft model established with CDK4-KO TNBC cells. The dynamics of the acquisition of resistance were indeed very similar (Fig. 1i in our manuscript and Fig. 7e in[53]), suggesting that CDK4 plays a prominent role in the development and therapeutic resistance in this cancer. We demonstrated that the formation of CDK4-KO tumors in xenografted mice does not result from CDK4 re-expression that could be secondary to acquired mutations (Fig. 1k). It is, however, puzzling why there is a latency period of approximately 30 days, in which, like treated cells[53] CDK4-KO TNBC cells do not grow in grafted mice. This

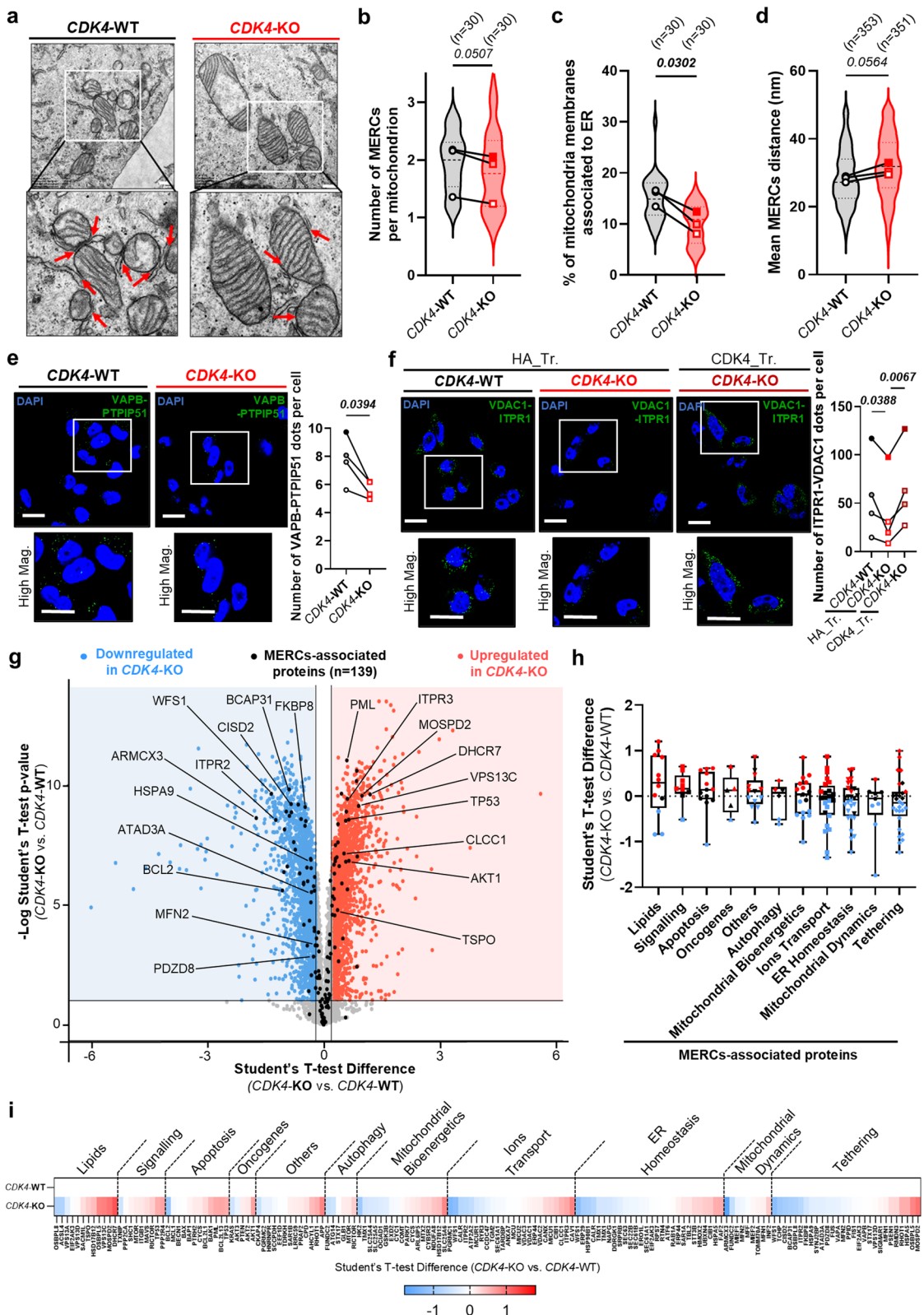

adaptation period may depend on the specific metabolic needs of the tumor cells that are required for their establishment in the host tissue and the interactions with surrounding cells for the development of the tumor microenvironment. Alternatively, an increased migration rate immediately after grafting the CDK4 KO cells would result in a decreased tumor growth in the implantation site. Further studies are

required, however, to elucidate the role of CDK4 in migration and early engraftment.

Another important finding of our study were the differences in mitochondrial dynamics and function in CDK4-KO cells, which add another layer of complexity to the understanding of the role of CDK4 in TNBC and possibly in other cancers. Mitochondria are vital

**Fig. 6 | CDK4 enhances Mitochondria-ER Contacts in TNBC. a** Representative electron micrographs of CDK4-WT and CDK4-KO MDA-MB-231 TNBC cells. Scale bars: 500 nm. Red arrows indicate Mitochondria-ER Contacts (MERCs). **b** Quantification of the number of MERCs per mitochondrion of CDK4-WT and -KO TNBC cells. Medians of N = 3 independent biological replicates from n=cells, represented as violin plot. Two-sided Paired T-test. **c** Percentage of mitochondrial membranes associated to ER. Means of N = 3 independent biological replicates from n=cells, represented as violin plot. Two-sided Paired T-test. **d** Mean ER-mitochondrion distance in analyzed MERCs. Means of N = 3 independent biological replicates representing n = 353 MERCs from 55 cells (WT) and n = 351 MERCs from 49 cells (KO), represented as violin plot. Two-sided Mann-Whitney test. **e** Proximity Ligation Assay (PLA) using VAPB and PTPIP51 antibodies in CDK4-WT and -KO TNBC cells. Representative pictures and associated quantification of VAPB-PTPIP51 dots per cell. Scale bar: 10 μm. N = 4 independent biological replicates. Two-sided Paired T-test. **f** Proximity Ligation Assay (PLA) using VDAC1 and ITPR1 antibodies in CDK4-

WT, CDK4-KO TNBC cells transfected with empty plasmid HA (HA_Tr.) and CDK4-KO TNBC cells expressing endogenous CDK4 (CDK4_Tr.). Representative pictures and associated quantification of ITPR1-VDAC1 dots per cell. Scale bar: 10 μm. N = 4 independent biological replicates. RM 1 way ANOVA; Tukey's multiple comparisons test. **g** Volcano plots of proteins differentially regulated in CDK4-WT and -KO TNBC cells, highlighting proteins significantly down-regulated (blue) and up-regulated (red) in CDK4-KO compared to CDK4-WT TNBC cells. N = 5 biological replicates. In black are indicated 139 MERCs-associated proteins found in the proteomic analysis. **h** Box plots of the subclustered classes of the 139 MERCs-associated proteins, from the top up-regulated to the top down-regulated according to the median of Student's T-test difference. Box spans from the first to the third quartile, marking the median with a distinct line. The whiskers reach out to the extreme maximum and minimum data point. **i** Heatmap of the 139 MERCs-associated proteins according to the median of Student's T-test difference. Exact *p-values* are displayed in italic (bold italic if <0.05).

organelles responsible for energy production, regulation of apoptosis, and various other cellular functions. Alterations in mitochondrial dynamics and function often have profound implications for cell survival and metabolism[54,55]. CDK4-KO cells exhibit defects in mitochondrial morphology that are the result of dysregulated mitochondrial dynamics, which can lead to impaired energy production, calcium signaling, and altered apoptotic responses.

The antiapoptotic effects of CDK4 deletion in TNBC involve the regulation of mitochondrial biology. Altered mitochondrial dynamics and function in CDK4-KO cells contribute to their resistance to apoptosis. Unlike WT cells, CDK4-KO TNBC cells failed to exhibit robust mitochondrial calcium uptake in response to apoptotic stress, a key trigger for apoptosis. Elevated intramitochondrial calcium induces mitochondrial permeability transition pore (mPTP) opening, disrupting mitochondrial membrane potential, dissipating the proton gradient, and releasing cytochrome c to activate caspase-9 in the intrinsic apoptotic pathway. z-vAD experiments further revealed that CDK4-KO cells resist not only apoptosis but also other cell death mechanisms (Fig. 3f)[56].

Calcium uptake into mitochondria is primarily mediated by mitochondria-endoplasmic reticulum contacts (MERCs), which comprise 10-20% of mitochondrial membranes and vary across species, tissues, and cell types[25]. We demonstrated in this study that CDK4 localizes at MERCs, modulating their tethering and influencing their signaling through PKA deregulation. Indeed, the decreased number of MERCs in CDK4-KO cells translates into decreased uptake of calcium into mitochondria in response to apoptotic stimuli, thereby protecting cells from death. We identified the protein kinase PKA as a target of CDK4 at the MERC interface. A recent study suggested that PKA promotes mitochondrial remodeling and ER-MT calcium signaling during the early phase of ER stress[57]. This enhanced ER-MT calcium signaling thus favors mitochondrial bioenergetics during the adaptive response. PKA signaling is a determinant of MERC formation, as it mediates the phosphorylation of key proteins involved in calcium signaling and mitochondrial fission, such as ITPR1[38] and the pro-fission protein DRP1[58], respectively. AKAP1, which mediates PKA subunit translocation to the OMM, also localizes at MERCs[59], reinforcing the importance of PKA signaling in this subcellular compartment. Here, we demonstrated an upstream mechanism of PKA regulation through CDK4 kinase activity. Of note, in the analysis of PKA motif phosphosites differentially regulated at MERCs, we identified many other calcium channels, such as VDAC1. This latter finding may facilitate the understanding of the biological complexity of mitochondrial PKA signaling[35,36,60]. Indeed, the regulation of PKA activity at the MERC interface remains underexplored but could explain how and why the activities of mitochondrial and cytosolic PKA have opposite effects on apoptosis.

Mitochondrial dysfunction in CDK4-KO TNBC cells had other physiological consequences in addition to protection against cell

death. Metabolic flexibility, which is required for the response to metabolic challenges in cancer, was compromised in CDK4-KO TNBC cells (Sup. Fig 11). Previous reports have provided evidence that OXPHOS and mitochondrial functions constitute metabolic vulnerabilities in metastatic breast cancers with resistance to the CDK4 inhibitor palbociclib[61]. Similar effects were also observed in pancreatic cancer[62], colorectal cancer[63], and melanoma[64]. In TNBC, CDK4-KO cells failed to enhance oxidative phosphorylation under metabolic stress, showing defective mitochondrial function. Metabolomic and gene expression analyses revealed decreased activity of calcium-regulated mitochondrial IDH3 in CDK4-KO cells, with reduced levels of α-ketoglutarate and succinate, key metabolites in the TCA cycle.

Our findings using cellular and mouse xenograft models suggested that CDK4 inactivation could have unexpected consequences in the clinical setting. Indeed, our bioinformatic analyses of data obtained from human breast cancer patients treated with CDK4/6 inhibitors further indicated a negative effect of CDK4 activity on survival and resistance to the treatment.

In summary, this study reveals a role for CDK4 in the regulation of cell death through the control of MERCs, directly impacting mitochondrial dynamics and calcium signaling. The identification of this function could lead to a new paradigm in the treatment of metastatic breast cancer and possibly other cancers through consideration of how CDK4i treatment could promote resistance to chemotherapy. Our findings also pave the way for defining new therapeutic strategies based on the metabolic vulnerabilities acquired when CDK4 is inhibited.

## Methods

### Cell culture, viability, plasmid transfection and probes

TNBC MDA-MB-231 (ATCC), BT-474 (ATCC) and ER + PR + MCF-7 (ATCC) were cultured in Roswell Park Memorial Institute (RPMI)-1640 medium with GlutaMAX (Gibco™, Ref: 61870036) supplemented with 10% heat-inactivated FBS (HyClone, Cat. No: SV30160, Lot: RB35956), 10 mM HEPES (Gibco™, Ref: 15630122) and 1 mM pyruvate (Gibco™, Ref: 11360070). TNBC HCC1806 (ATCC) were cultured in Dulbecco's Modified Eagle Medium (DMEM, Gibco Ref. 31966047, Lot. 2797902) with the same previous supplementation. MDA-MB-468 (ATCC) was maintained in 1:1 DMEM: Leibovitz's L-15 (Gibco 11415-049, Lot 11415-056), supplemented with 10% FBS (Fetal Bovine Serum, qualified, USDA-approved regions Lot: 2575611, reserve 2561393 S2). Cells were grown under normoxic conditions at 37 °C and 5% CO2. Triple-negative NST breast cancer luciferase-expressing cells were dissected and viable frozen from a xenografted primary breast tumor derived from female patient 10 (T70), as described in ref. 65. Frozen breast cancer cells were plated at 2 million viable cells per well in a 6-well tissue culture dish. 5 μg/mL D-firefly luciferin potassium salt (Biosynth, L-8220) was added to the cells, and bioluminescence was measured kinetically over a time

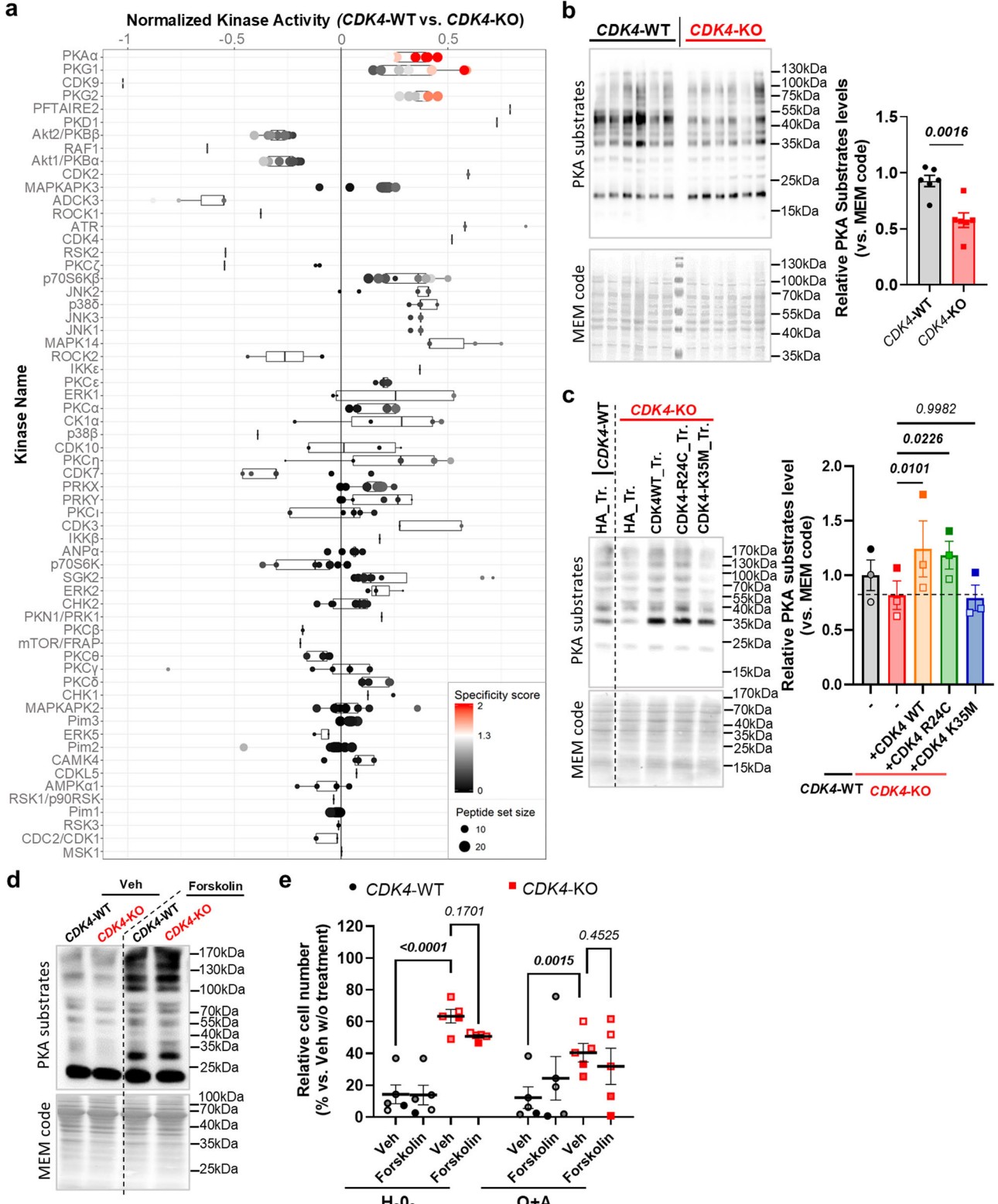

**Fig. 7 | PKA activity is regulated by CDK4 but is not sufficient to mediate apoptosis. a** Boxplot of kinome profiling/PAMgene analysis CDK4-WT and CDK4-KO TNBC cells. Comparison CDK4-KO vs. CDK4-WT TNBC cells. N = 4 independent biological replicates. Box spans from the first to the third quartile, marking the median with a distinct line. The whiskers reach out to the extreme maximum and minimum data point. **b** Immunoblots of Phospho-PKA substrates and MEM code in CDK4-WT and -KO TNBC cells. Relative quantity of Phospho-PKA substrates normalized to MEM code. Mean +/- SEM of N = 6 independent biological replicates. Two-sided Unpaired T-test. **c** Immunoblots of Phospho-PKA substrates and MEM code of CDK4-WT and -KO TNBC cells. Relative quantity of Phospho-PKA substrates relative to MEM code. Mean +/- SEM of N = 3 independent biological replicates. **d** Immunoblots of Phospho-PKA substrates and MEM code of CDK4-WT and -KO TNBC cells treated for 24 hours pretreatment with Forskolin (20 μM). **e** Relative cell number compared to control (no treatments) in CDK4-WT and -KO TNBC cells, upon 24 hours pretreatment and 3 consecutive days of Forskolin (20 μM). Cell death was induced in parallel with $H_2O_2$ (250μM-2hours) or Oligomycin and Antimycin (O + A) (1 + 10μM-2hours). Mean +/- SEM of N = 5 independent biological replicates. 2-way ANOVA; Tukey's multiple comparison tests. Exact *p-values* are displayed in italic (bold italic if <0.05).

course of 2 days by an automated Spark® Multimode Microplate Reader. CDK4/6 inhibitor Abemaciclib diluted in DMSO (HY-16297A/MedChemExpress), CDK2 inhibitor AUZ-454 (HY-15004/MedChemExpress) diluted in DMSO, Cisplatin (HY-17394/MedChemExpress) diluted in $H_2O$, 5-Fluorouraracil (5-FU) (F6627/Sigma Aldrich) diluted in $H_2O$, Doxorubicin (Catalog No.S1208/SellekChem) diluted in DMSO, TRAIL (310-04/Peprotech) diluted in PBS 1X. $UV_B$ was used at $30mJ/cm^2$. Viability tests were performed using PrestoBlue™ Cell Viability Reagent (A13261/ThermoFisher) according to manufacturer recommendations and 48 h after induction of cell death. The number of total cells was evaluated using Hoescht staining and analysis from CELENA® X High Content Imaging System (Labtech) from 3 days to 5 days after induction of cell death and according to figure legends. Z-VAD-FMK (HY-16658B/MedChemExpress) was diluted in DMSO and used in pretreatment for 3 hours and during cell death induction at 20 µM. Forskolin (F3917/Merck) was diluted in EtOH used in pretreatment for 48 hours and during cell death induction at 20 µM. The method used to generate MDA-MB-231 CDK4-KO cells via CRISPR/Cas9 gene editing is described in ref. 10. Empty-HA (Ctrl), CDK4-WT, CDK4-R24C, and CDK4-K35M plasmids were provided by Marcos Malumbres (CNIO, Spain). For transfection, cells were plated one day before transfection and were then incubated with a 2:1 ratio (µL:µg) of X-tremeGENE™ HP DNA Transfection Reagent (Roche, Ref: 6366236001) and plasmids for 20 min in Opti-MEM (Gibco™, Ref: 51985034). To track total and active mitochondria, we incubated cells with MitoTracker Green (Invitrogen™, Ref: M7514) and MitoTracker Red (Invitrogen™, Ref: M22425) (200 nM) or TMRM (ThermoFisherScientific, Ref: T668) (100 nM), respectively, in Opti-MEM (Gibco™, Ref: 51985034) for 30 min at 37 °C and 5% $CO_2$. *MMP image analysis.* The ratio between Mitotracker Red/Green was measured as described with Fiji/ImageJ 1.50b (NIH, Bethesda, MD, USA) (https://doi.org/10.1186/s13075-019-1974-z). Specifically, micrographs of at least 10 cells from 2 independent biological replicates were taken with the confocal microscope system Zeiss LSM 880. The images were median-filtered and selected from the Z-stack the middle plane for each cell separately, to avoid the artificial overlapping of Z-projections. We next split the color channels of the images converted to 8-bit. After setting the scale of the image and selecting the region of interest (i.e., the whole cells), the mean gray value of the two MitoTrackers, displayed in each image channel, were measured in grayscale (0-255, a.u.). Then, the ratio of Red MitoTracker/Green MitoTracker per cell and condition was calculated. For the three-dimensional representation, a stack was created with the 5 middle planes of magnifications from the median-filtered images, previously converted to 8-bit. The default binary threshold was applied for both channels, which were subsequently overlapped. The merged images were opened with the Volume Viewer plugin of Fiji/ImageJ 1.50b, and run in Volume mode, with tricubic smooth interpolation (Z-Aspect: 1, Sampling: 1) and white background. Finally, the maximum global aplha offset was applied, and the snapshots were taken.

## Animals

NOD SCID gamma (NSG) mice from The Jackson Laboratory were bred at the animal facility specific pathogen-free of the University of Lausanne, Epallinges, and housed under standard conditions (standard diet and water ad libitum) at 23 °C, 50% hygrometry, with 12 hrs light and 12 hours dark cycles. $2 \times 10^6$ MDA-MB-231 CDK4-WT or -KO cells in 50 µL of 1X PBS were injected into the fat pad of the $4^{th}$ mammary gland of 8-week-old females. Tumor volume was assessed by measuring the length and the width with a caliper. The mice received Cisplatin (4 or 8 mg/kg) by intraperitoneal injection when the tumors reached a mean size volume of 80-120 mm³, and the treatment was performed as shown in Fig. 2h. Tumor size did not exceed the maximal size (1.5 cm length, width, height) or volume (1000mm³) permitted by ethics committee. The experiments were approved by the Direction générale de l'agriculture, de la viticulture et des affaires vétérinaires (DGAV)

(Cantonal License #VD3726, National License #34118) according to art. 18 Animal Welfare Act (SR 455), art. 141 Animal Welfare Ordinance (SR 455.1), art. 30 Animal Experimentation Ordinance (SR 455.163). The conducted methods were in accordance with the animal care guidelines of Swiss laws, following 3 R recommendations.

## RT-qPCR and RNA sequencing

Reads were mapped to the mouse genome GRCh38 (Ensembl version 102) using STAR (v. 2.7.0 f; options: --outFilterType BySJout –outFilterMultimapNmax 20 –outMultimapperOrder Random –align SJoverhangMin 8 –alignSJDBoverhangMin 1 –outFilterMismatchNmax 999 –alignIntronMin 20 –alignIntronMax 1000000 –alignMates GapMax 1000000). Read counts in gene loci were evaluated with HTSeq-count (v. 0.13.5). Differential expression analysis was performed in R (v 4.1.1) using the DESqe2 package with $p$ value adjustments for multiple comparisons (Benjamini–Hochberg procedure). Principal component analysis (PCA) was performed in R using variance-stabilized expression data for the 500 genes with the greatest variability. GSEA was performed in R using MsigDB and the fgsa Bioconductor package. Kyoto Encyclopedia of Genes and Genomes (KEGG) and Gene Ontology (GO) enrichment analyses were performed in R using the ReactomePA Bioconductor package. For RT-qPCR, RNA was isolated by the Trizol-Chloroforme method extraction and subsequent isopropanol-based precipitation. Reverse transcription was performed with 400 ng of RNA using SuperScript™ II Reverse Transcriptase (ThermoFisher). 5 ng of cDNA was amplified with specific primers and Maxima™ SYBR green/ROX QPCR master mix (2X) (Thermofisher). The primer sequences are listed in Sup. Table 4.

## Proteomics and Phospho-Proteomics

Total cells were lysed in RIPA buffer, 2% SDS, 10 mM DTT. Tryptic Digestion was done following the SP3 method[66] using magnetic Sera-Mag Speedbeads (Cytiva 45152105050250, 50 mg/mL). After heating for 10 min at 75 °C, cysteines were alkylated with 32 mM (final) iodoacetamide for 45 min at RT in the dark. Beads were added at a ratio 10:1 (w:w) to samples, and proteins were precipitated on beads with ethanol (final concentration: 60 %). After 3 washes with 80% ethanol, on-beads digestion was performed in 40 µl of 50 mM Hepes pH 8.3 with 2 µg of trypsin (Promega #V5073). Supernatants were harvested for TMT labeling. Supernatants from SP3 digestion containing each 100 µg peptides in 50 µL Hepes pH 8.3 buffer were mixed each with 0.4 mg of TMT 10-plex reagent (Thermo Fisher Scientific product nr 90111) dissolved in anhydrous acetonitrile and incubated at RT for 90 min. Excess reagent was quenched by adding 7 µl of hydroxylamine 5% (v:v) and incubating 15 min at RT. After mixing and evaporation of excess acetonitrile, samples were acidified and desalted on SepPak C18 cartridges. Peptides were eluted in 50% acetonitrile and dried. Aliquots (1/10) of samples were analyzed directly by LC-MS to determine total proteome composition. Phosphopeptide enrichment by IMAC was performed on the remaining of TMT-labeled material (approx. 0.4-0.6 mg) with the High-Select™ Fe-NTA Phosphopeptide Enrichment Kit (Thermo Fisher Scientific, Prod. Number A32992) according to kit instructions. IMAC eluates were dried and resuspended for analysis in 2% MeCN, 0.05% TFA. Data-dependent LC-MS/MS analyses of samples were carried out on a Fusion Tribrid Orbitrap mass spectrometer (Thermo Fisher Scientific) connected through a nano-electrospray ion source to an Ultimate 3000 RSLCnano HPLC system (Dionex), via a FAIMS interface. Peptides were separated on a reversed-phase custom packed 45 cm C18 column (75µm ID, 100 Å, Reprosil Pur 1.9 µm particles, Dr. Maisch, Germany) with a 4-90% acetonitrile gradient in 0.1% formic acid (140 min). Data files were analysed with MaxQuant 1.6.14.0 incorporating the Andromeda search engine[67]. Cysteine carbamidomethylation and TMT labelling (peptide N-termini and lysine side chains) were selected as fixed modification while methionine oxidation and protein N-terminal acetylation were specified as variable

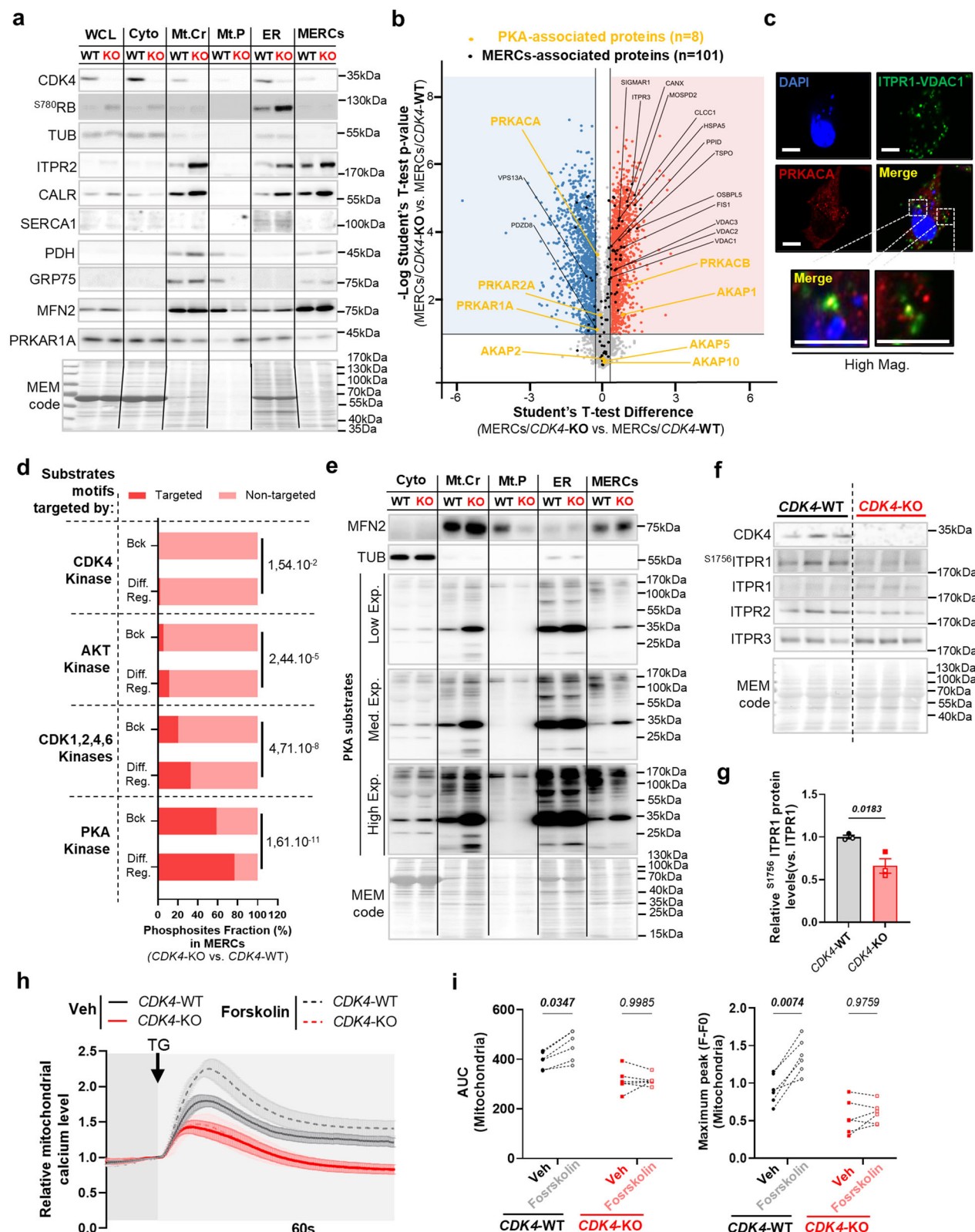

modifications. Both peptide and protein identifications were filtered at 1% FDR relative to hits against a decoy database built by reversing protein sequences. For TMT analysis, the raw reporter ion intensities generated by MaxQuant (with a mass tolerance of 0.003 Da) and summed for each protein group were used in all following steps to derive quantitation. All subsequent analyses were done with the Perseus software package (version 1.6.15.0). Contaminant proteins were

removed. After imputation of missing values (based on normal distribution using Perseus default parameters), t-tests were carried out among all conditions, with permutation-based FDR correction for multiple testing (Q-value threshold <0.01). The difference of means obtained from the tests were used for 1D enrichment analysis on associated GO/KEGG annotations as described[68]. The enrichment analysis was also FDR-filtered (Benjamini-Hochberg, Q-val<0.02). The

**Fig. 8 | MERCs-PKA activity is regulated by CDK4 and drives ER-MT calcium signaling. a** Immunoblots of CDK4, [S780]RB, Tubulin (TUB), ITPR2, CALR, SERCA1, PDH, GRP75, MFN2, PRKAR1A and MEM code in subcellular fractions of CDK4-WT and CDK4-KO TNBC cells. WCL=Whole cell lysate. Cyto=Cytosolic fraction. Mt.Cr=Crude mitochondria fractions. Mt.P=Pure mitochondria fractions. ER=Endoplasmic reticulum fraction. MERCs=Mitochondria-ER contacts fractions. Representative of N = 3 biological replicates. **b** Volcano plot of proteomics data from the MERCs fraction of CDK4-WT and -KO TNBC cells. MERCs- and PKA-associated proteins are represented respectively in dark and orange. **c** Immunofluorescence of MDA-MB-231 TNBC cells stained with DAPI (blue), PRKACA (red), ITPR1-VDAC1 puncta (green). Scale bars: 10 µm. Representative of N = 3 biological replicates. **d** Enrichment analysis on phospho-peptides found in MERCs fraction of CDK4-WT and -KO TNBC cells. Phosphosites displaying CDK4 kinase motifs, AKT kinase motifs, CDK1,2,4,6-kinase motifs or PKA kinase motifs are represented. Bck=Background represents the total number of phosphosites found in the phosphoproteomics while Diff.Ref.=Differentially regulated represents the number of phosphosites found significantly down- or up-regulated in MERC fraction of CDK4-

KO TNBC cells. N = 3 independent biological replicates. Benj. Hoch. FDR value is displaying for each substrate motif category. **e** Immunoblots of MFN2, Tubulin (TUB), Phospho-PKA substrates and MEM code of CDK4-WT and -KO TNBC cells on different subcellular fractions: whole cell lysate, cytosolic, crude mitochondria, pure mitochondria, ER and mitochondria-ER contacts. Representative of N = 3 biological replicates. **f, g** Immunoblots and relative protein levels of CDK4, [S1756]ITPR1, ITPR1, ITPR2, ITPR3 and MEM code of CDK4-WT and -KO TNBC cells. Mean +/- SEM of N = 3 independent biological replicates. Two-sided Unpaired T-test. **h, i** Relative mitochondrial calcium levels of CDK4-WT and -KO TNBC cells pretreated with or without Forskolin (20 µM) upon Thapsigargin (TG) (2 µM) injection. Arrow indicates the time of injection. Associated quantification of the area under the curves (AUC) and maximum amplitude/peak. Mean +/- SEM of N = 6 independent injections accounting 3 biological replicates representing a total of n = 89 cells (Veh/CDK4-WT), n = 88 cells (Veh/CDK4-KO), n = 82 cells (Forskolin/CDK4-WT) and n = 86 cells (Forskolin/CDK4-KO). RM 1 way ANOVA; Tukey's multiple comparisons test. Exact *p-values* are displayed in italic (bold italic if <0.05).

mass spectrometry proteomics data including raw output tables have been deposited to the ProteomeXchange Consortium via the PRIDE partner repository[69] with the dataset identifiers PXD046326, PXD046327 and PXD046353. Phosphosite analysis[70] was used to determine CDK4 phosphorylation score on S83 of PRKAR1A and PRKAR1B in Sup. Fig. 7b

### Kinome profiling (PamGene)

For kinome analysis, serine/threonine kinase (STK) microarrays were purchased from PamGene International BV. Each array contained 140 phosphorylatable peptides as well as 4 control peptides. Sample incubation, detection, and analysis were performed according to the manufacturer's instructions in a PamStation 12 instrument. In brief, extracts from BMDMs or human visceral adipose tissue (VAT) were prepared using M-PER mammalian extraction buffer (Thermo Scientific) containing Halt phosphatase inhibitor cocktail (1:50; Thermo Scientific) and Halt protease inhibitor cocktail, EDTA-free (1:50, Thermo Scientific) for 20 min on a rotating wheel at 4 °C. The lysates were then centrifuged at 16 000 rpm for 20 min to remove all debris. The supernatants were aliquoted, snap-frozen in liquid nitrogen, and stored at -80 °C until further processing. Prior to incubation with the kinase reaction mix, the arrays were blocked with 2% BSA for 30 cycles and washed three times with PK assay buffer. Kinase reactions were performed over a 1-h period with 5 µg of total extract and 400 µM ATP at 30 °C. Phosphorylated peptides were detected with a -FITC-conjugated anti-rabbit secondary antibody that recognizes a pool of anti-phosphoserine/threonine antibodies. The PamStation 12 instrument contains a 12-bit CCD camera suitable for imaging of FITC-labeled arrays. The images of the phosphorylated arrays were used for quantification with BioNavigator software (PamGene International BV). The generated heatmaps and BioNavigator score plots are explained in the results section and figure legends.

### Electron microscopy and MERCs analysis

Mouse tissue was cut into 1 mm³ pieces and fixed with 2.5% glutaraldehyde solution (EMS, Hatfield, PA, US) in phosphate buffer (PB; 0.1 M, pH 7.4) (Sigma, St. Louis, MO, US) for 1 h at room temperature (RT). Then, the samples were rinsed 3 times for 5 min each in PB buffer and postfixed with a fresh mixture of 1% osmium tetroxide (EMS, Hatfield, PA, US) and 1.5% potassium ferrocyanide (Sigma, St. Louis, MO, US) in PB buffer for 1 h at RT. The samples were then washed three times in distilled water and dehydrated in acetone solution (Sigma, St. Louis, MO, US) at graded concentrations (30%, 40 min; 70%, 40 min; 100%, 1 h; 100%, 2 h). This step was followed by infiltration in Epon resin (Sigma, St. Louis, MO, US) at graded concentrations (Epon 1/3 acetone, -2 h; Epon 3/1 acetone, -2 h, Epon 1/1, -4 h; Epon 1/1, -12 h) and polymerization for 48 h at 60 °C in an oven. Thin sections of 50 nm

were sliced on a Leica Ultracut microtome (Leica Mikrosysteme GmbH, Vienna, Austria) and picked up on a copper grid (2 × 1 mm; EMS, Hatfield, PA, US) coated with a polyetherimide (PEI) film (Sigma, St. Louis, MO, US). Sections were sequentially poststained with 2% uranyl acetate (Sigma, St. Louis, MO, US) in $H_2O$ for 10 min, rinsed several times with $H_2O$ followed by Reynolds lead citrate for 10 min and rinsed several times with $H_2O$. Large montages were acquired with a Philips CM100 transmission electron microscope (Thermo Fisher Scientific, Hillsboro, US) at an acceleration voltage of 80 kV with a TVIPS TemCam-F416 digital camera (TVIPS GmbH, Gauting, Germany), and alignment was performed using the Blendmont command-line program in IMOD software (Kremer, Mastronarde, et McIntosh 1996). MERCs were defined as regions where the distance between the ER and OMM membranes was less than 50 nm. N = 3 (for cells) and N = 2 (for xenograft tumors) independent biological replicates were analyzed. To avoid any baso-apical bias, cells displaying cytoplasm without nuclei were excluded. The quantification of the number of MERCs per mitochondrion, the length of MERCs, and the average/minimal distances below 50 nm between the ER and OMM membranes per MERC was performed using a custom macro in FiJi software kindly provided by G. Hajnoczky[26,71]. This macro was also used to calculate the number of mitochondria per cell and to assess the perimeter of each mitochondrion involved in a MERC contact with the ER membrane in TEM micrographs.

### MERC isolation

A total of $100 × 10^6$ *CDK4*-WT and *CDK4*-KO MDA-MB-231 TNBC cells were plated for isolation of subcellular fractions as described in ref. 72.

### Live calcium imaging and time-lapse microscopy

According to figure legends, chemical non-ratiometric and genetic ratiometric calcium probes were used. *Mitochondrial and cytosolic non-ratiometric probes*. For cytosolic calcium measurement, cells were loaded with 1 mL of Krebs solution with calcium (mM: NaCl 135.5, $MgCl_2$ 1.2, KCl 5.9, glucose 11.5, HEPES 11.5, $CaCl_2$ 1.8; final pH, 7.3) containing 5 µM of the cytosolic $Ca^{2+}$ indicator Fluo-4 AM (Invitrogen, Switzerland) by incubation for 20 min. For mitochondrial calcium measurement, cells were incubated with 1 mL of Krebs solution with calcium containing 1 µM of the fluorescent mitochondrial indicator Rhod-2 AM (Invitrogen, Switzerland) for 1 h at RT. The cells were then washed twice with calcium-free Krebs solution and all subsequent mitochondrial calcium measurements were performed in a zero calcium-containing buffer to exclude any contribution of SOCE to mitochondrial calcium uptake. The fluorescence intensity of chemical probes were monitored using a Zeiss LSM 780 Live confocal microscope system with a 40× oil immersion lens. For Fluo-4 AM, the excitation wavelength was set at 488 nm, and the emitted fluorescence was

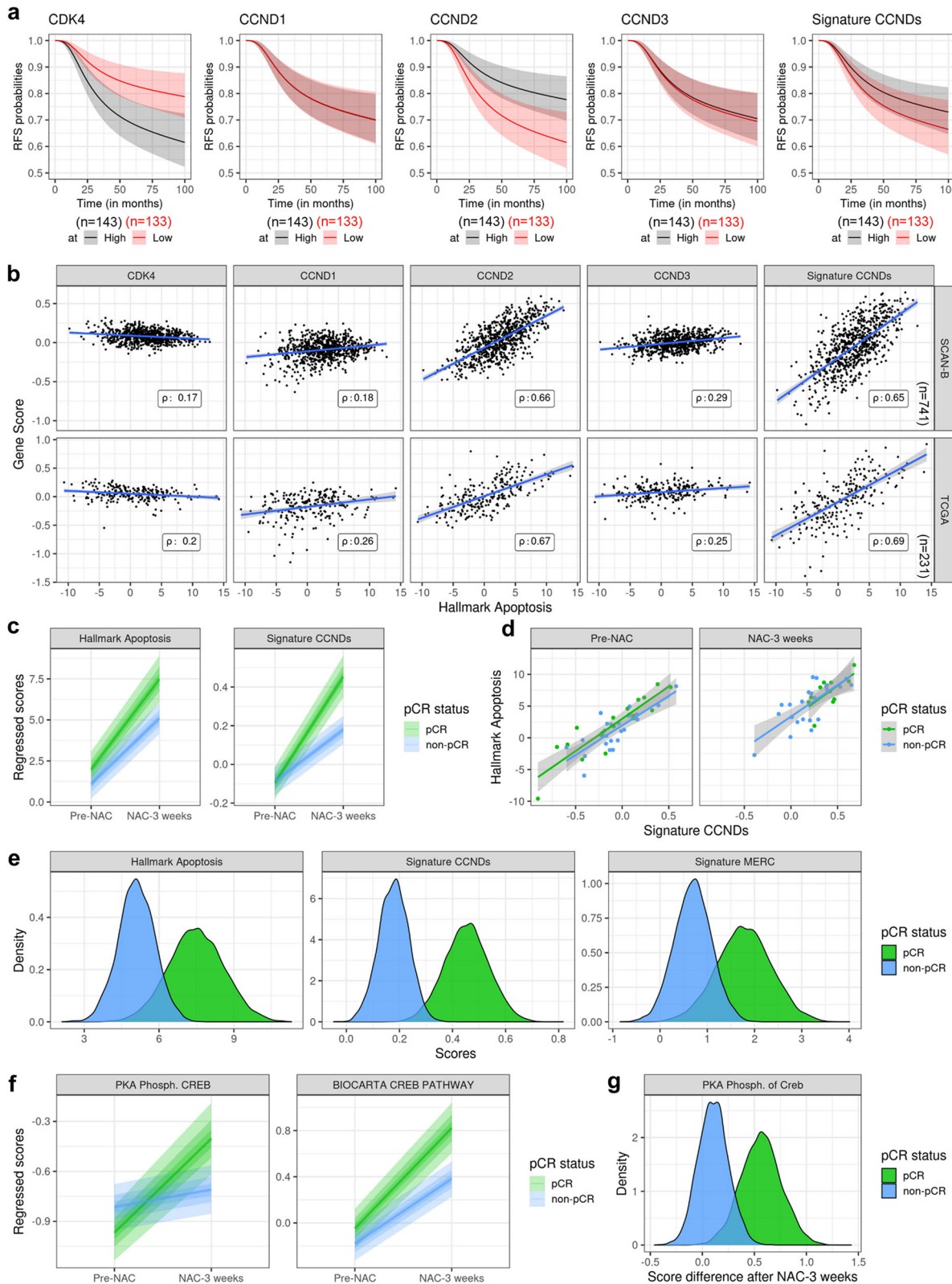

collected through a bandpass filter (495-525 nm). For Rhod-2 AM, the excitation wavelength was set at 532 nm, and the emitted fluorescence was collected through a bandpass filter (540–625 nm). *Mitochondrial Ratiometric probe*. FRET-based $Ca^{2+}$ sensors 4mtD3cpv[26] was used to measure mitochondrial calcium level in a zero calcium-containing buffer (0.14 M NaCl, 5 mM KCl, 1 mM $MgCl_2$, 10 mM HEPES, and 1 mM EGTA with 10 mM glucose). Mitochondrial $Ca^{2+}$ measurements were

performed at 37 °C using a wide-field Leica DMI6000B microscope with a 40× oil-immersion objective and an ORCA-Flash4.0 digital camera (HAMAMATSU). *Calcium kinetics calculation*. Thapsigargin (2 μM) or Histamine (50 μM) was independently added 60 sec after starting the acquisition of the region of interest (ROI) for each individual cell. Changes in probes fluorescence were calculated by reporting the peak fluorescence intensity with respect to the baseline

**Fig. 9 | CDK4 activity is positively correlated with apoptosis signature and better response to neoadjuvant chemotherapy (NAC) in TNBC patients.**
**a** Dichotomized standardized survival curves showing recurrence-free survival (RFS) probabilities in two subsets of patients according to their gene score expression based on median of CDK4, CCND1, CCND2, CCND3 and signature of CCNDs. 95% confidence intervals are displayed. SCAN-B TNBC patients treated with chemotherapy only. n = 143 (High) and n = 133 (Low) patients[39]. **b** Correlation of gene score enrichment between Hallmark Apoptosis and CDK4, CCND1, CCND2, CCND3 or a signature of CCNDs expression for SCAN-B and TCGA datasets[39,40]. Rho values correspond to the Spearman's rank correlation between both signatures in each dataset. n = 741 (SCAN-B) and n = 231 (TCGA) patients. **c** Regressed scores of Hallmark Apoptosis and signature of CCNDs in TNBC patients preceding (pre-NAC) or three weeks after neoadjuvant chemotherapy (NAC-3 weeks) and according to their pathological complete response (pCR) status. n = 26 (pCR) and n = 18 (non-pCR) patients pre-NAC. n = 24 (pCR) and n = 11 (non-pCR) patients NAC-3 weeks. 95% credible intervals are displayed[41]. **d** Correlation between hallmark apoptosis and the CCNDs signature in TNBC patients preceding (pre-NAC) or three weeks after neoadjuvant chemotherapy (NAC-3 weeks) and according to their pathological complete response (pCR) status. n = 26 (pCR) and n = 18 (non-pCR) patients pre-NAC. n = 24 (pCR) and n = 11 (non-pCR) patients NAC-3 weeks. 95% credible intervals are displayed. **e** Density differences in Hallmark Apoptosis, signature CCNDs and signature Mitochondria-ER contact (MERC) three weeks after neoadjuvant chemotherapy (NAC-3 weeks), stratified by pCR status. n = 24 (pCR) and n = 11 (non-pCR) patients. **f** Regressed scores of PKA Phosph. CREB and BIOCARTA CREB PATHWAY in TNBC patients preceding (pre-NAC) or three weeks after neoadjuvant chemotherapy (NAC-3 weeks) and according to their pathological complete response (pCR) status. n = 26 (pCR) and n = 18 (non-pCR) patients pre-NAC. n = 24 (pCR) and n = 11 (non-pCR) patients NAC-3 weeks. 95% credible intervals are displayed. **g** Density scores differences of PKA Phosph. CREB three weeks after neoadjuvant chemotherapy (NAC-3 weeks). n = 24 (pCR) and n = 11 (non-pCR) patients.

fluorescence intensity (as the corresponding relative cytosolic or mitochondrial calcium concentration), equivalent to the mean intensity during the first 60 sec of the acquisition. For ratiometric probe, the fluorescence ratio was analyzed with MetaFluor 6.3 (Universal Imaging) after removing of fluorescence background. The mean of one biological replicate included data from 2 to 6 technical replicates with injection (accounting for 20 to 250 analyzed cells according to figure legends). For the pictures obtained in Fig. 4e and in Sup. Movies 1 and 2, label-free images were acquired using a 3D-Cell Explorer-fluo (Nanolive SA, Tolochenaz, Switzerland) microscope.

**Immunoblotting, immunofluorescence staining and proximity ligation assay**

For immunoblot analysis, cells were lysed using M-PER™ Mammalian Protein Extraction Reagent (Thermo Scientific, Ref: 78501) complemented with 1X protease and phosphatase inhibitor cocktails (Thermo Scientific, Ref: 78429 and Ref: 1861277). After protein quantification using the Bradford assay, 15-20 μg of protein was loaded onto a gel, resolved by SDS–PAGE and then transferred to a nitrocellulose membrane (Amersham Protran, Ref: 10600002). Membranes were blocked with 1X TBS -containing 0.05% Tween/3% BSA for 1 h and incubated at 4 °C with primary antibodies overnight. The next day, the membranes were washed 3 times with 1X TBS -containing 0.05% Tween and further incubated with a secondary antibody diluted in 1X TBS -containing 0.05% Tween/5% skim milk for 1 h at RT. Membranes were then washed 3 times with 1X TBS -containing 0.05% Tween, and bands were detected using an ECL kit corresponding to the antibodies used (Amersham, Ref: RPN2235 and Advansta, Ref: K-12045-D20). *Immuno-fluorescence staining.* For in vivo samples, xenograft tumors were harvested, washed with 1X PBS, and fixed for 16 h in 4% paraformaldehyde at +4 °C. The xenograft tumors were then washed with 1X PBS and embedded in paraffin. Tumors were sectioned on a Microm HM325 microtome at a thickness of 4 μm. The sections were rehydrated, and antigen retrieval was performed in citrate buffer (pH 6). Following blocking with 5% normal goat serum (NGS) (in 1X PBS) for 1 h, the cells were incubated overnight at 4 °C with the following primary antibodies: anti-mouse cleaved Caspase-3 (Cell Signaling, 9661S; 1:100 dilution) and anti-rat Ki67 (eBioscience 41-5698-80; 1:100 dilution). The following day, the sections were washed in 1X PBS before incubation with Alexa Fluor 488-conjugated goat anti-mouse (1:500) and Alexa Fluor 568-conjugated goat anti-rat (1:500) secondary antibodies for 30 min at RT. The sections were counterstained with DAPI (Invitrogen). For in vitro samples, cells were rinsed with 1X PBS and then fixed with 4% Methanol-free paraformaldehyde. Cells were then blocked for 60mn at RT in blocking buffer (1X PBS / 5% Normal Donkey Serum / 0.3% Triton X-100). Primary antibodies were diluted according to Sup. Table 5 in 1X PBS/ 1% BSA / 0.3% Triton X-100 overnight at 4 °C.

The day after, cells were washed three times in 1X PBS for 5 mn each at RT. Cells were incubated with secondary antibody (from LuBioScience: Ref: 715-546-150-Alexa Fluor 488 Donkey Anti-Mouse IgG, Ref: 711-546-152-Alexa Fluor 488 Donkey Anti-Rabbit IgG, Ref: 715-606-150- Alexa Fluor 647 Donkey Anti-Mouse IgG, 711-606-152- Alexa Fluor 647 Donkey Anti-Rabbit IgG) diluted to 1/500 with Hoescht (1/10 000) in 1% BSA for 30mn at RT in the dark, then rinsed three times with 1X PBS before mounting. *MERCs image analysis.* Percentage of MERCs (normalized to mitochondria area) was analyzed using ImarisColoc plugin of the software Imaris 9.9.0 (Oxford instruments, Abingdon, Oxfordshire, England, United Kingdom). Specifically, the indicated parameter was calculated in Z-stack micrographs from cells immunostained for Calreticulin (ER marker) and ATP5 (mitochondria marker), captured with the confocal microscope system Zeiss LSM 880. For the three-dimensional representation, a binary threshold (35/255 a.u) based on JACoP-Costes' method[73] was applied for both channels, which were subsequently overlapped. The merged images were opened with the 3D View tool of Imaris 9.9.0 (Oxford instruments, Abingdon, Oxfordshire, England, United Kingdom) and normal shading and white background were applied. Finally, snapshots and animations were carried out. *MERCs image analysis.* Percentage of MERCs was evaluated through Mander's coefficient (normalized to mitochondria area) using ImarisColoc plugin analyzed using ImarisColoc plugin of the software Imaris 9.9.0 (Oxford instruments, Abingdon, Oxfordshire, England, United Kingdom). Specifically, the indicated parameter was calculated in Z-stack micrographs from at least X cells (from X independent biological replicates) immunostained for Calreticulin (ER marker) and ATP5 (mitochondria marker), captured with the confocal microscope system Zeiss LSM 880. For the three-dimensional representation, a binary threshold (35/255 a.u) based on JACoP-Costes' method was applied for both channels[73], which were subsequently overlapped. The merged images were opened with the 3D View tool of Imaris 9.9.0 (Oxford instruments, Abingdon, Oxfordshire, England, United Kingdom) and normal shading and white background were applied. Finally, snapshots and animations were carried out.

For the PLA, Duolink® reagents were used (Merck, Ref: DUO92014, DUO92002 DUO92004). Cells were washed with 1X PBS and fixed for 10 min in 10% paraformaldehyde. An equal volume of 1 M glycine was added, and the cells were then washed with 100 mM glycine for 15 min. The cells were permeabilized with 0.1% Triton-X-100 and incubated overnight at 4 °C with primary antibodies. Cells were washed with 1X PBS containing 0.3% Tween and incubated with PLA probes. Ligation and polymerization were performed according to the manufacturer's recommendations (Merck, Duolink® procedure). All primary antibodies and dilutions used are listed in Sup. Table 5. PLA punctae quantification was performed using "Analyze particles" tool of FiJi software.

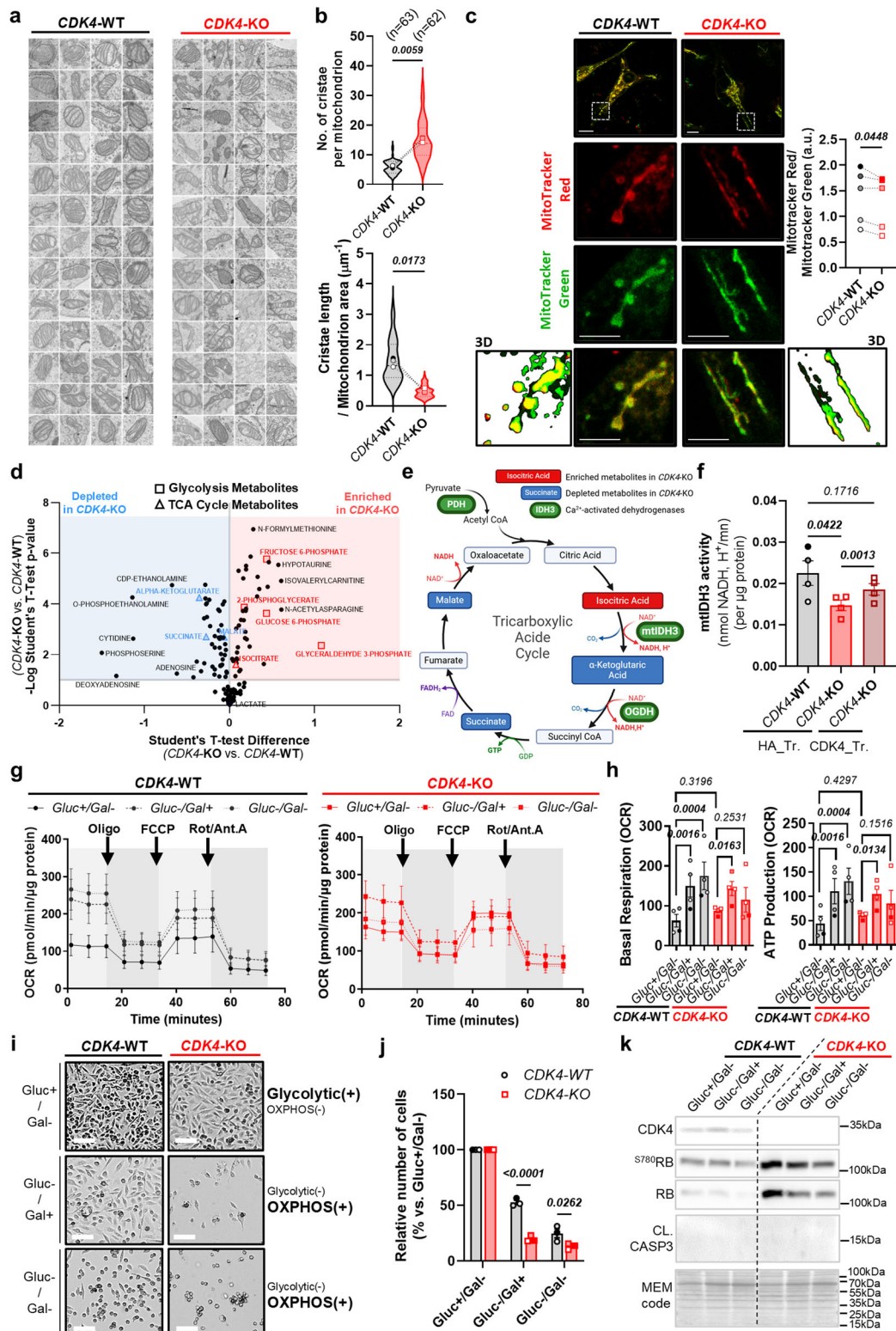

### Multiple pathway targeted metabolite analysis

Cells were pre-extracted and homogenized by the addition of 1000 μL of MeOH:H₂O (4:1), in the Cryolys Precellys 24 sample Homogenizer (2 ×20 sec at 9500 g, Bertin Technologies, Rockville, MD, US) with ceramic beads. The bead beater was air-cooled down at a flow rate of 110 L/min at 6 bar. Homogenized extracts were centrifuged for 15 min at 4000 g at 4 °C (Hermle, Gosheim, Germany). The resulting supernatant was collected and evaporated to dryness in a vacuum concentrator (LabConco, Missouri, US). Dried sample extracts were resuspended in MeOH:H₂O (4:1, v/v) according to the total protein content evaluated using BCA Protein Assay Kit (Thermo Scientific, Masschusetts, US) Cell extracts were then analyzed by Hydrophilic Interaction Liquid Chromatography coupled to tandem mass spectrometry (HILIC - MS/MS) as previously described by Gallart-Ayala and van der Velpen[74,75]. In brief, to maximize the metabolome coverage, the data were acquired in positive and in negative ionization modes using a 6495 triple quadrupole system (QqQ) interfaced with 1290 UHPLC system (Agilent Technologies).

**Fig. 10 | CDK4 promotes mitochondrial fitness and metabolic flexibility through balanced calcium signaling. a** Representative electron micrographs of mitochondria cristae from CDK4-WT and -KO TNBC cells. N = 3 independent biological replicates. **b** Quantification of the number of cristae per mitochondria and cristae length/mitochondria area from CDK4-WT and -KO TNBC cells. Means of N = 3 independent biological replicates from a total of n mitochondria, represented with violin plot. Two-sided Paired T-tests. **c** Representative micrographs of CDK4-WT and -KO TNBC cells stained with MitoTracker Red and Green. Scale bars: 10 μm. Quantification of MitoTracker Red/Green fluorescence intensities in CDK4-WT and -KO TNBC cells. N = 5 independent biological replicates representing a total of n = 145 cells (CDK4-WT) or 157 cells (CDK4-KO). Two-sided Paired T-tests. **d** Metabolomics analysis from multi-pathway analysis from CDK4-WT and -KO TNBC cells. N = 5 biological replicates. **e** TCA cycle scheme with annotated metabolites. Depleted and enriched (blue and red) TCA metabolites in CDK4-KO cells (p-value < 0.05), and calcium-dependent enzymes (green), are displayed. N = 5 biological replicates. Multiple Student's t-test. **f** Mitochondrial isocitrate-dehydrogenase 3 (mtIDH3) activity normalized by protein quantity and measured in CDK4-WT, CDK4-KO TNBC cells transfected with empty plasmid HA (HA_Tr.) and CDK4-KO TNBC cells expressing endogenous CDK4 (CDK4_Tr.). Mean +/-SEM of N = 4 biological replicates. RM 1 way ANOVA; Tukey's multiple comparisons test. **g** Seahorse curves of oxygen consumption rates (OCR) from CDK4-WT and -KO TNBC cells, in media containing Glucose (10 mM)/No galactose (Gluc + /Gal-), No glucose/Galactose (10 mM) (Gluc-/Gal + ), or No glucose/No galactose (Gluc-/Gal-). Mean +/-SEM of N = 4 independent biological replicates. **h** Seahorse quantitative analysis of oxygen consumption rate (OCR). Basal OCR (resting conditions) and ATP-linked production OCR (upon Oligomycin treatment). Mean +/-SEM of N = 4 independent biological replicates. 2-way ANOVA; Tukey's multiple comparison tests. **i, j** Representative pictures and relative number of CDK4-WT and -KO TNBC cells 72 h after continuous with Gluc + /Gal, Gluc-/Gal + , Gluc-/Gal- media. Scale bars: 10 μm. 2-way ANOVA; Sidák's multiple comparison tests. Mean +/-SEM of N = 3 independent biological replicates. **k** Immunoblots of CDK4, $^{S780}$RB, RB, Cleaved Caspase 3 (CL.CASP 3), and MEM code of CDK4-WT and -KO TNBC cells cultured for 48 h with Gluc+/Gal, Gluc-/Gal + , Gluc-/Gal- media. Representative of N = 2 biological replicates. Exact *p-values* are displayed in italic (bold italic if <0.05).

## IDH activity assay

The IDH Activity Assay Kit (Signa-Aldrich, Ref:MAK062) was used to assess of mitochondrial IDH3 activity. Two days before the assay, one million cells per dish were plated in 10 cm dishes. On the day of the assay, the cells were harvested by scraping with 500 μL of IDH Assay Buffer, and the mixture was centrifuged for 10 min at 16000 g and 4 °C. Ten microliters of the supernatant was then used to evaluate IDH3 activity according to the manufacturer's recommendations. Only the cofactor NAD + , preferentially used by mitochondrial IDH3, was added to the reaction buffer mix to distinguish mitochondrial IDH3 activity from the activity of cytosolic and peroxisomal IDH1 and IDH2.

## Mitochondrial bioenergetics

Cells were cultured in XF96cell culture microplates (Agilent Seahorse) at a density of 20'000 cells/well the day prior the assay. The utility plate was hydrated with sterile water overnight in a non-CO$_2$ incubator. Water was replaced by Calibrant XF medium for 1 h in non-CO$_2$ incubator. Prior the assay, cells were washed 2 times and incubated with XF assay medium (RPMI 1145 10X, containing 2 mM of glutamine (ThermoFisher Scientific), 1 mM of pyruvate, 1 mg/L of folic acid, 1 mM of HEPES (Gibco) and pH adjusted to 7.4) for 1 h in non-CO$_2$ incubator. Glucose 20 mM (Glu+/Gal-), Galactose 10 mM (Glu-/Gal+) or no supplementation (Glu-/Gal-) were used for the different assays. Mitochondria stress assay was performed using Oligomycin (1.5 μM), for carbonyl cyanide 4-trifluoromethoxyphenylhydrazon (FCCP) (1 μM) and for Rotenone and Antimycin A (1 μM) (Sigma-Aldrich). OCR and ECAR measurement were normalized to protein content per well.

## Clinical data analysis

*Data pre-processing.* SCAN-B and TCGA bulk RNA-sequencing (RNA-seq) datasets were downloaded and pre-processed as previously described[76]. *Data normalization and signature scores.* All the 3 datasets were normalized using EMBER's procedure with the ember R package[76]. Shortly, for each sample individually, genes are ranked from lowest to highest expression, rankings are divided by an average ranking of stable genes and each sample is embedded in the EMBER space. The embedding removes batch effects and allows direct comparison of signature scores across datasets. The gene signatures scores were calculated by summing up the normalized gene expression levels after applying EMBER's procedure. The Molecular Signature Database (MSigDB)[77] was used for standard molecular signatures. Survival analysis: Standardised survival probabilities[78] were calculated using the R package flexsurv (v2.3[79]). The median of either the genes or the gene signatures was used as a threshold to dichotomize the variables and standardised survival probabilities were adjusted by age and tumor stage. Patients with triple-negative breast cancer (TNBC) that received chemotherapy only were considered for the analysis. *Comparison of gene signature scores.* The gene signature scores between pre and post NAC treatments were compared using the stan_glmer function from the rstanarm R package (v2.21.3)[80]. A hierarchical model was calculated with default weak priors. The formula used was "gene signature ~ timepoint * pCR status + (1 | patient ID)", where gene signature corresponds to the score for the respective signature, timepoint is either pre or post NAC, pCR status is the pathologic complete response and patient ID corresponds to the ID of a specific patient. Hierarchical models were used since the measurements between pre- and post-NAC are not independent. In total 4 chains with 4000 iterations each were calculated. Linear predictors of the hierarchical model were extracted using the function *add_linpred_draw* from the package tidybayes (v3.0.2).

## Data representation and statistical analysis

In bar plots, individual values are presented as the mean ± SEM of N independent biological replicates for in vitro experiments or of N mice for in vivo experiments. Medians are used for dataset not following normal distributions according to figure legends. In whisker and violin plots, n indicates the number of mitochondria, MERCs, or cells, as specified in figure legends. The Shapiro–Wilk normality test was applied to raw data before proceeding to any analysis. Further statistical analyses and tests are indicated in figure legends. For comparisons between two groups of non-normally distributed data, the nonparametric Mann–Whitney test was performed. For comparisons between two groups of normally distributed data, the following two-sided parametric tests were performed: Student's t test (equal variance) and Welch's t test (nonequal variance). Paired t-test was also performed to avoid any batch effect and remove the variability of each biological independent replicate. For comparisons among more than two groups, repeated measures (RM) (for paired data) or ordinary (for unpaired data) one-way analysis of variance (ANOVA) was used, and Tukey's multiple comparisons test (with a single pooled variance) was subsequently performed. For data from in vivo experiments, two-sided Grubbs' test was performed to find outliers, which were removed from further analysis if the *p* value was less than 0.05. All statistical analyses were performed using GraphPad Prism 9.1.0 software (ns: not significant; *p < 0.05; **p < 0.01; ***p < 0.001; ****p < 0.0001). BioRender and Inkscape softwares were used to draw Fig. 2h (Created in BioRender. Parashar, K. (2025) https://BioRender.com/z50s631) and Fig. 10e (Created in BioRender. Parashar, K. (2025) https://BioRender.com/m54b855).

**Reporting summary**

Further information on research design is available in the Nature Portfolio Reporting Summary linked to this article.

## Data availability

Source data are provided with this paper. Generated datasets are publicly accessible as following: RNA-seq dataset (GSE278524) (https://www.ncbi.nlm.nih.gov/geo/query/acc.cgi?acc=GSE278524), proteomic and phosphoproteomic datasets (PXD046326, PXD046327 and PXD046353 on ProteomeXchange Consortium via the PRIDE partner repository[69]) (respectively https://proteomecentral.proteomexchange.org/cgi/GetDataset?ID=PXD046326, https://proteomecentral.proteomexchange.org/cgi/GetDataset?ID=PXD046327, https://proteomecentral.proteomexchange.org/cgi/GetDataset?ID=PXD046353). The neoadjuvant chemotherapies (NAC) and SCAN-B RNA-sequencing were downloaded from GEO using the accession codes GSE123845[41] and GSE60789 [https://www.ncbi.nlm.nih.gov/geo/query/acc.cgi[39] respectively. TCGA bulk RNA-sequencing (RNA-seq) datasets were downloaded from European Genome-Phenome Archive (http://www.ebi.ac.uk/ega/), under accession number EGAS00000000083[40]. Any remaining data will be available from the corresponding author if needed. Source data are provided with this paper.

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

## Acknowledgements

The authors would like to thank the lab members of Prof. Lluis Fajas group for the helpful discussions, Manfredo Quadroni and the Proteomic Analysis Facility (PAF), Arnaud Paradis and the Cellular Imaging Facility (CIF), the Genomic Technologies Facility (GTF), the technical staff from the Metabolomics Platform (MEP), L. Battista (EPFL) and Eric Aria-Fernandez for technical assistance, Jean Daraspe and Antonio Mucciolo from Electron Microscopy Facility (EMF), the FBM animal facility platform, and the Scientific Service at the Center for Integrative Genomics (SSC). This work was supported by the University of Lausanne. J.L-A. was supported by a FPI fellowship (FPI17/BES-2017-081354) from the Ministerio de Ciencia e Innovación (Spain), and a Short-Term Fellowship (8855) from the European Molecular Biology Organization (EMBO). K.H. was supported by the Austrian Science Fund (FWF) grant J4597-B. G. Sflomos was supported by the ISREC Foundation and the Innovative Medicines Initiative Joint Undertaking (grant agreement no. 115188) for the PREDECT consortium. We acknowledge the Swiss National Science Foundation grant (31003A_143369 and 310030_207688) to L.F. group.

## Author contributions

D.V.Z. and L.F. conceived and designed the study. D.V.Z. and K.P. performed and analyzed most of in vitro experiments. LM-C generated the CRISPR-Cas9 cell line. L.L.-E. performed kinome experiments. L.L.-E. and L.F. analyzed kinome experiments. DVZ, WC, KH, MI and X-PB performed in vivo experiments. D.V.Z., N.Z., and Y.G. performed calcium imaging. J.L.-A. and MMM performed mitochondrial cristae analysis and 3D confocal image analysis. C.R. performed tissue processing and histological immunofluorescence. M.-A.B. and J.R. performed MERCs purification. G.P. performed PLA experiments. H.G.-A. and J.I. supervised the acquisition and analyzed the metabolomics data. G.S. and C.B. generated and cultured NST breast cancer patient-derived cells. C.R. performed bioinformatic analyses on clinical datasets. D.V.Z. and L.F. wrote the manuscript. All authors approved the final version of the article.

## Competing interests

The authors declare no competing interests.
