## [Peer Review file · Nature Communications]

CDK4 Inactivation Hinders Cell Death and Metabolism Remodeling Mitochondria-ER contacts in Triple-Negative Breast cancer cells

Corresponding Author: Professor LLuis Fajas

Version 1:

Reviewer comments:

Reviewer #1

(Remarks to the Author)

This manuscript by Ziegler et al. identifies a new mechanism in which CDK4 is able to regulate MERCs and thereby regulate mitochondrial fission and calcium levels (and metabolism, downstream). They link this regulation of mitochondria to the surprising finding that CDK4 inhibition does not lead to increased cell death in TNBC treatment, instead it conveys resistance to therapy. The conclusion is that by inhibiting (or knocking out) CDK4, mitochondria are reprogrammed to be more resistant to apoptotic cues.

I would like to recognize the authors for this nice study. The experiments and analyses done are generally of good quality and clear, and I think the story is exciting and will certainly be of interest to the readers of Nature Communications. Please find my comments below, with points of improvement to try and ensure the interpretation matches the performed experiments.

Major points:

1) In figure 4, the authors show well-designed experiments to try and quantify ER to Mitochondria Calcium transport. For this they used a variety of calcium probes (Rhod-2AM and Fluo-4). However, these probes are both ratiometric, and because of this the starting point is 1. This gives the impression that starting Calcium levels in the ER and Mitochondria are equal in both WT and KO cells. This could well be the case but to reinforce the interpretation that lower mitochondrial calcium entry after TG and Histamine is due to reduced ER-Mitochondrial Ca transfer in KO cells, the authors should try to quantify absolute calcium concentrations. Starting mitochondrial calcium levels in KO cells could be much lower or higher than in WT, and the difference in Calcium entry may thus be due to other effects. This should be checked empirically as calcium levels in mitochondria are such a key metric for this paper. This can be done using Aequorin (please see PMID: 28382319) for example.

As a small aside, reading the M&M, it is not entirely clear that the actual measurement was done without extracellular Calcium, it just mentions cells were washed with Calcium free solution. Please confirm that the measurements were indeed done without extracellular Calcium, this should be made more apparent in the M&M. In addition, it can be mentioned that one of the reasons this was done was to exclude the contribution of SOCE to mitochondrial calcium uptake.

2) I appreciate the detailed explanation of the statistical approaches in the M&M, especially the checks for normality. However, I was then surprised to see the appearance of paired t-tests in a situation where I would think this is not the correct statistical test, most notably throughout figure 4. Is a statistical test on the actual trace not possible instead? Paired t-tests are generally done when one is changing an aspect of the same subject, which in these experiments is not the case (comparing WT vs KO). Why was this test chosen for just these measurements?

This manuscript handles statistics and graph representation quite well overall, especially compared to other papers. However, I feel that it would help if the authors had a look at this paper (PMID: 32346721). I think implementing some of those reporting methods would benefit the interpretation of the graphs (also have a think about SEM versus SD). Especially the use of the violin plots, taking # of mitochondria as N is a little questionable to me.

3) The EM experiments in figure 5 look convincing at first reading. However, it is not clear to me how these experiments were done. The gold standard for quantifying ER-Mitochondria contact sites is performing a whole cell FIB-SEM analysis.

However, such an experiment is technically challenging, time consuming and expensive, and as such I don't believe this is required for this manuscript. Having said that, the details of the analysis performed in this manuscript must be spelled out. Did the authors delineate and count every single mitochondrion in their EM micrographs? Which mitochondria were imaged? How was it decided which ones to image? From equivalent cross sections of the cells? As I can't judge what the current method of analysis was, I would suggest the authors have a look at the methods described in these papers (PMID: 28108524 and 36952181) to ensure a robust quantification of ER-Mitochondria contact sites in both cell types.

4) The presence of both PKA and CDK4 at MERCs (figure 6F) is intriguing. Both are cytosolic kinases. The mechanism for the mitochondrial localization of PKA has been hinted at and the authors indeed detect the AKAP's in their MERC fraction. However, the presence of CDK4 is less explained. A previous study (PMID: 25578653) has interpreted their CDK4 mitochondrial localization data as CDK4 being transported inside mitochondria. Both these observations should be expanded on to make sure the interpretation of CDK4 and PKA at MERC's in this manuscript is correct. This claim would be much more convincing by being able to image CDK4 and PKA at these contact sites (and not inside mitochondria) with high resolution fluorescence microscopy, or techniques like FRET/FLIM. Immunogold EM to show both proteins at these contact sites (as done for PMID 27113756, figure 1C) would be even better, but I understand this may be technically challenging. I leave it up to the authors to decide what will be most straightforward to do for them.

As an alternative, if these approaches seem too much work, the manuscript should be rewritten in a way to make the claim that all this signalling happens at MERC's less strong, as "just" the western blot of MERC isolation is not enough to support this. I feel that having at least some additional good fluorescent data to show these proteins at these mitochondrial sites would improve the manuscript though.

5) The authors identify the CDK4-PKA axis as a regulator of MERC's, and clearly show that the phosphorylation activity of CDK4 is required for this. However, I was missing an analysis of actual PKA phosphorylation levels. Is the site where CDK4 phosphorylates PKA known? At the moment the experimental data show when CDK4 is knocked out or unable to function as a kinase, PKA substrate phosphorylation is lower, but this is correlation, not causation. The author's interpretation of the CDK4-PKA axis being the cause would require identification and analysis of PKA phosphorylation levels and showing that mutating the CDK4 site leads to similar phenotypes as the CDK4 knock out. I realize these may not be straightforward experiments, but then the interpretation needs to be toned down or explained as a limitation of the study.

6) Supplementary table 1: I appreciate the author's work to try to distil a list of core MERC proteins. However, some of these proteins are quite suspect as "core" MERC proteins (like mTOR), while at the same time other established MERC proteins are not listed (ie VPS13D). The labels are also a bit shaky. VAPB is rightly labelled as a tethering protein, but its near identical isoform VAPA is listed as "other". Please have a look at this list again, and seeing the pleiotropic functions of some of these proteins (for example eIF2aK3 is involved in ER homeostasis, but also "tethers"), I would even suggest scrapping the subcategories. The later interpretation that CDK4-PKA specifically regulates tethers seems too forced due to the lack of robustness and seeming randomness in assigning these categories.

7) This is up to the authors, but I would suggest changing the title of this study to reflect the link to MERC's and a functional role in regulating them. I feel the current title does not do justice to some of the major findings of the data.

Some more minor points:

1) Supplementary figure 5A: Please show some representative micrographs of the xenograft tumor EM data.

2) Supplementary figure 5C: It is not clear how this graph is calculated and what is being measured. Is this a Mander's or Pearson coefficient? It should be mentioned in the legend or the M&M. Regardless of this I find the fluorescent images to seem overexposed and not very informative. It would help if there could be more representative cells + each cell shown bigger.

3) Figure 6B: It is not clear to me how the authors blot for "PKA substrates" as the antibody or method is not described.

4) Figure 6C: No statistics shown even though N=3

5) Supplementary figure 6C: The graph shows a lot of ITPR3 hits being enriched in CDK4KO, yet the text states they are all downregulated?

6) In the text it is stated on page 9 "... which proved a direct effect of CDK4." The word proved is dangerous in biology, this should be changed to something less absolute.

7) Figure 7C: The images of the mitotracker green/red cells are not representative of the graph. I understand the authors want to indicate the difference visually, but this is a bit excessive. In addition, I wonder if using flow cytometry may not be a better way to quantify this? Could the authors provide some examples here of other studies that use this method? In addition to this, the manner of data analysis for this experiment is not described, neither in the text nor in the M&M.

Reviewer #2

(Remarks to the Author)

The authors investigate how the loss or inhibition of CDK4 leads to resistance in triple negative breast cancer cells. This is of course an important question clinically and the underlying molecular mechanisms have not been identified. The results indicate that CDK4 affects or regulates mitochondria-ER interactions and this may be mediated by PKA.

Overall, this manuscript contains a large amount of data and most experiments have been done well with the appropriate controls. There are just a few points that need to be improved in order to make the results more robust.

This work reports important results and with appropriate revisions, this manuscript would be a good contribution to Nature Communications.

Here are the issues that need to be addressed:

1. Figure 1; there are questions regarding the phosphorylation of S780 in RB. In the CDK4-KO cells, total RB is much higher than in the controls and S780 is also increased, nevertheless the ratio is decreased. In the Abema treated cells, both total RB levels and S780 levels are decreased compared to Veh but the ratio is only decreased at D3. In addition, both CDK4 and CDK6 levels are increased in Abema treated cells. All of this is a little confusion and one has to wonder what is the level of CDK2 activity? In addition, this reviewer wonders if the right experiment would be to treat CDK4-KO cells with Abema....
2. Figure 2C; similar than the first question; why is S780 increased across the board including in Veh? Also, this data should be quantified. It is also contrasting with 2F, which needs explanation. Could this be due to CDK2 activity?
3. Figure 2G; there is an issue with the D14;CDK4-KO / Veh because the data is spread widely, especially compared to all the other points (black/blue/green). Therefore, the red data points cannot be trusted and the average is misleading. This of course weakens the point the authors are trying to make.
4. Figure 3; did the authors measure mitochondrial DNA content as an alternative determination of the number or size of the mitochondria?
5. Page 8; "These results suggest that CDK4 is required for ER-MT calcium signaling in TNBC cells." This should be rephrased since CDK4 could either directly or indirectly be involved in this and the authors do not want to mislead the readers.
6. Page 10; "...PKA regulatory subunit PRKAR1A in MERCs (Fig. 6F) and..." This is not shown in 6F and therefore maybe it is "Sup. Fig. 6F"?
7. Figure 6G is hard to read. What do the authors want to show here?
8. From all the data, the part that is the weakest is to show that the interactions of the mitochondria to the ER are altered. Any data the authors could add would be beneficial...maybe they can get some inspiration from <https://doi.org/10.1038/s41586-023-06956-y>

Reviewer #3

(Remarks to the Author)

In the manuscript by Zeigler et al., a comparison of CDK4-WT vs CDK4-KO MDA-MD-231 is performed, under the conclusion that CDK4 is not required for proliferation, therefore, cell phenotypes in CDK4-KO cells are independent of proliferation changes. Overall, the relationships between chemosensitivity and metabolism are important, and the context of these pathways through MERC has the potential to reveal both physiological and patho-physiological contexts for organelle tethering. Within the manuscript, there are many observations and pathways that are interrogated, yet this broad approach fails to return to the primary objective of the manuscript and appears to be highly unfocused with copious observations that are weakly linked between the figures and not related to the disease model system.

Major concerns:

A single parental cell line is used in the study, and no clinical data/samples are investigated to support. Also, the metastatic cells don't express ER, PR, or E-cad, and harbor mutant p53, so the consequences of CDK4 deletion need to be positioned in this context, and perhaps investigated without all the confounding mutations.

The authors show that CDK4-WT and CDK4-KO grow similarly in culture, but vastly different in xenografts within a partial microenvironment (e.g. macrophage, dendritic cells are likely present). Given this observation, it appears unsubstantiated to disregard the proliferation changes in vivo without detailed interrogation as they indeed demonstrate a lack of proliferation for 3 weeks. For example, does macrophage depletion eliminate the three week lag? Also, if the CDK4-KO cells metastasize immediately after implantation, the primary site would be expected to grow slower; this is also not addressed.

The cell death studies employ some unusual inducers, for example oligomycin and antimycin without rationale (also, why treat TNBC with peroxide?). As the mitochondrial pathway is of interest to the authors, some demonstration beyond cleaved caspase-3 (which cleaves either mechanistically or consequentially given a variety of cell death pathways) is necessary.

Does elimination of BAK/BAX block death, does zVAD-fmk delay death, etc. Figure 2 shows that 40% of CDK4-KO cells are dead, yet no C3 cleavage; same with Cisplatin, most CDK4-KO cells are dead in figure 2b, yet the authors don't provide explanation.

No evidence of equal stress within the cells are provided; for example, does cisplatin cause the same number of DNA lesions independent of genotype.

Many of the cell death studies also disregard more modern mechanisms of the intrinsic pathway and suggest that calcium signaling is essential for apoptosis, yet the mitochondrial pathway and PTP are not mechanistically linked.

The immunofluorescence for mitochondria could benefit with clearer markers/image capture, as data in figure 3D and S3K are blurry, and the images don't match the quantification of mitochondria presented in figure S3...these cells have more than 25 mitochondria, on average.

The calcium studies are interesting, but not presented in a context of the apoptosis or metabolism.

ER Tracker is not a reliable marker to perform co-localization studies as it is not perfectly localized...this is demonstrated in figure S5C, as the ER signal is throughout the majority of the cells. A different marker or imaging capture technique (Imaris,

STED, etc) is necessary to establish conclusions.

The metabolism studies could benefit from bioenergetics studies to report oxygen consumption, OCR, ECAR, mitochondrial ATP generation, etc. As presented, the metabolism work is cataloging changes with no return to the focus of the manuscript on heightened metabolic sensitivity (although it is not clear what this means or if it's observed in TNBC).

Reviewer #4

(Remarks to the Author)

In the manuscript "CDK4 inactivation balances resistance to apoptosis with heightened metabolic activity in triple negative breast cancer" Ziegler and colleagues aim to understand how CDK4 regulates the fate of TNBC tumor cell lines and cell line xenografts. They identify that using CDK4 genetic knock out results in reduced apoptosis and does not inhibit xenograft growth because CDK4 enhances mitochondria-ER contact and alters mitochondrial calcium flux and apoptosis inhibition.

This paper answers an important question, which is why CDK4/6 inhibition, so clinically impactful in ER positive breast cancer, fails in ER negative breast cancer. They used a wide array of experimental and analytic approaches to reach this conclusion. They used drug treatment with cisplatin and metabolic stress to show that CDK4 KO protects MDA-MB 231 cells from apoptosis. They found decreased calcium flux and found decreased MERCs using transmission EM. Proteomic analysis demonstrated that MERC tether proteins were downregulated. This effect appears dependent on PKA alpha, mediated through effects on calcium related channels.

Overall, the main conclusion of the paper i.e. CDK4 KO results in apoptosis resistance via increased MERC formation and alterations in ER_MT calcium signaling is well supported by the data. However, prior to publication, certain clarifications could help improve the overall story.

- What is the evidence that CDK4 directly phosphorylates PKA alpha? The authors have shown that CDK4 activity is necessary and sufficient for PKA phosphorylation but is this direct or through an intermediary kinase? This detail would be important to determine that the CDK4 KO effect is specifically related to loss of CDK4 activity alone?

- How does CDK4 KO affect the levels of mitochondrial proteins involved in apoptosis regulation as BIM, BAX, BCL2 and PUMA? The authors show that there is decreased mitochondrial membrane potential in CDK4 KO cells. Does this result in reduced outer mitochondrial membrane permeabilization as a means of reducing likelihood of apoptosis? These additional data would help better understand why CDK4 KO alters apoptosis thresholds in addition to altering calcium flux? See <https://www.ncbi.nlm.nih.gov/pmc/articles/PMC7325165/> for more details on this functional approach.

- The authors show that CDK4 KO alters isocitrate, alpha-ketoglutarate, malate and succinate levels and that galactose reduces CDK4-KO cell viability. This is an intriguing observation but not well fleshed out and seems to detract from the overall flow of the paper. What is the impact of altered metabolism on apoptosis vulnerability might be one way to tie these strands together.

Version 2:

Reviewer comments:

Reviewer #1

(Remarks to the Author)

I have gone through the author's responses, and overall, it is clear the authors have approached the comments in a serious and systematic way, which I appreciate. A lot of extra work has been done that has resolved most of my comments. However, even though I am not a fan of multiple rounds of revision, there are some issues that I feel still need to be addressed.

Fig 3 K – L : There seems to be a discrepancy in the differences between the graphs, where 3K shows quite some variability in the WT, this is almost gone in 3L. Are these experiments done independently from each other? Why?

Sup. Fig 5a: I understand that the authors took the probe used to measure absolute calcium levels and subjected cells expressing the probe to histamine. However, this figure panel is confusing and weakens the message. It is not referenced in the text anywhere, and seems a (less convincing) copy of fig 5F. Furthermore, it seems to contradict the result from figure 5a, where 5A shows a difference in basal calcium, fig S5A now shows an identical starting value. How can this be?

Sup. Fig 5i-k : "Finally, combined knockdown of ITPR3, VDAC1 and/or MCU was sufficient to rescue the partial cell death induced by H₂O₂ and O₂A, but not cisplatin in CDK4- WT cells (Sup. Fig 5i-k). "

This statement seems too strong when looking at the figure. The data show a very minor increase in cell death, this should be reflected in the text, as the knockdowns are certainly not sufficient to rescue the effect.

Comment on statistics: I recognize that the authors prefer to use a paired t-test but I have to disagree with their reasoning. It is not that a paired t-test is "typically" used for measurements of the same individual, it is that that is a necessary assumption. I feel the author's pain in having to deal with "unnecessary" variability due to cell competency, age etc, but this is not an

excuse to use the wrong statistical test. I suggest the authors look into using non-linear mixed model statistics, which is able to deal with this additional variability, or use unpaired t-tests/ANOVA's.

In addition, why were some of the tests changed to a paired t-test, like in figure 6b,c? The paired t-test in that experiment is also not permissible in my view.

As a side comment: I would like to recognize the author's honesty and scientific integrity in displaying openly when a statistical test shows a borderline p value (ie 0.0507 or so) instead of trying to massage the data.

Figure S6d: I appreciate the extra work done by the authors, and the methodology seems ok to me now. However, I am not really satisfied with the staining. Not only does the Calreticulin seem too punctate, there is a marked difference between WT and KO in staining intensity. Is this a representable difference? Staining intensity may affect Mander's independent of any contact site difference. Please have another look at this and carefully evaluate your ER staining and any differences between WT and KO.

Minor comments:

Graphical abstract: I like the graphical abstract, but the arrows and info that arise below the ER-Mito drawing is not clear enough I think. Please have a look at improving the clarity.

This is probably a personal thing but I still find the word "proven", used throughout the text just too strong. It's also unnecessary as other words like show or indicate etc are fine.

Title: I appreciate the authors have taken up the suggestion of changing the title, but the new title seems to be missing some info. Perhaps the authors meant: CDK4 Inactivation Hinders Cell Death and Metabolism through Mitochondria-ER contact site remodelling in Triple-Negative Breast cancer cells ?

Reviewer #2

(Remarks to the Author)

The authors have done extensive revisions and have improved their manuscript.

I believe that Figure 9 (mostly added in response to a reviewer comment) should be moved to supplemental data since it is much less convincing than the other data.

Other than that, it would be nice if the authors had provided an overview over all the changes that were made in order to make it easier for the reviewers.

Reviewer #3

(Remarks to the Author)

The authors have addressed the vast majority of concerns raised during the review process, along with reorganizing the text and data presentation for increased clarity.

Reviewer #4

(Remarks to the Author)

The authors have adequately addressed my concerns

Version 3:

Reviewer comments:

Reviewer #1

(Remarks to the Author)

I would like to acknowledge the authors' hard work and detailed responses to address all my concerns. I congratulate the authors on this nice study and I fully support publication.

Response to reviewers

Reviewer #1

This manuscript by Ziegler et al. identifies a new mechanism in which CDK4 is able to regulate MERCs and thereby regulate mitochondrial fission and calcium levels (and metabolism, downstream). They link this regulation of mitochondria to the surprising finding that CDK4 inhibition does not lead to increased cell death in TNBC treatment, instead it conveys resistance to therapy. The conclusion is that by inhibiting (or knocking out) CDK4, mitochondria are reprogrammed to be more resistant to apoptotic cues.

I would like to recognize the authors for this nice study. The experiments and analyses done are generally of good quality and clear, and I think the story is exciting and will certainly be of interest to the readers of Nature Communications.

Please find my comments below, with points of improvement to try and ensure the interpretation matches the performed experiments.

Major points:

1) In figure 4, the authors show well-designed experiments to try and quantify ER to Mitochondria Calcium transport. For this they used a variety of calcium probes (Rhod-2AM and Fluo-4). However, these probes are both ratiometric, and because of this the starting point is 1. This gives the impression that starting Calcium levels in the ER and Mitochondria are equal in both WT and KO cells. This could well be the case but to reinforce the interpretation that lower mitochondrial calcium entry after TG and Histamine is due to reduced ER-Mitochondrial Ca transfer in KO cells, the authors should try to quantify absolute calcium concentrations. Starting mitochondrial calcium levels in KO cells could be much lower or higher than in WT, and the difference in Calcium entry may thus be due to other effects. This should be checked empirically as calcium levels in mitochondria are such a key metric for this paper. This can be done using Aequorin (please see PMID: 28382319) for example.

We thank the reviewer for this important comment. We quantified absolute calcium concentrations using a FRET-based Ca²⁺ sensors 4mtD3cpv (PMID: 16720273) in a zero calcium-containing buffer. Cells were infected with adenovirus encoding this sensor. We show now that basal mitochondrial calcium is indeed slightly higher in CDK4-KO cells in basal conditions, likely because the increased expression of some of the main calcium channels ITPR1, ITPR3 or VDAC1. Furthermore, we also tested the response of these cells to Histamine, and we found that the maximum peak (not the AUC) was significantly decreased in CDK4-KO, compared to -WT cells. Altogether, these results explain some of our protein expression results and confirm the tendency to have less immediate ER-MT calcium flux signaling, because of the decreased number of MERCs.

As a small aside, reading the M&M, it is not entirely clear that the actual measurement was done without extracellular Calcium, it just mentions cells were washed with Calcium free solution. Please confirm that the measurements were indeed done without extracellular Calcium, this should be made more apparent in the M&M. In addition, it can be mentioned that one of the reasons this was done was to exclude the contribution of SOCE to mitochondrial calcium uptake.

We are sorry for the lack of clarity in the M&M section. Indeed, all the presented measurements were performed without extracellular calcium. Following reviewer's suggestions, we also mention now that this was done to exclude the contribution of SOCE to mitochondrial calcium uptake. The sentence mentioning cells washing in calcium free solution in the material and methods section was completed as following p25: "The cells were then washed twice with calcium-free Krebs solution and all subsequent mitochondrial calcium measurements were performed in a zero calcium-containing buffer to exclude any contribution of SOCE to mitochondrial calcium uptake."

2) I appreciate the detailed explanation of the statistical approaches in the M&M, especially the checks for normality. However, I was then surprised to see the appearance of paired t-tests in a situation where I would think this is not the correct statistical test, most notably throughout figure 4. Is a statistical test on the actual trace not possible instead? Paired t-tests are generally done when one is changing an aspect of the same subject, which in these experiments is not the case (comparing WT vs KO). Why was this test chosen for just these measurements?

We thank the reviewer for this feedback on our statistical methods. First of all, the reviewer is correct in noting that paired t-tests are typically used for related samples where the same subjects are measured under different conditions. We decided, however, to have one mean point per individual biological replicate. Then we chose a paired t-test to avoid any batch effect and remove the variability of each biological independent replicate. To further address this question, we have re-analyzed the data of old Figure 4 of each individual biological replicates, using a TWO-WAY ANOVA test, which confirmed this variability across biological replicates independently of the condition probably due to cells competency, probe assimilation by cells or component (histamine/thapsigargin) efficiency during time. To clarify this point, the revised manuscript includes now a new sentence in the M&M section as following p30: "Paired t-test was also performed to avoid any batch effect and remove the variability of each biological independent replicate."

This manuscript handles statistics and graph representation quite well overall, especially compared to other papers. However, I feel that it would help if the authors had a look at this paper (PMID: 32346721). I think implementing some of those reporting methods would benefit the interpretation of the graphs (also have a think about SEM versus SD). Especially the use of the violin plots, taking # of mitochondria as N is a little questionable to me.

Based on the recommendation of this reviewer, we implemented the methods previously described (PMID: 32346721), and we corrected all the graph representation. Specially, we explained now in each figure legends what violin plots takes into account n number of elements (mitochondria or cells). N and dots represent now in the whole manuscript only independent biological replicates. According to figure legends, N represents now only biological independent replicates that are now taken into account to do more robust statistical analysis.

3) The EM experiments in figure 5 look convincing at first reading. However, it is not clear to me how these experiments were done. The gold standard for quantifying ER-Mitochondria contact sites is performing a whole cell FIB-SEM analysis. However, such an experiment is technically challenging, time consuming and expensive, and as such I don't believe this is required for this manuscript. Having said that, the details of the analysis performed in this manuscript must be spelled out. Did the authors delineate and count every single mitochondrion in their EM micrographs? Which mitochondria were imaged? How was it decided which ones to image? From equivalent cross sections of the cells? As I can't judge what the current method of analysis was, I would suggest the authors have a look at the methods described in these papers (PMID: 28108524 and 36952181) to ensure a robust quantification of ER-Mitochondria contact sites in both cell types.

We apologize for the lack of clarity for measuring ER-mitochondria contact sites in electron micrograph. We agree with the reviewer that FIB-SEM whole cell analysis is challenging and out of our expertise. Following reviewer's suggestion, and to facilitate the reading of the manuscript, we added more details in the M&M section concerning the analysis that we performed. We used a macro on Fiji described in DOI: 10.1016/j.bj.2016.11.735 and kindly provided by Pr. Gyorgy Hajnoczky, we delineated manually both ER and mitochondria (n=8 to 12 per cell), and whether the contact was a real one (<50nm) and also some parameters including the % of mitochondria attached to ER (addition of all the ER-mitochondrion interfaces length and divided by mitochondrion perimeter), but also the mean and the maximal ER-mitochondria distance.

We added the following sentence to M&M section in Electron Microscopy subsection.

4) The presence of both PKA and CDK4 at MERCs (figure 6F) is intriguing. Both are cytosolic kinases. The mechanism for the mitochondrial localization of PKA has been hinted at and the authors indeed detect the AKAP's in their MERC fraction. However, the presence of CDK4 is less explained. A previous study (PMID: 25578653) has interpreted their CDK4 mitochondrial localization data as CDK4 being transported inside mitochondria. Both these observations should be expanded on to make sure the interpretation of CDK4 and PKA at MERC's in this manuscript is correct. This claim would be much more convincing by being able to image CDK4 and PKA at these contact sites (and not inside mitochondria) with high resolution fluorescence microscopy, or techniques like FRET/FLIM. Immunogold EM to show both proteins at these contact sites (as done for PMID 27113756, figure 1C) would be even better, but I understand this may be technically challenging. I leave it up to the authors to decide what will be most straightforward to do for them. As an alternative, if these approaches seem too much work, the manuscript should be rewritten in a way to make the claim that all this signaling happens at MERC's less strong, as "just" the western blot of MERC isolation is not enough to support this. I feel that having at least some additional good fluorescent data to show these proteins at these mitochondrial sites would improve the manuscript though.

We thank the reviewer for this comment. We added supplemental data to this biochemical fractionation of MERC fraction. Because of a lack of time and/or tools/expertise for high resolution fluorescence microscopy, FRET/FLIM or Immunogold EM, we decided to combine Proximity ligation assay and immunofluorescence.

Unfortunately, we did not success to find a good antibody for CDK4 that was specific in immunofluorescence experiments (six different antibodies, Cdk4 antibody [EPR4513] Abcam ab108355; Cdk4 (DCS-35) Abcam ab31112; Cdk4 (H-22) Santa Cruz Biotechnology sc-601; Cdk4 (C-22) Santa Cruz Biotechnology sc-260; CDK4 Monoclonal Antibody Thermo Fisher Scientific MA5-12984; Cdk4 (D9G3E) Rabbit mAb Cell Signaling Technology 127905; CDK4-Merck- ZoomAb ZRB1588).

To assess the localization of PKA at MERCs we performed proximity ligation assays using a ITPR1-VDAC1 antibody, which labels MERCs, and in parallel another antibody targeting PRKACA, which is the catalytic subunit of the PKA complex. Using this strategy, we proved that PRKACA localized at MERCs in the MDA-MB-231 TNBC cells. This result is now shown in Figure 8c.

5) The authors identify the CDK4-PKA axis as a regulator of MERC's, and clearly show that the phosphorylation activity of CDK4 is required for this. However, I was missing an analysis of actual PKA phosphorylation levels. Is the site where CDK4 phosphorylates PKA known? At the moment the experimental data show when CDK4 is knocked out or unable to function as a kinase, PKA substrate phosphorylation is lower, but this is correlation, not causation. The author's interpretation of the CDK4-PKA axis being the cause would require identification and analysis of PKA phosphorylation levels and showing that mutating the CDK4 site leads to similar phenotypes as the CDK4 knock out. I realize these may not be straightforward experiments, but then the interpretation needs to be toned down or explained as a limitation of the study.

We agree with the reviewer that we did not address, in the first version of the manuscript, the causative effects of the observed decrease in PKA activity in CDK4-KO cells. Phosphoproteomics analysis (Supplementary Figure 8b) showed that the phosphorylation of PKACA, which is the catalytic subunit, was decreased in the CDK4 KO cells, but did not contain any putative CDK4 phosphorylation site. This suggested that PKA itself was not a direct target of CDK4. So, we further tried to identify other PKA-associated putative CDK4-target based on 1°/downregulation of some phosphosites of this target (Supplementary Figure 8b) and 2°/ the presence in this phosphosite of a CDK4-motif. We identified two regulatory subunits of PKA, namely PRKAR1A and PRKAR1B, as two CDK4 putative target on the Serine 83 (Supplementary Figure 7b). We found that CDK4 was preferentially interacting with PRKAR1A instead of PRAKR1B (Supplementary Figure 7c). These new results strongly suggest that CDK4 regulates the activity of PKA through, at least, the phosphorylation of the regulatory subunit PKAR1A.

6) Supplementary table 1: I appreciate the author's work to try to distill a list of core MERC proteins. However, some of these proteins are quite suspect as "core" MERC proteins (like mTOR), while at the same time other established MERC proteins are not listed (ie VPS13D). The labels are also a bit shaky. VAPB is rightly labelled as a tethering protein, but its near identical isoform VAPA is listed as "other". Please have a look at this list again, and seeing the pleiotropic functions of some of these proteins (for example eIF2aK3 is involved in ER homeostasis, but also "tethers"), I would even suggest scrapping the subcategories. The later interpretation that CDK4-PKA specifically regulates tethers seems too forced due to the lack of robustness and seeming randomness in assigning these categories.

We appreciate the reviewer's comment. Our goal was to understand how the significant changes in MERC proteomics specifically impact the CDK4-KO phenotype. Analyzing individual proteins from the all the detected MERC-associated proteins using a volcano plot did not seem meaningful. Therefore, we chose to distill a list of core MERC proteins by clustering them based on their primary known functions. Following this reviewer suggestion, we have now refined the list of MERC-associated proteins, including 21 proteins that were missing (such as VPS13D, but also CALR, MICU1, PINK1, S100A1 or TXNIP). We removed the word "core" MERCs proteins and used only the term "MERC-associated proteins". We also reevaluated through this list the pleiotropic functions of some proteins (such as EIF2AK3) and reformatted the new Supplementary Table 2 with all known functions of each of these 176 MERC-associated proteins.

7) This is up to the authors, but I would suggest changing the title of this study to reflect the link to MERC's and a functional role in regulating them. I feel the current title does not do justice to soe of the major findings of the data.

To better reflect the data shown in this manuscript, as suggested by this reviewer, we changed the title to "***CDK4 Inactivation Hinders Cell Death and Metabolism Remodeling Mitochondria-ER contacts in Triple-Negative Breast cancer cells.***"

Some more minor points:

1) Supplementary figure 5A: Please show some representative micrographs of the xenograft tumor EM data.

Please find some representative micrographs of the tumors developed from grafted cells. We have selected one of each genotype which are now displayed in Supplementary Figure 6c.

AM437-CGK-029267-Box102-D1-TUMEUR-Ziegler-Montage1

Zoom 16,7% Cellule 2 TUMOR 1 - 29267

WT-TUMORS

Zoom 75%

AM-20221223-AM438-Box102-E4-CKG-029269-Montage2

Zoom 16,7% Cellule 6 TUMOR 2 - 29269

WT-TUMORS

Zoom 75%

AM-20231011-AM450-A1-Box115-Montage04-Ziegler

Zoom 16,7% Cellule 3 TUMOR 1 - 29279

KO-TUMORS

Zoom 75%

Zoom 16,7% Cellule 6 TUMOR 2 - 29281

KO-TUMORS

Zoom 75%

2) Supplementary figure 5C: It is not clear how this graph is calculated and what is being measured. Is this a Mander's or Pearson coefficient? It should be mentioned in the legend or the M&M. Regardless of this I find the fluorescent images to seem overexposed and not very informative. It would help if there could be more representative cells + each cell shown bigger.

We agree with the reviewer that there was no clear indication about how the graph was calculated. The percentage of MERCs was representing thanks to the Mander's coefficient (normalized to the mitochondria ara). As other critics arose from this specific panel (notably for not accurate ER staining), we reperformed experiments using more specific ER and specific mitochondrial antibodies, respectively Calreticulin (CALR) and ATP5A. The ER

is now better defined and we used Imaris to calculate in three 3D the percentage of colocalization of these two antibodies.

It is now better explained in both M&M and figure legend. In a specific subsection of the Material & Methods we say now "MERCs image analysis. Percentage of MERCs was evaluated through Mander's coefficient (normalized to mitochondria area) using ImarisColoc plugin of the software Imaris 9.9.0 (Oxford instruments, Abingdon, Oxfordshire, England, United Kingdom). Specifically, the indicated parameter was calculated in Z-stack micrographs from at least X cells (from X independent biological replicates) immunostained for Calreticulin (ER marker) and ATP5 (mitochondria marker), captured with the confocal microscope system Zeiss LSM 880. For the three-dimensional representation, a binary threshold (35/255 a.u) based on JACoP-Costes' method (DOI: 10.1529/biophysj.103.038422) was applied for both channels, which were subsequently overlapped. The merged images were opened with the 3D View tool of Imaris 9.9.0 (Oxford instruments, Abingdon, Oxfordshire, England, United Kingdom) and normal shading and white background were applied. Finally, snapshots and animations were carried out."

In the figure legend of Supplementary Figure 6d it says now: "...This value represents percentage of thresholded Manders' coefficient A as indicated as % of MERCs..."

3) Figure 6B: It is not clear to me how the authors blot for "PKA substrates" as the antibody or method is not described.

We used an antibody targeting specific phosphomotifs targeted by PK, namely RRXS*/T*). As indicated in the data sheet of this antibody: "Rabbit mAb detects peptides and proteins containing a phospho-Ser/Thr residue with arginine at the -3 and -2 positions. It is a useful tool in identifying new substrates of PKA. The antibody recognizes other -3 arginine-bearing phospho-Ser/Thr peptides, such as substrate motifs for Akt and PKC, to a lesser extent. It does not recognize the non-phosphorylated substrate motif peptides." According to the manufacturer, this antibody is referenced for Western Blot use in at least 156 research papers, including notably DOI: 10.1038/s41467-023-39710-z, DOI: 10.1038/s41467-023-39715-8, DOI: 10.1126/sciadv.adh1069, DOI: 10.1186/s12964-023-01081-9, DOI: 10.1038/s41467-023-37585-8.

4) Figure 6C: No statistics shown even though N=3

We added the two other values for the two other independent biological replicates and performed suitable statistical test accordingly.

5) Supplementary figure 6C: The graph shows a lot of ITPR3 hits being enriched in CDK4KO, yet the text states they are all downregulated?

We thank the reviewer for this comment. We corrected the sentence to highlight that ITPR3 is indeed actually enriched in CDK4-KO cells.

6) In the text it is stated on page 9 "... which proved a direct effect of CDK4." The word proved is dangerous in biology, this should be changed to something less absolute.

We corrected this sentence. We removed the word "proved" and "direct" and replaced by "...highlighting a CDK4-mediated effect on this activity..." now Page 12.

7) Figure 7C: The images of the mitotracker green/red cells are not representative of the graph. I understand the authors want to indicate the difference visually, but this is a bit excessive. In addition, I wonder if using flow cytometry may not be a better way to quantify this? Could the authors provide some examples here of other studies that use this method? In addition to this, the manner of data analysis for this experiment is not described, neither in the text nor in the M&M.

We are sorry for this misunderstanding. We agree that the cells chosen in Figure 7C did not accurately represent the decrease observed in the plot. To address this issue, we have changed the representative images of Figure 7C in the new version of the manuscript, as well as their corresponding magnifications.

Regarding the proposal to use flow cytometry to quantify the global mitochondrial membrane potential (MMP) using the ratio of the two MitoTrackers instead of confocal microscopy, the first method was questioned (DOI: 10.1002/cyto.a.20033). This is because even after the best calibration of the fluorescence signals of the two markers in flow cytometry, the result may be affected by the indirect inclusion of mitochondrial mass in its measurement (larger/smaller mitochondria may show an increase/decrease in the fluorescence signal of the Red MitoTracker, which may change the ratio, but not necessarily due to a change in their MMP). Although it is possible to mitigate this limitation (DOI: 10.1002/cyto.a.23705), including mitochondrial mass in the calculation of the MMP via MitoTrackers ratio may be statistically risky in our case, since the CDK4-KO condition itself alters (increases) the size of the mitochondria (see Figure 3B-C). Therefore, instead of using a microscopic method analogous to flow cytometry that includes mitochondrial mass in the numerator and denominator (e.g., ratio of the two integrated densities [Mean Gray Value x Area]), we decided to measure the ratio between the two mean densities [Mean Gray Value]. This eliminates the influence of mitochondrial area, and instead it is the denominator itself (Mean Gray Value of MitoTracker Green) that serves to normalize the size of mitochondria between CDK4-WT and CDK4-KO cells, since MitoTracker Green stains all mitochondria similarly, regardless of their MMP.

Some examples of other studies using and/or proposing this method include: DOI: 10.1186/s13075-019-1974-z ; DOI: 10.1002/cyto.a.20033 ; DOI: 10.1002/0471143030.cb0425s46

The way of data analysis for this experiment is described in the new version of the manuscript, both in the text and in the Material and methods section, and some references have been included

Reviewer #2

The authors investigate how the loss or inhibition of CDK4 leads to resistance in triple negative breast cancer cells. This is of course an important question clinically and the underlying molecular mechanisms have not been identified. The results indicate that CDK4 affects or regulates mitochondria-ER interactions and this may be mediated by PKA.

Overall, this manuscript contains a large amount of data and most experiments have been done well with the appropriate controls. There are just a few points that need to be improved in order to make the results more robust.

This work reports important results and with appropriate revisions, this manuscript would be a good contribution to Nature Communications.

Here are the issues that need to be addressed:

1. Figure 1; there are questions regarding the phosphorylation of S780 in RB. In the CDK4-KO cells, total RB is much higher than in the controls and S780 is also increased, nevertheless the ratio is decreased. In the Abema treated cells, both total RB levels and S780 levels are decreased compared to Veh but the ratio is only decreased at D3. In addition, both CDK4 and CDK6 levels are increased in Abema treated cells. All of this is a little confusion and one has to wonder what is the level of CDK2 activity? In addition, this reviewer wonders if the right experiment would be to treat CDK4-KO cells with Abema....

We appreciate that this reviewer raises this question, and we agree with the reviewer that these data may be confusing. We systematically observe an increase in total RB, as well as phosphorylated RB in the CDK4 KO cells, not only in this specific breast cancer cell line, but also in other cell types, such as mouse embryonic fibroblast. It is interesting to notice, however, that this is not the case when both CDK4 and 6 are inactivated by Abemaciclib, suggesting that the raise in RB total levels is mediated, directly or indirectly, by CDK6. The compensating increased CDK6 activity could contribute to a regulatory feedback that would facilitate RB gene expression, possibly mediated by the E2Fs transcription factors. Indeed, we show in the manuscript (sup. fig. 1) that the expression of cyclin D1 and D3 is increased in the CDK4 KO cells.

Concerning the lack of changes in Rb phosphorylation at day 23, previous studies showed, similar to our results, that after prolonged treatment with CDK4/6 inhibitors, cells become resistant and increase RB phosphorylation (doi [10.1158/0008-5472.CAN-15-0728](https://doi.org/10.1158/0008-5472.CAN-15-0728)). This is also observed in preclinical and clinical studies (DOI: [10.1016/j.ccell.2020.03.010](https://doi.org/10.1016/j.ccell.2020.03.010)).

We agree with this reviewer that CDK2 could compensate the lack of CDK4/6 activity. Following reviewer's suggestion, we analyzed the expression and activity of CDK2. We found that in CDK4 KO, or Abemaciclib-treated cells, the expression of CDK2 was not higher. However, the inhibition of CDK2 activity using AUZ-454 blunted RB phosphorylation, suggesting that CDK2 compensates, at least partially, for CDK4 deletion as far as RB phosphorylation is concerned. All these results are now summarized in Supplementary Figure 1a-b.

Finally, we treated CDK4-KO cells with Abemaciclib and observed that cell death resistance remains unchanged, highlighting the preponderant role of CDK4 on CDK6 to mediate cell death. These new results are now presented in the new figure 2e.

2. Figure 2C; similar than the first question; why is S780 increased across the board including in Veh? Also, this data should be quantified. It is also contrasting with 2F, which needs explanation. Could this be due to CDK2 activity?

Phosphorylation of RB at S780 was quantified in Figure 1b (last panel). According to the level of this phosphorylation in CDK4-KO treated cells upon CDK4/6 inhibitor and CDK2, we hypothesized that CDK2 was not necessarily more active. Furthermore, CDK2 activity remains unchanged as analyzed by the kinome profiling

(PamGene; Figure 7a), and also by the Integrative Inferred Kinase Activity (INKA) analysis from phosphoproteomics data. This is in contrary to PRKACA, as highlighted in the following figure.

3. Figure 2G; there is an issue with the D14; CDK4-KO / Veh because the data is spread widely, especially compared to all the other points (black/blue/green). Therefore, the red data points cannot be trusted and the average is misleading. This of course weakens the point the authors are trying to make.

We agree that at day 14, the data points were dispersed, specifically in the CDK4 KO tumors. This was due to a limited number of mice and high mortality because the high dose used of the treatment. To address this issue we performed new experiments to increase the number of mice (from N=4-5 to N=8/group) and thus decreased the dose of cisplatin (from 8 to 4 mg/kg). We show now in the new figure 2 more consistent and statistically significant results, especially for the WT-tumor xenografts where the cisplatin effect is clear. Furthermore, we reduced the data dispersion at day 14 in KO-tumor conditions and still confirmed the non-significant reduction of CDK4-KO tumor volumes upon cisplatin treatment.

4. Figure 3; did the authors measure mitochondrial DNA content as an alternative determination of the number or size of the mitochondria?

Following reviewer's suggestion, we quantified mitochondria-encoded genes, namely *ND1* and *16S*, by qPCR and normalized to the nuclear-encoded gene *18S*. Based on these two mitochondria-encoded genes, we observed an enhanced content of mitochondrial DNA in CDK4-KO cells, compared to -WT cells. This result is now displayed in Supplementary Figure 4b and detailed in the main text.

5. Page 8; "These results suggest that CDK4 is required for ER-MT calcium signaling in TNBC cells." This should be rephrased since CDK4 could either directly or indirectly be involved in this and the authors do not want to mislead the readers.

We have now rephrased this sentence, and we are now saying that "CDK4 inactivation ultimately results in decreased ER-MT calcium signaling in TNBC". Page 10.

6. Page 10; "...PKA regulatory subunit PRKAR1A in MERCs (Fig. 6F) and..." This is not shown in 6F and therefore maybe it is "Sup. Fig. 6F"?

We are sorry about the confusion. We are indeed showing the expression of PRKAR1A in the western blot in new Figure 8a.

7. Figure 6G is hard to read. What do the authors want to show here?

We agree with this reviewer that the panel 6G is not well illustrated. This was an analysis of recurrent motifs in the phosphoproteomic data of MERC fraction. We were showing here that among all the phosphopeptides found deregulated, there was a significant enrichment of peptides containing CDK4 kinase motif, but also AKT and PKA kinase motifs compared to the background. We did not find another way to represent this important result but : 1°/ we simplified the figure and 2°/ we have now better explained this in the figure legend as following:

“Legend Figure 8d (d) Enrichment analysis on phospho-peptides found in MERCs fraction of CDK4-WT and -KO TNBC cells. Phosphosites displaying CDK4 kinase motifs, AKT kinase motifs, CDK1,2,4,6-kinase motifs or PKA kinase motifs are represented. Bck=Background represents the total number of phosphosites found in the phosphoproteomics while Diff.Ref.=Differentially regulated represents the number of phosphosites found significantly down or up-regulated in MERC fraction of CDK4-KO TNBC cells. N=3 replicates. Benj. Hoch. FDR value is displaying for each substrate motif category.

We also divided the previous figure 6 et two new figures (now 7 et 8), while the Figure 7 deals with general PKA activity in our model and Figure 8 is focused on PKA activity specifically at MERCs interface, and how it can modulate ER-MT calcium transfer.

8. From all the data, the part that is the weakest is to show that the interactions of the mitochondria to the ER are altered. Any data the authors could add would be beneficial...maybe they can get some inspiration from <https://doi.org/10.1038/s41586-023-06956-y>

We agree with this reviewer that more efforts could be directed to prove the mitochondria-ER interactions. Unfortunately, we failed analyze three-dimensional electron microscopy data (as described in the suggested reference by the reviewer). However, to strengthen this part on interactions between ER and mitochondria, we added new data on Proximity ligation assays and co-immunofluorescence analyses.

Firstly, we performed a new Proximity Ligation Assay experiment targeting two other tethers at MERCs, namely VAPB (ER-resident protein) and PTPIP51 (OMM protein) knowing to interact and localize at MERCs (PMID: 28132811). As displayed now in new Fig 6e, we observed a decreased of VAPB-PTPIP51 puncta in CDK4-KO cells, suggesting a loss of mitochondria-ER contacts. In the new Sup. Figure 6e-f, we also added multiple negative and positive controls in our both PLA experiments (VAPB-PTPIP51 and ITPR1-VDAC1), displaying a high specificity of these two independent assays.

Secondly, we extended our 2D experiment observations with 3D analysis through co-immunofluorescence using 3D Imaris Software (Sup. Figure 6d). This new technique is now described also in Material & Methods section.

Reviewer #3

In the manuscript by Zeigler et al., a comparison of CDK4-WT vs CDK4-KO MDA-MD-231 is performed, under the conclusion that CDK4 is not required for proliferation, therefore, cell phenotypes in CDK4-KO cells are independent of proliferation changes. Overall, the relationships between chemosensitivity and metabolism are important, and the context of these pathways through MERC has the potential to reveal both physiological and patho-physiological contexts for organelle tethering. Within the manuscript, there are many observations and pathways that are interrogated, yet this broad approach fails to return to the primary objective of the manuscript and appears to be highly unfocused with copious observations that are weakly linked between the figures and not related to the disease model system.

We appreciate the reviewer's comments and understand the concern regarding the focus and coherence of our manuscript. Our primary objective was to explore the role of CDK4 beyond its well-known function in cell proliferation, particularly in the context of metabolic regulation and mitochondrial function in triple-negative breast cancer (TNBC) cells. We aimed to demonstrate that CDK4 inactivation not only impacts cell proliferation but also significantly alters mitochondrial-endoplasmic reticulum contacts (MERCs), thereby influencing metabolic vulnerabilities and resistance to cell death.

To address the reviewer's concerns about the perceived lack of focus, we have reorganized the results section to more clearly connect our findings to the central hypothesis. Specifically, we have ensured that each figure directly supports the narrative of CDK4's role in MERC regulation, mitochondrial dynamics, and their implications for TNBC cell survival and metabolic adaptation. We have also added more clinical data to better relate our findings to the TNBC disease. We have emphasized how the observed metabolic vulnerabilities in CDK4-KO cells could be exploited therapeutically, thus bringing the study's findings back to their potential clinical implications.

We hope these revisions will address the reviewer's concerns and clarify the manuscript's focus on the critical role of CDK4 in TNBC beyond its traditional involvement in cell proliferation.

Major concerns:

A single parental cell line is used in the study, and no clinical data/samples are investigated to support. Also, the metastatic cells don't express ER, PR, or E-cad, and harbor mutant p53, so the consequences of CDK4 deletion need to be positioned in this context, and perhaps investigated without all the confounding mutations.

We thank the reviewer for their insightful comments. We agree that using a single cell line is not fully representative of triple-negative breast cancer (TNBC). Due to time and space constraints, it is not feasible to repeat all experiments with multiple models. However, we have addressed this concern by testing the resistance to apoptosis in other breast cancer cell lines. Unfortunately, we were not able to find any p53-WT TNBC cell lines to test the role of p53 in acquiring this cell death resistance. Nevertheless, we tested cell death resistance with RB-proficient (HCC1806 and BT-474) and RB-deficient (MDA-MB-468) TNBC cell lines, and a control ER+/PR+ (MCF-7) cell line. All these cells showed a response to CDK4/6inhibitor as evidenced by reduced phosphorylation of S780 (Sup. Fig 2h). While RB-proficient cells display cell death resistance to chemotherapies after CDK4/6i pretreatment, RB-deficient TNBC and ER+/PR+ cell lines do not display these antagonistic effect (Figure 2g and Sup. Figure 2i). Altogether, these additional experiments demonstrated that CDK4 inhibition results in cell death resistance preferentially in RB-proficient/ER-/PR- cell lines, and regardless of p53 status.

Furthermore, we have utilized a more relevant cellular model: triple-negative breast cancer organoids derived directly from a breast cancer patient, as first described in our new data, presented in Figure 2f, show that CDK4 inhibition similarly confers resistance to apoptosis in this organoid model, further supporting our findings.

The reviewer also raised a valid point regarding the lack of clinical sample investigation. In order to strengthen this clinical part of the study, we generated a full new figure with clinical data analysis regarding our study (Figure 9). While TNBC patients are not typically treated with CDK4/6 inhibitors, we conducted an in-depth analysis of clinical data from the large study made by Sweden Cancerome Analysis Network - Breast (SCAN-B) (PMID: 25722745), comprising 2,929 breast cancer patients. Our analysis revealed that the expression of cyclins D, regulating CDK4 activity, and particularly cyclin D2, is positively correlated with increased survival in TNBC patients. Accordingly, this increased survival was related to a positive correlation with hallmark of apoptosis in these samples. This positive correlation between Cyclins D expression (thus CDK4 activity) and hallmark of apoptosis was also found in another dataset of ER- breast cancer (regardless of HER2) samples from the Cancer Genome Atlas (TCGA).

To evaluate response to neoadjuvant chemotherapy, we then analyzed another cohort of 227 patients breast cancer patients (PMID: 33268821), where biological samples were collected before and after neoadjuvant chemotherapy (NAC) 3 weeks after and for up to 6 months depending on the pathological Complete Response (pCR) status of the patient. Based on the n=156 MERCS-associated protein found (Sup. Table 1), we proposed a MERCS signature that was enhanced after NAC treatment, such as CyclinD expressions (Figure 9e). Interestingly, we also evidenced that PKA-associated signatures were particularly enhanced in patient with pathological complete response 3 weeks after NAC. (Figure 9f-g).

In contrast, our new analysis did not find the same positive correlation between CDK4 expression and survival in TNBC patients as reported in the previous version of our manuscript (previously former Sup. Fig 2a). The previous analysis used Affymetrix data, while the new analysis utilized RNA sequencing data. We believe this discrepancy underscores the importance of post-translational modifications and interactions with regulatory subunits, such as cyclins, in regulating CDK4 activity, rather than its expression levels alone. We erased then this former Sup. Fig 2a to build this new Figure 9.

Altogether, we think that this new figure enhances the whole clinical relevance of our study.

The authors show that CDK4-WT and CDK4-KO grow similarly in culture, but vastly different in xenografts within a partial microenvironment (e.g. macrophage, dendritic cells are likely present). Given this observation, it appears unsubstantiated to disregard the proliferation changes in vivo without detailed interrogation as they indeed demonstrate a lack of proliferation for 3 weeks. For example, does macrophage depletion eliminate the three week lag? Also, if the CDK4-KO cells metastasize immediately after implantation, the primary site would be expected to grow slower; this is also not addressed.

We thank the reviewer for this important observation and agree that immune system-related issues could potentially explain the lack of proliferation of CDK4 KO cells in vivo for a period of time. However, the participation of the immune system may be limited in our model, because NSG mice have defective macrophage and dendritic cell function.

In contrast, we fully agree with the reviewer that increased migration of the CDK4 KO cells could explain the refractory period in tumor growth. Data from an independent project in our lab indicates that CDK4 KO cells exhibit increased migration compared to WT cells. In vitro experiments using a Boyden chamber show that KO cells migrate faster than WT cells. We have data to suggest a direct effect of CDK4 on the actin cytoskeleton. Moreover, in a preliminary experiment, we analyzed the presence of metastasis in this mouse model, and we found that some mice grafted with CDK4 KO cells formed metastasis in liver and lymph nodes, whereas non of the mice grafted with WT cells developed metastasis. This result should be taken with caution because the experiment was not specifically designed for metastasis studies. We are currently performing in vivo experiments using luciferase-expressing CDK4 WT and KO cells to monitor metastasis at early points post-grafting. We have now included a discussion of these findings in the text.

The cell death studies employ some unusual inducers, for example oligomycin and antimycin without rationale (also, why treat TNBC with peroxide?). As the mitochondrial pathway is of interest to the authors, some demonstration beyond cleaved caspase-3 (which cleaves either mechanistically or consequentially given a variety of cell death pathways) is necessary.

To address the critic that this reviewer raised, we have now included in the manuscript new data using other distinct pro-apoptotic and more classical chemotherapeutic agents used in TNBC context, including doxorubicin (DOXO), 5-Fluorouracil (5-FU) and other known apoptosis inducers, such as UV_B treatment (UV_B) and TRAIL. We show now that in a first screen (Supplementary Fig 2a-b). Since CDK4 KO cells were particularly resistant to cell death induced by H2O2 and O+A, we chose these treatments for further experiments.

We also have divided the old Figure 2 et two new figures, now Figure 3 and Figure 4. The Figure 3 now focused on the resistance of CDK4-KO and CDK4/6i pretreated cells to chemotherapies, adding evidence also in other cell lines, patient-derived xenograft cells and *in vivo*. The Figure 4 now includes some mechanistical aspects proving that mitochondrial pathway of cell death is deregulated in CDK4-KO cells.

The previous section on “apoptosis” is now divided in two new subsections describing the results of these two figures.

Does eliminate of BAK/BAX block death, does zVAD-fmk delay death, etc. Figure 2 shows that 40% of CDK4-KO cells are dead, yet no C3 cleavage; same with Cisplatin, most CDK4-KO cells are dead in figure 2b, yet the authors don't provide explanation.

We thank this reviewer for this important comment, and we agree that mechanisms other mechanisms than apoptosis could also participate in the resistance to cell death observed in the CDK4 KO cells. Following this reviewer suggestion, we provide now new data showing that zVAD treatment only partially rescued H₂O₂- and O+A-induced cell death (new Fig 3f), implicating additional cell death mechanisms in response to these treatments. We now removed the term apoptosis to integrate a general cell death term that reflects more the reality of CDK4-KO resistance. We discussed that this cell death resistance is partly mediated by apoptosis (as seen with z-VAD + Cisplatin or Doxorubicin treatment in WT-cells, that mimic the resistance in KO cells, but this cell death resistance is extended as seen with zVAD + H2O2 or O+A treatment in WT-cells, that do mimic this resistance.

We also added data concerning expression of pro and anti-apoptotic members, respectively BAK, BAX, cytochrome c and BCL-2, BCL-XL, in CDK4-WT and -KO (now in Figure 3c). We observed a significant increased of most of these proteins, notably BCL2 and BCL-xL and BAX, but not BAD or cytochrome c. As the way to demonstrate the involvement of BAX/BAX in our cell death conditions is to use MEF double KO for these proteins, we thought that these models were probably beyond the scope of our study and for sure would not fit with our TNBC cell models. Furthermore, as the study is now first focused on chemotherapies, multiple other studies demonstrated the implication of BAX/BAK in these chemotherapy-induced cell deaths (cisplatin with PMID: 26996126, 5-FU with PMID: 9792140, Doxorubicin with PMID: 12193597).

No evidence of equal stress within the cells are provided; for example, does cisplatin cause the same number of DNA lesions independent of genotype.

We thank the reviewer for this comment. To determine the DNA lesions caused by stress and whether they are independent of genotype, we stained CDK4 WT and KO cells for GH2AX, an early marker of DNA double-strand breaks, highly specific for monitoring DNA damage initiation. We show now no significant differences in the DNA damage induced by cisplatin chemotherapy between CDK4 WT and KO cells. This is now shown in new Figure 3 (panels a and b) and described in the text.

Many of the cell death studies also disregard more modern mechanisms of the intrinsic pathway and suggest that calcium signaling is essential for apoptosis, yet the mitochondrial pathway and PTP are not mechanistically linked.

To address this reviewer concern, we have now investigated the role of CDK4 in MOMP, a critical and irreversible step in apoptosis characterized by the release of proteins from the intermembrane space into the cytosol. Minority MOMP, where only a subset of mitochondria undergo permeabilization, can result in failed apoptosis. Our results show that CDK4 wild-type (CDK4-WT) cells experience a significant drop in mitochondrial membrane potential after O+A treatment, indicative of efficient MOMP. In contrast, this decline in membrane potential was notably limited in CDK4-KO cells, suggesting that the absence of CDK4 diminishes MOMP efficiency and thus hampers the execution of apoptosis. This is now shown in the Figure 3 (Panels k and l).

The immunofluorescence for mitochondria could benefit with clearer markers/image capture, as data in figure 3D and S3K are blurry, and the images don't match the quantification of mitochondria presented in figure S3...these cells have more than 25 mitochondria, on average.

We apologize for the lack of clarity, probably due to low-quality pictures. We updated better quality images of mitochondrial markers in the new figure 4 and supplemental figure 4. We are also sorry for the misunderstanding regarding the number of mitochondria. Indeed, 25 mitochondria were counted in the experiment of EM, but not in the IF analysis. This is now better explained in the figure legend.

The calcium studies are interesting, but not presented in a context of the apoptosis or metabolism.

We thank the reviewer for this comment. We presented the calcium studies, notably the mitochondrial calcium uptake, in response to H₂O₂ and O+A stimuli, and demonstrated a dampened mitochondrial calcium accumulation in KO cells, in correlation with reduced cell death (new Figure 3g-h-i-j). In addition to these first results, we designed an experiment to assess whether ER-MT calcium fluxes may be determinant in the induction of cell death, pretreating cells with combination of siRNA against some key ER-MT calcium channels, namely ITPR3, VDAC1 and MCU. All these results are now presented in Supplementary Figure 5 i-j-k and showed that the combined knocking down of ITPR3/VDAC1 or VDAC1/MCU are able to partially block cell death in CDK4-WT cells. Taken together, these results strongly suggest a ER-MT calcium flux component in the induction of cell death in TNBC at least through H₂O₂ and O+A treatment.

ER Tracker is not a reliable marker to perform co-localization studies as it is not perfectly localized...this is demonstrated in figure S5C, as the ER signal is throughout the majority of the cells. A different marker or imaging capture technique (Imaris, STED, etc) is necessary to establish conclusions.

Following this reviewer suggestion, we used another technique through co-immunostaining of mitochondria and ER with two specific antibodies, namely Calreticulin for ER and ATP5 for mitochondria. We also explore colocalization of these two ER and mitochondrial signals, using 3D Imaris Software. All these data are now shown in new Sup. Figure 6d.

The metabolism studies could benefit from bioenergetics studies to report oxygen consumption, OCR, ECAR, mitochondrial ATP generation, etc. As presented, the metabolism work is cataloging changes with no return to the focus of the manuscript on heightened metabolic sensitivity (although it is not clear what this means or if it's observed in TNBC).

We thank the reviewer for this comment. To address the bioenergetic status of CDK4-WT and -KO, cells, we performed experiments using Seahorse technology and added these data in new Figure 10. We found that CDK4-KO cells did not display major changes in OXPHOS capacities at least in basal conditions (20mM Glucose, Pyruvate and Glutamine), where cells appeared to be more glycolytic (Sup. Figure 10c). We then challenged the cells with distinct media to force cells to rely on oxidative phosphorylation. This was achieved by replacing glucose by galactose or by totally suppressing glucose (Figure 10g-h). Under these conditions, CDK4 KO cells displayed less flexibility than CDK4 WT cells to boost oxidative metabolism, as measured by oxygen consumption. (Fig 10g-h and Sup. Fig 10c-d). All these data are incorporated now in Figure 10.

Reviewer #4

In the manuscript "CDK4 inactivation balances resistance to apoptosis with heightened metabolic activity in triple negative breast cancer" Ziegler and colleagues aim to understand how CDK4 regulates the fate of TNBC tumor cell lines and cell line xenografts. They identify that using CDK4 genetic knock out results in reduced apoptosis and does not inhibit xenograft growth because CDK4 enhances mitochondria-ER contact and alters mitochondrial calcium flux and apoptosis inhibition.

This paper answers an important question, which is why CDK4/6 inhibition, so clinically impactful in ER positive breast cancer, fails in ER negative breast cancer. They used a wide array of experimental and analytic approaches to reach this conclusion. They used drug treatment with cisplatin and metabolic stress to show that CDK4 KO protects MDA-MB 231 cells from apoptosis. They found decreased calcium flux and found decreased MERCs using transmission EM. Proteomic analysis demonstrated that MERC tether proteins were downregulated. This effect appears dependent on PKA alpha, mediated through effects on calcium related channels. Overall, the main conclusion of the paper i.e. CDK4 KO results in apoptosis resistance via increased MERC formation and alterations in ER_MIT calcium signaling is well supported by the data. However, prior to publication, certain clarifications could help improve the overall story.

- What is the evidence that CDK4 directly phosphorylates PKA alpha? The authors have shown that CDK4 activity is necessary and sufficient for PKA phosphorylation but is this direct or through an intermediary kinase? This detail would be important to determine that the CDK4 KO effect is specifically related to loss of CDK4 activity alone?

We thank the reviewer for this comment. We agree with the reviewer that we did not address in the first version of the manuscript the causative effects of the observed decrease in PKA activity in CDK4KO cells. Phosphoproteomics analysis (Supplementary Figure 8b) showed that PKACA, which is the catalytic subunit, did not contain any CDK4 phosphorylation site. This suggested that PKA itself was not a direct target of CDK4. Further analysis showed, however, that the PKA regulatory subunit 1A (PKAR1A) phosphorylation at serine 83, which is a consensus CDK4 phosphorylation site, was decreased in the CDK4 KO cells

These new results strongly suggest that CDK4 regulates the activity of PKA through phosphorylation of the regulatory subunit PKAR1A.

- How does CDK4 KO affect the levels of mitochondrial proteins involved in apoptosis regulation as BIM, BAX, BCL2 and PUMA? The authors show that there is decreased mitochondrial membrane potential in CDK4 KO cells. Does this result in reduced outer mitochondrial membrane permeabilization as a means of reducing likelihood of apoptosis? These additional data would help better understand why CDK4 KO alters apoptosis thresholds in addition to altering calcium flux? See <https://www.ncbi.nlm.nih.gov/pmc/articles/PMC7325165/> for more details on this functional approach.

We thank the reviewer for this important point. To address this reviewer's question, we blotted for some main anti-apoptotic proteins BCL2, BCL-xL and pro-apoptotic proteins BAX, BAD and Cytochrome c (now in Figure 3c-d). We observed that the two main anti-apoptotic proteins BCL2 and BCL-xL, but also the pro-apoptotic protein BAD, are upregulated in CDK4-KO cells. In contrary, BAX or Cytochrome c are not differentially regulated. We concluded that these variations in the expression of these proteins may explain some of the cell death resistance mechanisms of our study. These results were incorporated into the new Figure 3 depicting how CDK4 affects mitochondrial effectors of apoptosis (with caspase-3 cleavage, mitochondrial calcium uptake and Mitochondria Outer Membrane Permeabilization).

- The authors show that CDK4 KO alters isocitrate, alpha-ketoglutarate, malate and succinate levels and that galactose reduces CDK4-KO cell viability. This is an intriguing observation but not well fleshed out and seems to detract from the overall flow of the paper. What is the impact of altered metabolism on apoptosis vulnerability might be one way to tie these strands together.

We thank the reviewer for this comment and apologize for the confusion. It is important to notice that the calcium dyshomeostasis disrupts both mitochondrial metabolic and apoptotic activities, both mediated by calcium signaling. We do not suggest or demonstrate that altered metabolism impact apoptosis. Indeed, we show in the new version of the manuscript that the cell death that we observed in CDK4 KO cells in response to galactose was not mitochondria and apoptosis-mediated, as demonstrated by the analysis caspase 3 cleavage, showing no changes in cells culture in media without glucose (Gal-/Glu-) or with galactose instead of glucose (Gal+/Glu-). In summary, we only proved that defects in calcium signaling induce resistance to apoptosis, but also metabolic dysfunction in CDK4 KO cells.

REVIEWER COMMENTS

Reviewer #1 (Remarks to the Author):

I have gone through the author's responses, and overall, it is clear the authors have approached the comments in a serious and systematic way, which I appreciate. A lot of extra work has been done that has resolved most of my comments. However, even though I am not a fan of multiple rounds of revision, there are some issues that I feel still need to be addressed.

Reviewer 1's thoughtful and comprehensive review is greatly appreciated. Below, the specific questions and remarks raised are addressed:

- Fig 3 K – L : There seems to be a discrepancy in the differences between the graphs, where 3K shows quite some variability in the WT, this is almost gone in 3L. Are these experiments done independently from each other? Why?

We appreciate the reviewer for drawing attention to this point. We revisited both our raw and analyzed data and did not identify any issues with the analysis. The reduced variability may stem from the fact that 3L is a ratio derived from the 3K data, which inherently affects the variability as shown in the following calculations:

Fig3k		TMRM FI / Mitotracker Green FI				Fig 3l	
Biological Replicates	CDK4-WT		CDK4-KO		Biological Replicates	% of reduction (TMRM / Mitotracker green FI) upon O+A	
	Veh	O+A	Veh	O+A		CDK4-WT	CDK4-KO
#1	0,584884	0,128484	0,354226	0,124828	#1	78,03256714	64,76035074
#2	0,651854	0,137213	0,505082	0,262297	#2	78,95034778	48,06848383
#3	0,332259	0,092791	0,436663	0,196356	#3	72,07269028	55,03260897
			0,780325671	0,647603507			
			0,789503478	0,480684325			
			0,720726903	0,550325995			

Here is a screenshot of the raw data for these two panels. (available in the datasheet Source data.xls). To calculate the ratio the formula was:

$(\text{TMRM/MT tracker FI (Veh)} - \text{TMRM/MT tracker FI (O+A)}) / (\text{TMRM/MT tracker FI (Veh)})$, as following:

$$\text{CDK4-WT(\#1)} = (0.584884 - 0.128484) / 0.584884 = 0.7803 = 78.03\%$$

$$\text{CDK4-WT(\#2)} = (0.651854 - 0.137213) / 0.651854 = 0.7895 = 78.95\%$$

$$\text{CDK4-WT(\#3)} = (0.332259 - 0.092791) / 0.332259 = 0.7207 = 72.07\%$$

- Sup. Fig 5a: I understand that the authors took the probe used to measure absolute calcium levels and subjected cells expressing the probe to histamine. However, this figure panel is confusing and weakens the message. It is not referenced in the text anywhere, and seems

a (less convincing) copy of fig 5F. Furthermore, it seems to contradict the result from figure 5a, where 5A shows a difference in basal calcium, fig S5A now shows an identical starting value. How can this be?

We thank this reviewer for the comment. We agree that the original presentation could have been clearer, and we have made several revisions to address the concerns raised.

We have reorganized Supplementary Figure 5 to improve clarity. Specifically, we have consolidated the non-ratiometric measurements for mitochondrial calcium (previously Sup. Fig 5d-e) into the new **Sup. Fig 5a-b**. In addition, the former Sup. Fig 5a is now presented as **Sup. Fig 5c**. We explain now in the text that "The ratiometric mitochondrial calcium probe displayed, to a lesser extent, the same reduced maximum peaks and immediate ER-to-mitochondria calcium fluxes in CDK4-KO cells (Sup. Fig 5c-d). However, no difference in the secondary ER-to-mitochondria calcium transfer was observed in CDK4-KO cells with this ratiometric probe, as evidenced by a similar area under the curve (Sup. Fig 5e). This suggests mid-term compensatory effects due to enhanced ITPR1, ITPR3, and VDAC1 expressions in CDK4-KO cells (Fig 5b)"

We have also addressed the contradiction in basal calcium levels. In the experiments presented in Sup. Fig 5c, we focused on measuring the dynamics of mitochondrial calcium upon histamine stimulation. To accurately compare the calcium flux between WT and CDK4-KO cells, we normalized the YFP/CFP ratio to 1 at the time point immediately before histamine injection. This normalization aligns the starting values and allows for a direct comparison of the calcium responses following stimulation. We have updated the figure legend "Representative curve based on 4mtD3CPV fluorescence (Ratio YFP/CFP) and **normalized to baseline before injection** from N=3 independent biological replicates...", and adjusted the x-axis label in Sup. Fig 5c to "**Relative mitochondrial calcium level (F/F_0)**" to clearly indicate that the data have been normalized. This should prevent any misunderstanding regarding the starting values. In Figure 5A, the difference in basal calcium levels observed represents absolute measurements without normalization, highlighting physiological differences between WT and CDK4-KO cells. In contrast, Sup. Fig 5c presents normalized data to specifically analyze the calcium dynamics upon stimulation, which is why the starting values appear identical.

The ratiometric probe data in Sup. Fig 5c demonstrates similar patterns to the non-ratiometric measurements, confirming the reduced immediate ER-to-mitochondria calcium transfer in CDK4-KO cells. The absence of a significant difference in the secondary ER-to-mitochondria calcium transfer (as shown by the area under the curve) in CDK4-KO cells suggests a potential compensatory mechanism. We hypothesize that the enhanced expression of ITPR1, ITPR3, and VDAC1 in CDK4-KO cells (Fig 5b) may contribute to this effect.

- Sup. Fig 5i-k: "Finally, combined knockdown of ITPR3, VDAC1 and/or MCU was sufficient to rescue the partial cell death induced by H₂O₂ and O+A, but not cisplatin in CDK4- WT cells

(Sup. Fig 5i-k). "This statement seems too strong when looking at the figure. The data show a very minor increase in cell death, this should be reflected in the text, as the knockdowns are certainly not sufficient to rescue the effect.

We agree with this reviewer that these results were overstated in the text. We have now down tuned the sentence and say now in page 9 : "Finally, the partial knockdown of VDAC1 with ITPR3 or MCU (reduction of 30 to 79% protein expression) allowed a minor but significant rescue from 13 to 16% of the cell death induced by H2O2 and O+A, but not by cisplatin in CDK4-WT cells (Sup. Fig 5i-k)". We also include the quantitative metrics for knockdown efficiency and cell death rescue in the main text. These percentages are now also reflected in Supplementary Figures 5i and 5k."

- Comment on statistics: I recognize that the authors prefer to use a paired t-test but I have to disagree with their reasoning. It is not that a paired t-test is "typically" used for measurements of the same individual, it is that that is a necessary assumption. I feel the author's pain in having to deal with "unnecessary" variability due to cell competency, age etc, but this is not an excuse to use the wrong statistical test. I suggest the authors look into using non-linear mixed model statistics, which is able to deal with this additional variability, or use unpaired t-tests/ANOVA's.

We thank the reviewer for this important comment. We reviewed again our methodology with our biostatistics service to address this critic as follows.

The independence of biological replicates is one of the most important assumptions when using statistical tests such as the Student t-test. In the case of a batch-effect, this assumption of independence is not met, and the paired t-test is used to take this situation into account. We have reviewed several papers and textbooks in statistics; while an experiment where the same individual is measured twice is the most commonly-cited example of use of the paired t-test, they all suggest that other situations of non-independence can (and should) be handled using a paired t-test. This is discussed for example in the "SuperPlots" paper mentioned by the reviewer in the previous round, Lord et al. (PMID: 32346721) and in other highly cited papers such as "The Differences and Similarities Between Two-Sample T-Test and Paired T-Test" by Xu et al (2017)(PMID: 28904516) and "A study of clustered data and approaches to its analysis" by Galbraith et al., 2010) (PMID: 20702692). We found also this example in textbooks such as "Introduction to statistics and data analysis" by Peck et al. and "Essential Medical Statistics" by Kirkwood et al. As such and based on these numerous reference, we believe that our use of the paired t-test is correct.

Figure S1 of the “Superplots paper” (PMID: 32346721) with the use from B to F of paired T-tests. In the text it is mentioned : “By encoding the biological replicate into the data, such trends can be revealed without normalizing to a control group: P values can then be calculated using statistical tests that take into account linkages among samples (e.g., a paired or ratio t test). In fact, not taking into account linkages can make the t test too conservative, yielding false negatives (Galbraith et al., 2010).”

Not taking into account this non-independence between linked samples will often make the t-test too conservative (increasing the number of false negatives).

In order to avoid any bias and overinterpretation from our side, we did again all the statistical tests using a 2 Way-ANOVA and having factor 1 as biological replicate and factor 2 as genotype with N biological replicates. n were taken as technical replicates for each N. We put three examples (Fig 3h, 4d or 5a) here to show that it was only enhancing the statistical power of the test.

Fig 3h (Paired T-test)

Fig 3h (2-Way ANOVA)

Fig 4d (Paired T-test)

Fig 4d (2-Way ANOVA)

Two-way ANOVA		Ordinary			
Alpha		0.05			
Source of Variation	% of total variation	P value	P value summary	Significant?	
Interaction	0.5563	0.5212	ns	No	
Biological Replicate	19.56	<0.0001	****	Yes	
Genotype	8.107	<0.0001	****	Yes	

Two-way ANOVA		Ordinary			
Alpha		0.05			
Source of Variation	% of total variation	P value	P value summary	Significant?	
Interaction	0.1309	0.8281	ns	No	
Biological Replicate	28.90	<0.0001	****	Yes	
Genotype	11.19	<0.0001	****	Yes	

Fig 5a (Paired T-test)

Fig 5a (2-Way ANOVA)

Two-way ANOVA		Ordinary			
Alpha		0.05			
Source of Variation	% of total variation	P value	P value summary	Significant?	
Interaction	0.6313	0.0531	ns	No	
Biological Replicate	1.883	0.0002	***	Yes	
Genotype	3.588	<0.0001	****	Yes	

In the case that there was no difference for factor 1, namely biological replicate, for instance in Fig 3h, we replaced the paired T-test by an unpaired T-test (Fig 1d, Fig 3l, Fig 8g). For the other, we kept T-tests as the most accurate (not overstating) test for our data.

- In addition, why were some of the tests changed to a paired t-test, like in figure 6b,c? The paired t-test in that experiment is also not permissible in my view.

To address the remarks of the reviewer (and also suggested by the “Super Plot” paper) we decided not to take anymore n=number of cells or mitochondria to do the statistical test. We clustered by biological replicates. We also noticed that we forget to cluster the data by independent biological replicates for Fig6d (we updated the Figure), as we did for Fig 6b and 6c and Sup. Fig 6a.

- As a side comment: I would like to recognize the author's honesty and scientific integrity in displaying openly when a statistical test shows a borderline p value (ie 0.0507 or so) instead of trying to massage the data.

We appreciate this reviewer's comment.

- Figure S6d: I appreciate the extra work done by the authors, and the methodology seems ok to me now. However, I am not really satisfied with the staining. Not only does the Calreticulin seem too punctate, there is a marked difference between WT and KO in staining intensity. Is this a representable difference? Staining intensity may affect Mander's independent of any contact site difference. Please have another look at this and carefully evaluate your ER staining and any differences between WT and KO.

The reviewer raises an important point that we initially overlooked. Upon re-examining our analyses, we found that calreticulin fluorescence intensity was indeed reduced in the CDK4-KO, as the reviewer correctly noted, which may have biased the co-immunofluorescence analysis with this antibody. Given the challenges we previously faced in staining the ER, we expanded our search for a reliable ER-marker antibody to ensure accurate ER staining and Mander's correlation coefficient for co-localization. We opted for SEC61B (referenced in Figure 3 of PMID: 36973273) and Calnexin (CANX), both well-established ER markers. Proteomics data showed no difference in protein expression between CDK4-WT and -KO for these markers, and we confirmed that immunofluorescence conditions did not vary in basal intensity between the two cell lines. Though less pronounced than with CALR (and the biased effect previously mentioned), we still found that CDK4-KO cells exhibit a reduced Mander's coefficient compared to CDK4-WT for ATP5A-SEC61B and ATP5A-CANX. This data is now included in Supplementary Figures 6d (SEC61B) and 6e (CANX)

Minor comments:

- Graphical abstract: I like the graphical abstract, but the arrows and info that arise below the ER-Mito drawing is not clear enough I think. Please have a look at improving the clarity.

We thank the reviewer for this important remark. We added information to the different arrows (inhibition or stimulation) going down the MERCs scheme. We hope it reflects better and clarify our main results.

- This is probably a personal thing but I still find the word "proven", used throughout the text just too strong. It's also unnecessary as other words like show or indicate etc are fine.

In order to avoid any overinterpretation, we removed the word "proven" and replaced it by "displayed", "demonstrated" or "acquired" in the Discussion section, pages 16 and 19.

- Title: I appreciate the authors have taken up the suggestion of changing the title, but the new title seems to be missing some info. Perhaps the authors meant: CDK4 Inactivation

Hinders Cell Death and Metabolism through remodelling Mitochondria-ER contact sites in Triple-Negative Breast cancer cells ?

We appreciate the reviewer's suggestion. However, to meet the word limit requirements during submission, we removed "through" and "sites" from the title. In our view, the reviewer's proposed title and our original title are very similar in meaning.

Reviewer #2 (Remarks to the Author):

The authors have done extensive revisions and have improved their manuscript. I believe that Figure 9 (mostly added in response to a reviewer comment) should be moved to supplemental data since it is much less convincing than the other data. Other than that, it would be nice if the authors had provided an overview over all the changes that were made in order to make it easier for the reviewers.

We are thankful to the reviewer#2 for the different points addressed during the previous round of revision, and which allowed an improved overall quality of the current manuscript. Nevertheless, we apologize for the lack of clarity in the rearrangement and changes done since the first version of the manuscript. For the Figure 9, we think that this new figure includes a significant amount of clinical data that extend our mechanistic study model and increase the overall relevance of our study. Based on this statement, we decided to keep it with the main figures.

Reviewer #3 (Remarks to the Author):

The authors have addressed the vast majority of concerns raised during the review process, along with reorganizing the text and data presentation for increased clarity.

We thank this reviewer for his constructive comments through the whole review process.

Reviewer #4 (Remarks to the Author):

The authors have adequately addressed my concerns

We thank this reviewer for his constructive comments through the whole review process.